# Inherited deficiency of DIAPH1 identifies a DNA double strand break repair pathway regulated by γ-actin

DNA double strand break repair (DSBR) represents a fundamental process required to maintain genome stability and prevent the onset of disease. Whilst cell cycle phase and the chromatin context largely dictate which repair pathway is utilised to restore damaged DNA, it has been recently shown that nuclear actin filaments play a major role in clustering DNA breaks to facilitate DSBR by homologous recombination (HR). However, the mechanism with which nuclear actin and the different actin nucleating factors regulate HR is unclear. Interestingly, patients with biallelic mutations in the actin nucleating factor *DIAPH1* exhibit a striking overlap of clinical features with the HR deficiency disorders, Nijmegen Breakage Syndrome (NBS) and Warsaw Breakage Syndrome (WABS). This suggests that DIAPH1 may play a role in regulating HR and that some of the clinical deficits associated with *DIAPH1* mutations may be caused by an underlying DSBR defect. In keeping with this clinical similarity, we demonstrate that cells from DIAL (DIAPH1 Loss-of-function) Syndrome patients display an HR repair defect comparable to loss of NBS1. Moreover, we show that this DSBR defect is also observed in a subset of patients with Baraitser-Winter Cerebrofrontofacial (BWCFF) syndrome associated with mutations in *ACTG1* (γ-actin) but not *ACTB* (β-actin). Lastly, we demonstrate that DIAPH1 and γ-actin promote HR-dependent repair by facilitating the relocalisation of the MRE11/RAD50/NBS1 complex to sites of DNA breaks to initiate end-resection. Taken together, these data provide a mechanistic explanation for the overlapping clinical symptoms exhibited by patients with DIAL syndrome, BWCFF syndrome and NBS.

Actin is a small globular protein that can assemble into linear or branched filaments (termed filamentous or F-actin), which represent a major constituent of the cellular cytoskeleton. Of the six different actin paralogs present in mammalian cells, only β-actin and γ-actin are ubiquitously expressed[1]. However, despite near identical amino acid sequences, individual functions of β-actin and γ-actin in regulating cell motility, cell adhesion, vesicular transport, phagocytosis or mitotic progression can be differentially regulated via their association with specific factors that promote either microfilament formation, branching, bundling, cross-linking or disassembly[1]. The three major classes of actin-binding proteins that promote actin nucleation into filaments are the Arp2/3 complex, formins and Spire-family members, each of which exhibit distinct structural properties and regulatory mechanisms that determine their function[2]. The Arp2/3 complex is activated by the Wiskott-Aldrich Syndrome (WAS) family of proteins e.g., WASP, N-WASP and WASH, and primarily catalyses the nucleation of branched actin filaments[3]. In contrast, the Spire and formin domain-containing proteins tend to promote the formation of linear F-actin.

✉e-mail: g.s.stewart@bham.ac.uk

The Spire proteins form a seeding actin polymer for filament elongation through binding several actin monomers at once via their multiple WASP-homology 2 (WH2) domains[4]. Whereas in contrast, the formin proteins bind to the barbed end of an actin filament and promote polymerisation by bringing actin monomers in close proximity to the filament end[5].

DIAPH1 (diaphanous-related formin 1 or mDia1) is a member of the actin nucleator formin family of proteins, which are defined by the presence of their formin-homology, FH1 and FH2 domains[6]. The FH1 domain is required for the interaction with the actin monomer-binding protein, profilin, whereas the FH2 domain is responsible for actin filament nucleation[6]. The Diaphanous-related formins (DRFs) comprise a small subgroup of formins, containing three related proteins (DIAPH1-3) that are activated by binding Rho-type GTPases[5]. Typically the DRFs are involved in organising various cytoskeletal structures, such as cilia or the cytokinetic contractile ring[5]. However, more often than not they have non-redundant roles with each other despite being involved within the same cellular processes e.g., activation of DIAPH3 at the cytokinetic furrow triggers production of a β-actin contractile ring at the cell equator whereas DIAPH1 is localised to the cortex of the mitotic cell where it promotes the formation of γ-actin filaments to regulate furrow positioning and the rate of membrane ingression[7].

The importance of DIAPH1 in regulating distinct actin-dependent cellular functions initially stemmed from the identification of dominant activating mutations in *DIAPH1* as an underlying cause of non-syndromic hearing loss with/without thrombocytopenia[8,9]. Notably, all the identified mutations are located within the C-terminal diaphanous autoregulatory domain (DAD), which block the ability of the DAD to interact with the N-terminal diaphanous inhibitory domain (DID) resulting in a constitutively active protein[8,9]. More recently, biallelic inactivating variants in *DIAPH1* have been identified in a subset of patients presenting with Seizures, Cortical Blindness and Microcephaly Syndrome (SCBMS, OMIM:616632). This autosomal recessive neurodevelopmental disorder is characterized by microcephaly, early-onset seizures, severely developmental delay/intellectual disability and cortical blindness. Additionally, affected individuals commonly exhibit poor overall growth and short stature[10–14]. Notably, the clinical phenotype exhibited by patients with loss of DIAPH1 overlap with those displayed by patients with Baraitser-Winter Cerebrofrontalfacial (BWCFF) syndrome caused by de novo mutations in either *ACTB* or *ACTG1* encoding β- and γ-actin respectively (BWCFF1, OMIM:243310; BWCFF2, OMIM:614583)[15–23]. Despite the extensive clinical variability observed between patients with BWCFF syndrome, the presence of microcephaly, hearing loss, seizures, growth retardation and intellectual disability would strongly indicate that the developmental abnormalities resulting from an absence of functional DIAPH1 are caused by disrupting actin-dependent cellular functions.

Interestingly, the clinical features exhibited by patients with loss of DIAPH1 or missense mutations in β-actin or γ-actin, such as microcephaly, growth retardation, immunodeficiency, intellectual disability, seizures and hearing loss, are highly reminiscent of those typically associated with the DNA repair deficiency disorders Nijmegen Breakage Syndrome (NBS, OMIM:251260) and Warsaw Breakage Syndrome (WABS, OMIM:613398)[24–26]. This suggests that some of the clinical features arising from disrupting the actin nucleation pathway could be caused by an underlying DNA repair defect. Consistent with this, it has been shown that the WASP-Arp2/3-containing actin nucleating complex is required to mobilise and cluster DNA double strand breaks (DSBs) targeted for repair by the homologous recombination (HR) repair pathway[27,28]. Notably however, this process was reported to be independent of the Spire and Diaphanous family of actin nucleators[27]. More recently, it has been demonstrated that the WASP-Arp2/3-containing actin nucleating complex as well as DIAPH1 are required to maintain genome stability by protecting stalled replication forks from uncontrolled nucleolytic degradation[29,30]. Taken together, there is mounting evidence that actin filament formation is important for maintaining genome stability through multiple mechanisms, including promoting efficient DNA DSB repair (DSBR), replication fork stability and chromosome segregation. However, it is not clear which actin nucleators or actin paralogs play a major role in each of these processes.

Here we identify DIAL (DIAPH1 Loss-of-function) syndrome as a previously unrecognised radiosensitivity disorder associated with an inability to repair DNA DSBs induced by ionising radiation (IR) and the chemotherapeutic agents, camptothecin (CPT) and etoposide (ETOP). Interestingly, this phenotype is shared with a subset of BWCFF syndrome patients with mutations in *ACTG1* but not *ACTB* suggesting that DIAPH1 and γ-actin function within the same pathway to regulate DSBR. Lastly, we demonstrate that both DIAPH1 and γ-actin are required to promote HR and that this is mediated through their ability to promote relocalisation of the MRE11/RAD50/NBS1 (MRN) complex to DNA DSBs and their subsequent end-resection. Based on these observations, our data highlights a critical role for DIAPH1 and γ-actin in regulating DNA repair and indicates that a failure to repair DSBs may cause some of the clinical features exhibited by patients with DIAL syndrome and BWCFF syndrome that overlap with NBS and WABS.

## Results

### Biallelic mutations in *DIAPH1* cause a neurodevelopmental disorder with a range of clinical symptoms

A single patient (P1) exhibiting microcephaly, behavioural abnormalities, intellectual disabilities, cortical blindness, recurrent infections, a reduction in immunoglobulin levels, seizures and a significant midline facial granuloma with a suspected diagnosis of NBS was referred for genetic testing. Following whole exome sequencing (WES), a homozygous frameshift variant, c.2108dupC; p.Pro704Thrfs, was identified in the *DIAPH1* gene. In keeping with this *DIAPH1* gene variant being the underlying cause of disease, a small number of individuals with neurodevelopmental deficits, primarily characterised by the presence of microcephaly, short stature, seizures, intellectual disability and visual impairment have been previously identified with biallelic, loss-of-function mutations in *DIAPH1*[10–14]. Based on this, it has been suggested to categorize these patients with SCBMS or Seizures, Cortical Blindness and Microcephaly Syndrome[13]. Interestingly, three affected individuals with SCBMS developed B cell lymphoma suggesting that, in a manner similar to loss of WASP, mutations in *DIAPH1* may also increase the risk of developing lymphoid tumours[13].

To establish whether pathogenic *DIAPH1* gene variants represent a significant cause of neurodevelopmental disease, we used WES on a cohort of patients exhibiting a range of neurological and developmental deficits overlapping with SCBMS. From this analysis, we identified a cohort of >30 patients with biallelic variants in *DIAPH1* exhibiting clinical symptoms consistent with those exhibited by patient P1 i.e., microcephaly, intellectual disability, impaired vision and seizures. Notably, the core clinical phenotypes ascribed to SCBMS were not universally observed in all patients within this large cohort indicating some phenotypic variability. Approximately a third of affected individuals developed recurrent infections indicating that whilst clinical immunodeficiency is commonly associated with DIAPH1 dysfunction, it is not ubiquitously observed in all patients. In addition to recurrent infections, several patients were noted to have developed hypothyroidism and/or lymphopenia, which had not been previously reported in other affected individuals. In contrast, hearing impairment and thrombocytopenia, which have been previously linked with dominant activating mutations in *DIAPH1* were not observed. Lastly, one patient within this cohort developed Hodgkin lymphoma and another developed a large B cell lymphoma. A detailed clinical description of all the patients with biallelic variants in *DIAPH1* will be reported elsewhere. However, a brief clinical description of the

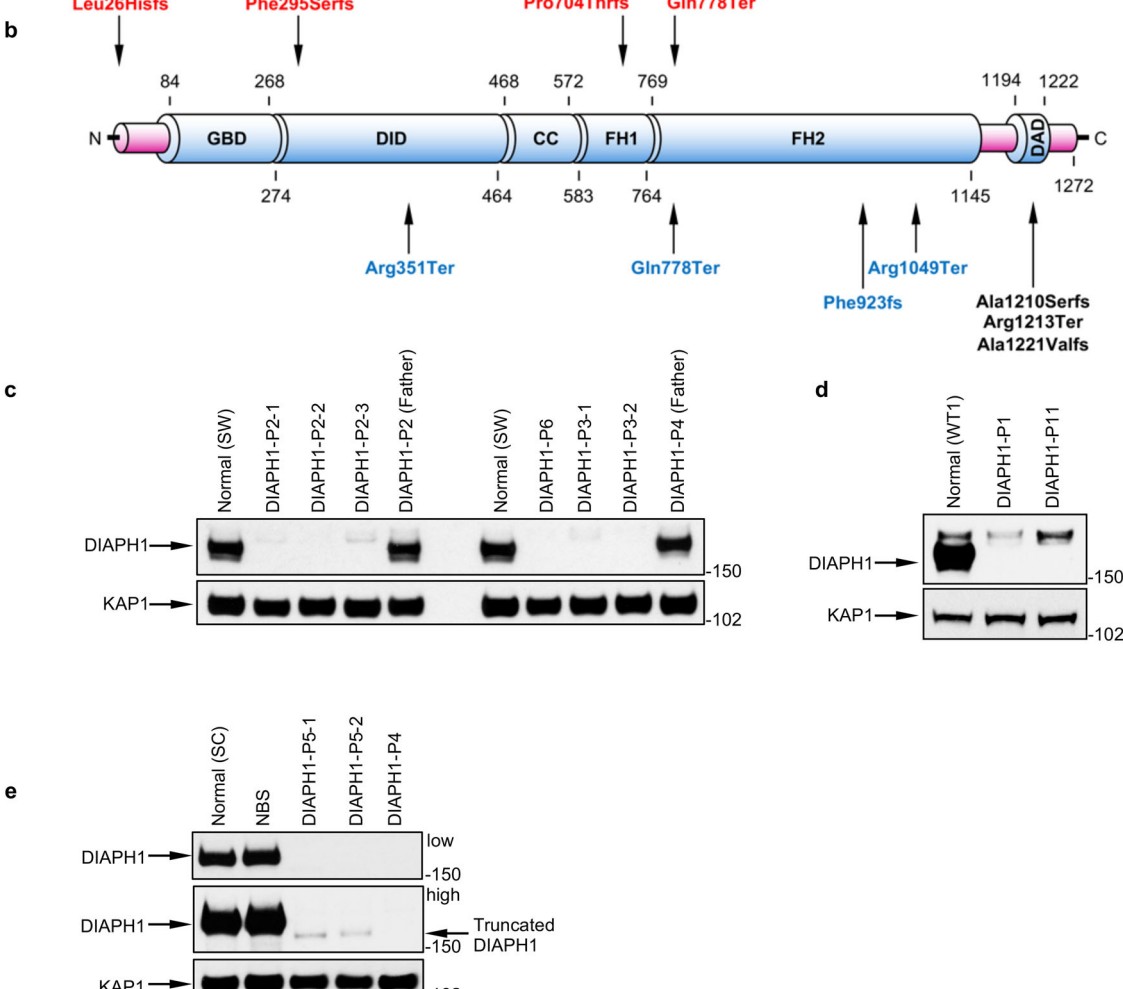

**a**

| Individual | Ancestry | DIAPH1 mutation 1 | DIAPH1 mutation 2 | Microcephaly | Seizures | Recurrent respiratory infections | Vision |
|---|---|---|---|---|---|---|---|
| P1 | Pakistan | c.2108dupC; p.Pro704Thrfs | c.2108dupC; p.Pro704Thrfs | Yes | Yes | Yes | Impaired |
| P2-1 | Egypt | c.2108dupC; p.Pro704Thrfs | c.2108dupC; p.Pro704Thrfs | Yes | Yes | No | Impaired |
| P2-2 | Egypt | c.2108dupC; p.Pro704Thrfs | c.2108dupC; p.Pro704Thrfs | Yes | Yes | No | Impaired |
| P2-3 | Egypt | c.2108dupC; p.Pro704Thrfs | c.2108dupC; p.Pro704Thrfs | Yes | Yes | No | Impaired |
| P3-1 | Egypt | c.75insA; p.Leu26Hisfs | c.75insA; p.Leu26Hisfs | Yes | Yes | No | Mildly impaired |
| P3-2 | Egypt | c.75insA; p.Leu26Hisfs | c.75insA; p.Leu26Hisfs | Yes | Yes | No | Mildly impaired |
| P4 | Pakistan | c.2582-3C>A | c.2582-3C>A | Yes | Yes | No | Blind |
| P5-1 | Pakistan | c.3569_3574+2 | c.3569_3574+2 | Yes | Yes | Yes | Blind |
| P5-2 | Pakistan | c.3569_3574+2 | c.3569_3574+2 | Yes | Yes | Yes | Blind |
| P6 | Pakistan | c.884del; p.Phe295Serfs | c.884del; p.Phe295Serfs | Yes | Yes | Yes | Impaired |
| P11 | Saudi Arabia | c.2332C>T; Gln778Ter | c.2332C>T; Gln778Ter | Yes | Yes | No | Blind |

patients with biallelic *DIAPH1* variants that have been functionally characterised within this manuscript is shown in Fig. 1a, b.

Interestingly, the majority (>75%) of *DIAPH1* gene variants we identified by WES were homozygous frameshift or nonsense mutations suggesting that this disease predominantly arises as a consequence of a complete loss of DIAPH1 protein. Consistent with this premise,

Western blot analysis on extracts from 11 patient-derived cell lines that were available showed a complete loss of full length DIAPH1 expression (Fig. 1c–e). Based on this and the clinical phenotypic variation observed within our cohort, we propose to rename the disorder associated with DIAPH1 deficiency, DIAL (<u>DIA</u>PH1 <u>L</u>oss-of-function) syndrome rather than SCBMS to prevent misdiagnosis based on the

**Fig. 1 | Loss of DIAPH1 is associated with microcephaly, seizures, recurrent respiratory infections and visual impairment. a** Table showing a subset of *DIAPH1* gene variants identified within our cohort of patients and whether they exhibit the primary clinical features of DIAL syndrome: microcephaly, seizures, recurrent respiratory infections and visual impairment. **b** Diagrammatic representation of the DIAPH1 protein highlighting individual domains. Recessive mutations in DIAPH1 identified within our cohort of patients for which cell lines were available are shown in red. Previously reported autosomal recessive variants in DIAPH1 associated with SCBMS are shown in blue. Dominant mutations in DIAPH1 associated with hearing loss and/or thrombocytopenia are shown in black. FH1-2

Formin Homology domains 1-2, GBD GTPase binding domain, CC coiled-coil, DID DIAPH1 inhibitory domain, DAD DIAPH1 activation domain. **c–e** Western blots on cell extracts derived from DIAL syndrome patients indicating the level of DIAPH1 protein expression. **c, d** (Fibroblasts) WT1 and SW are fibroblasts from unrelated healthy controls. **e** (Lymphoblastoid cell lines - LCLs) SC and NBS are LCLs from a healthy control and a patient with Nijmegen Breakage Syndrome (NBS) carrying a homozygous c.657del5 mutation in *NBN,* respectively. KAP1 was used as a protein loading control. **c–e** $n = 1$ independent experiments. Source data are provided as a Source Data file.

lack of certain key clinical features of the disease suggested by the designation SCBMS.

## DIAPH1 promotes repair of DNA double strand breaks

Given the phenotypic similarity between the clinical features exhibited by DIAL syndrome patients with those typically associated with the DSBR deficiency disorders, NBS and WABS, we hypothesised that DIAPH1 may also function to repair DNA DSBs. To test this hypothesis, we initially exposed WT *DIAPH1* fibroblasts and fibroblasts derived from patient P1, which completely lack DIAPH1 protein, to 3 Gy of IR and examined the kinetics of H2AX phosphorylation (γ-H2AX), as a marker of DNA damage repair, over a 24 h time course using Western blotting. In support of our hypothesis, the patient-derived fibroblasts retained γ-H2AX at the 24 h time point when compared to the WT DIAPH1 fibroblast, which is indicative of a DSBR defect (Fig. 2a). Reassuringly, this phenotype was recapitulated when DIAPH1 was transiently knocked down with siRNA in HeLa cells (Supplementary Fig. 1a). Importantly however, recovery of IR-induced γ-H2AX foci was restored in patient P1 fibroblasts when complemented with exogenous WT *DIAPH1* (Fig. 2b and Supplementary Fig. 2a).

To further investigate the possibility that DIAPH1 is required to repair DNA DSBs, we utilised immunofluorescence microscopy to assess how DIAL syndrome cells repair IR-induced DNA breaks using γ-H2AX foci as a marker of damaged DNA. Consistent with our findings using Western blotting, patient P1 fibroblasts exhibited elevated levels of γ-H2AX foci per cell at 24 h post-irradiation when compared to WT fibroblasts (Supplementary Fig. 1b). To expand upon this analysis, we monitored the resolution of γ-H2AX and 53BP1 foci in complemented patient P1-derived fibroblasts following exposure to three different genotoxins that induce DSBs: IR, a topoisomerase 1 inhibitor, CPT or a topoisomerase 2 inhibitor, ETOP. In keeping with a role for DIAPH1 facilitating DSBR, patient P1-derived fibroblasts complemented with the empty vector displayed increased levels of residual γ-H2AX foci at the 24 h timepoint following exposure to IR, CPT and ETOP compared to its isogenic counterpart re-expressing WT DIAPH1 (Fig. 2c, d and Supplementary Fig. 2b–e). Crucially, we could recapitulate this phenotype by knocking down DIAPH1 with siRNA in HeLa cells (Supplementary Fig. 3a–c). To verify that the DSBR defect was not specific to cells from DIAL syndrome patient P1, we monitored the resolution of IR-induced DSBs by immunofluorescence in four additional patient-derived fibroblasts all devoid of detectable DIAPH1 expression and used a fibroblast cell line derived from a patient with NBS for comparison. In line with our previous data, all four additional DIAL syndrome patient-derived cell lines exhibited defective resolution of IR-induced DSBs when compared to a fibroblast cell line with WT *DIAPH1*, albeit not as severe as the NBS1 deficient fibroblast cell line (Supplementary Fig. 4a, b).

To assess the impact that loss of DIAPH1-dependent DSBR had on genome stability, we used immunofluorescence and metaphase spread analysis to quantify micronuclei formation and chromosome breakage as markers of genome instability in cell lines devoid of DIAPH1 following exposure to IR, CPT and ETOP. Consistent with a role for DIAPH1-dependent repair being essential for maintaining genomic integrity, five different DIAL syndrome patient-derived cell lines all

displayed higher levels of micronuclei following exposure to IR, that were intermediate between the WT and NBS cell lines (Supplementary Fig. 4c, d). Again this phenotype could be recapitulated in HeLa cells following DIAPH1 knockdown with siRNA (Supplementary Fig. 3d, e) and could be complemented in the DIAPH1 deficient patient-derived fibroblasts by re-expressing exogenous WT *DIAPH1* (Fig. 2e, f). Moreover, similar results were obtained following exposure to either CPT or ETOP indicating that this phenotype was not specific to IR-induced DSBs (Fig. 2e, f and Supplementary Fig. 3d, e).

Mirroring our observations using micronuclei as a marker of genome instability, both HeLa cells depleted of DIAPH1 and a DIAL syndrome patient-derived lymphoblastoid cell line (LCL) displayed elevated levels of unrepaired chromosome breaks following exposure to either IR, CPT or ETOP (Fig. 2g and Supplementary Fig. 4e). Additionally, cells depleted of DIAPH1 exhibited an increased sensitivity to IR that could not be explained by any obvious alterations in their cell cycle profile (Supplementary Fig. 5a, b). Taken together, our observations clearly indicate that DIAPH1 plays an important role in promoting the repair of DSBs induced by either IR or topoisomerase 1/2 poisons.

Since the MRN complex is known to be involved in activating the ATM-dependent DNA damage response (DDR)[31], we hypothesized that the clinical and cellular similarities between DIAL syndrome and NBS may arise as a consequence of defective DNA damage-induced ATM activation. To examine this, we exposed a WT LCL and LCLs derived from an NBS, Ataxia-Telangiectasia (A-T) and DIAL syndrome patient to IR and used Western blotting coupled with phospho-specific antibodies to known ATM substrates to assess ATM activation in response to DNA damage. However, unlike the NBS and A-T cell lines, the LCL lacking DIAPH1 did not display any observable defects in the activation of ATM in response to IR (Supplementary Fig. 5c). This indicates that the DSBR defect arising as a consequence of DIAPH1 loss is not caused by an inability to activate ATM.

## DIAPH1 is required for the repair of DNA DSBs by HR

Whilst, it has been previously demonstrated that nuclear actin is important for regulating both NHEJ- and HR-dependent repair of DSBs[32,33], much of the work carried out within this area has used chemical inhibitors that disrupt actin polymerisation. As such it is hard to identify how loss of individual factors that regulate actin nucleation affect the different DSBR pathways. More recently, it was shown that the WASP-Arp2/3 complex is critical for mobilising and clustering DSBs destined for repair by HR[27,28]. However, this process was shown to be independent of diaphanous, the Drosophila homolog of DIAPH1[27]. Since this observation conflicted with our own observations, we monitored the resolution of γ-H2AX foci in DIAL syndrome cells irradiated in either G1- and G2-phase of the cell cycle to determine whether loss of DIAPH1 affected NHEJ- and/or HR-dependent DSBR. Strikingly, P1 fibroblasts complemented with an empty vector displayed elevated levels of γ-H2AX foci at 8 h post-irradiation in G2- but not G1-phase cells compared to its WT DIAPH1 complemented counterpart indicating that the loss of DIAPH1 specifically imparts a defect in HR-dependent DSBR (Fig. 3a). To verify this finding, we monitored sister chromatid exchange (SCE) as a measure of productive HR in cells with/

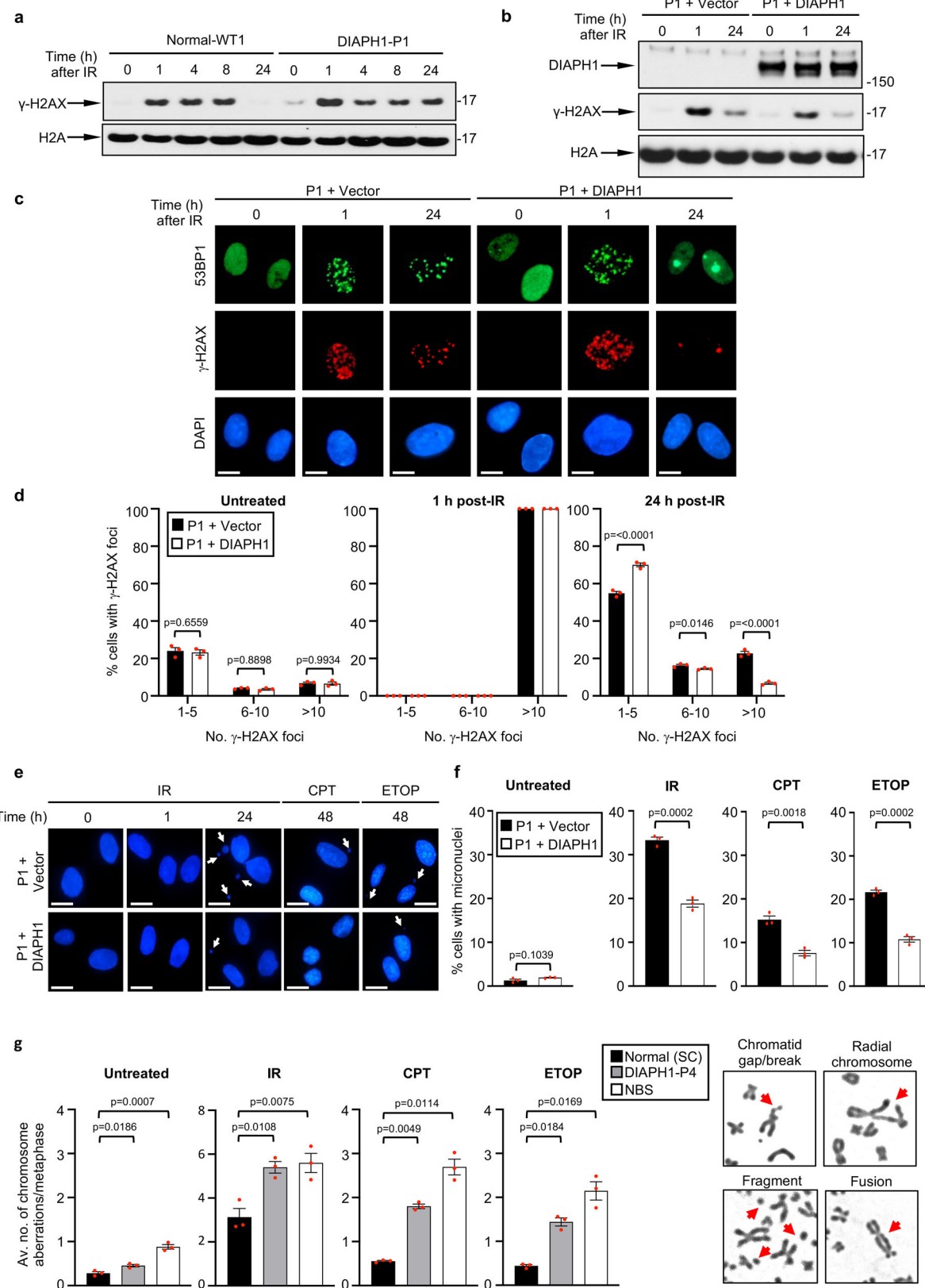

without DIAPH1 following treatment with either CPT or ETOP, which predominantly induce DSBs in S-phase of the cell cycle[34]. A cell line derived from a patient with NBS was used as a positive control for an HR deficiency. Consistent with a role for DIAPH1 regulating HR-dependent repair, the DIAL syndrome cell line displayed a significant reduction in CPT- and ETOP-induced SCEs comparable to that

observed in the NBS cell line (Fig. 3b). Again, we observed a similar phenotype when we depleted DIAPH1 in HeLa cells using siRNA (Supplementary Fig. 5d). To investigate HR in the context of a DIAPH1 deficiency more directly, we utilised a CRISPR-based assay that relies on HR to insert an in-frame mClover tag into the *LMNA* gene following the induction of a DSB by Cas9[35]. In line with our observations using

**Fig. 2 | Cells lacking DIAPH1 display an inability to repair DSBs. a** WT (WT1) and DIAL syndrome fibroblasts (DIAPH1-P1) were mock treated or irradiated with 3 Gy of IR and harvested for Western blotting at time points post-irradiation. Cell extracts were separated by SDS-PAGE and Western blotting was carried out with the antibodies indicated. Representative of $n = 2$ independent experiments. **b** DIAPH1-P1 fibroblasts complemented with either an empty vector (P1 + vector) or HA-tagged WT DIAPH1 (P1 + DIAPH1) were treated as in (**a**). Representative of $n = 2$ independent experiments. **c, d** Quantification of γ-H2AX foci in complemented DIAPH1-P1 fibroblasts before and after exposure to 3 Gy IR. γ-H2AX foci were visualised by immunofluorescence microscopy and quantified in a minimum of 500 cells were counted per time point, per experiment. The mean of $n = 3$ independent experiments is shown with the SEM. Representative images of the cells at different time points before and post-irradiation are shown (**c**). The scale bars represent 10 μm. **e, f** Micronuclei were quantified from cells described in (**c, d**) before and after exposure to 3 Gy IR (24 h), 100 nM CPT (1 h exposure followed by 48 h recovery) and 10 μM ETOP (30 min exposure followed by 48 h recovery). The mean of $n = 3$ independent experiments is shown with the SEM. A minimum of 500 cells were counted per time point, per experiment. Representative images of the cells at different time points before and after treatment are shown (**e**). The scale bars represent 10 μm. **g** Quantification of chromosome aberrations in a WT lymphoblastoid cell line (LCL) (Normal - SC), DIAL syndrome LCL (DIAPH1-P5) and NBS LCL (NBS) before and 24 h after 1 Gy of IR, 0.5 nM CPT or 10 nM ETOP. Chromosome aberrations includes chromatid/chromosome gaps/breaks, chromatid/chromosome fragments and chromosome radials/exchanges. Representative images of each type of aberration are shown. The mean of $n = 3$ independent experiments is shown with the SEM. A minimum of 50 metaphases were counted per cell line in each experiment. Statistical significance was calculated using: (**d**) a two-way ANOVA with multiple comparisons ([untreated] $p = 0.278$, [24 h post-IR] $p = 0.0116$), (**f**) an unpaired Student's $t$-test (two-sided, equal variance) and (**g**) an ordinary one-way ANOVA with multiple comparisons. Source data are provided as a Source Data file.

SCE quantification, depletion of DIAPH1 gave rise to a two-fold reduction in HR comparable to the depletion of either BRCA2 or NBS1 (Fig. 3c and Supplementary Fig. 5e). Notably, using the DR-GFP reporter assay, only a mild reduction in HR was observed when DIAPH1 was depleted (Supplementary Fig. 5f). This indicates that there are differential requirements for the repair of DSBs induced in artificial reporter systems versus endogenous loci.

To gain insight into where DIAPH1 is functioning within the HR-dependent DSBR pathway, we investigated whether cells lacking DIAPH1 exhibited any defects in the recruitment of critical factors that are known to promote HR at sites of DNA damage. To this end, we quantified the ability of patient P1 fibroblasts complemented with either an empty vector or WT *DIAPH1* to form BRCA1, RPA and RAD51 foci following exposure to CPT using immunofluorescence microscopy. We opted to use CPT since DNA DSBs induced by CPT are solely repaired by HR, whereas in contrast DSBs induced by IR or ETOP can be repaired by either NHEJ or HR, which would complicate our analysis. Furthermore, to ensure our analysis was not skewed by any alterations in cell cycle distribution, BRCA1, RPA2 and RAD51 foci quantification was only carried out in S/G2-phase cells marked by CENPF/mitosin expression. Strikingly, cells lacking DIAPH1 exhibited a significant reduction in the number of BRCA1, RPA2 and RAD51 foci induced by CPT (Fig. 3d–f and Supplementary Fig. 6a–c), which suggests that like WASP and the Arp2/3 complex, DIAPH1 also regulates HR at an early stage during the repair process.

## DIAPH1 regulates DNA DSB end-resection by facilitating the relocalisation of the MRN complex to DSBs

A critical decision point that determines whether a DSB is going to be repaired by NHEJ or HR is resection of the DSB end initiated by the MRN nuclease complex in conjunction with CtIP[36]. It has been proposed that the role of nuclear actin in facilitating DNA end-resection is mediated in part by MRE11 recruiting the Arp2/3 complex to DSBs, which triggers localised actin filament formation that promotes mobilisation and subsequent clustering of the DSBs to the nuclear periphery[27,28]. It was suggested that this DSB clustering somehow enhances HR-dependent repair possibly via a feedback mechanism that potentiates further end-resection. However, precise details of this feedback mechanism are unclear.

To assess whether loss of DIAPH1 affects DSB end-resection, we used super-resolution stochastic optical resolution reconstruction microscopy (STORM) to monitor the recruitment of MRE11 to CPT-induced DSBs at sites of ongoing DNA replication. Strikingly, we observed that loss of DIAPH1 compromised the ability of MRE11 to be relocalised to both damaged replication forks (Fig. 4a) and the surrounding chromatin (Supplementary Fig. 6d). This indicates that DIAPH1 functions upstream of the MRN complex during DSB end-resection. Consistent with this, cells lacking DIAPH1 also failed to efficiently localise BRCA1 and RPA2 to damaged replication forks

(Fig. 4b) and the surrounding chromatin (Supplementary Fig. 6d). To further substantiate a role for DIAPH1 in targeting the MRN complex to DSBs, we used the ER-mCherry-LacI-FokI-DD nuclease system, which induces DSBs specifically within an integrated Lac-operator array[37], to assess the relocalisation of MRE11 to a defined DSB in the absence of DIAPH1. Consistent with our observations using STORM, cells depleted of DIAPH1 exhibited a significant defect in the recruitment of MRE11 to FokI-induced DSBs (Fig. 4c and Supplementary Fig. 7a).

To directly demonstrate that DIAPH1 regulates DSB end-resection, we utilised *Asi*SI-ER U-2-OS cells coupled with qPCR to examine the amount of single-stranded DNA (ssDNA) generated by resection at two specific loci 335 bp and 1618 bp away from the *Asi*SI-induced DSB[38] (Supplementary Fig. 7b). Reassuringly, using this assay we observed that depletion of DIAPH1 led to a significant reduction in the amount of ssDNA present at resected DSBs (Fig. 4d). Lastly, we quantified native BrdU foci in cells with/without DIAPH1 following exposure to IR as an alternative approach for measuring DSB end-resection[38]. The MRE11 inhibitor, mirin, was added as a control to block MRE11-dependent DSB end-resection. Consistent with our observations using the *Asi*SI-ER U-2-OS reporter, cells depleted of DIAPH1 exhibited a substantial reduction in the amount of IR-induced BrdU foci, which was comparable to irradiated WT cells treated with mirin (Fig. 4e and Supplementary Fig. 7c). Taken together, our data strongly indicates that DIAPH1 functions to promote HR via facilitating MRE11-dependent DSB end-resection.

One obvious question that arises from our observations is whether DIAPH1 is directly regulating the MRN complex at sites of DNA damage or whether this is mediated via another factor. DIAPH1 is known to be predominantly cytoplasmic[39], which would support the idea that DIAPH1 is functioning to regulate HR indirectly. However, it has been previously shown that a small fraction of DIAPH1 resides within the nucleus[40] indicating that DIAPH1 could be acting to potentiate DSB end-resection directly via an interaction with one or more known DNA repair proteins. Consistent with previous reports relating to the cellular distribution of DIAPH1, we also observed high levels of DIAPH1 in the cytoplasm and low but detectable levels in nucleoplasm and on chromatin (Supplementary Fig. 8a). To investigate whether DIAPH1 relocates to sites of DNA DSBs, we utilised the ER-mCherry-LacI-FokI-DD nuclease system. Strikingly, endogenous DIAPH1 relocalised to approximately 50% of MRE11 positive FokI-induced DSBs (Fig. 5a). In keeping with this observation, we observed a robust increase in the proximity ligation signal between DIAPH1 and γ-H2AX in EdU positive cells following exposure to CPT that was absent in both non-S-phase cells and cells lacking DIAPH1 (Fig. 5b). This indicates that DIAPH1 is actively recruited to DSBs.

Given the presence of DIAPH1 in the nucleus, we over-expressed GFP-DIAPH1 in HEK293FT cells and used GFP-Trap to affinity purify DIAPH1 containing protein complexes from whole cell extracts to investigate whether it might bind proteins involved in the HR pathway.

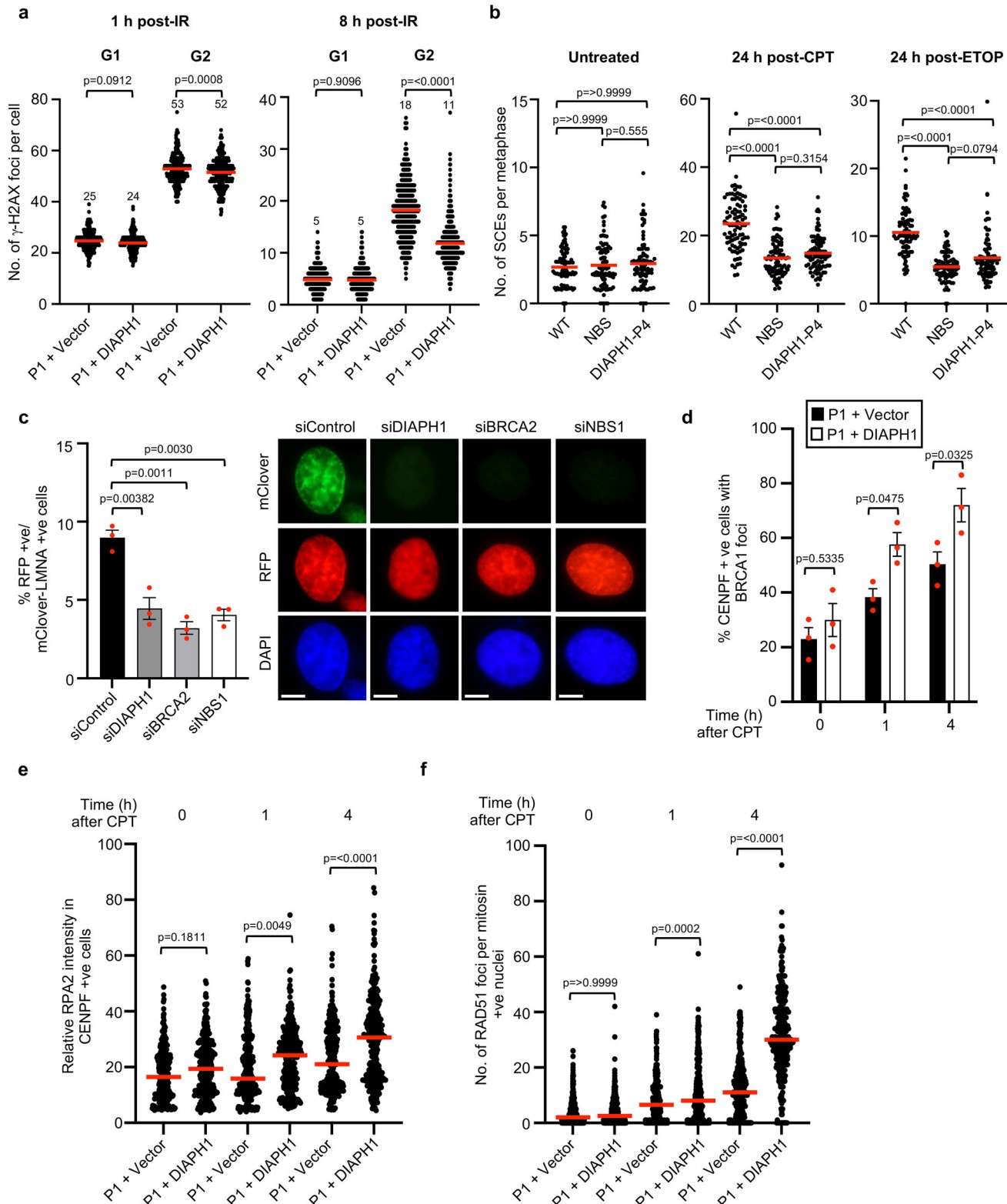

Notably, despite the fact that the majority of the GFP-tagged DIAPH1 purified from whole cell extracts most likely originates from the cytoplasm, we still detected a significant interaction between GFP-DIAPH1 and the HR factors, NBS1, MRE11 and RAD51 but not RPA2 (Fig. 5c). Based on this, it is reasonable to surmise that the DNA repair function of DIAPH1 is mediated in part by an interaction with the MRN complex in the nucleus. To investigate the relationship between DIAPH1 and the MRN complex within the DDR further, we depleted or

inhibited ATM, MDC1 or MRE11 in cells expressing the FokI nuclease and assessed whether this had any impact on DIAPH1 recruitment to DSBs. Unexpectedly, inhibition of ATM or loss/inhibition of MRE11 significantly reduced the relocalisation of DIAPH1 to FokI-induced DSBs (Fig. 5d and Supplementary Fig. 8b). In contrast, depletion of MDC1, which is required to relocalise the MRN to damaged chromatin had very little effect on DIAPH1 recruitment to DSBs (Fig. 5d and Supplementary Fig. 8b). This observation suggests

**Fig. 3 | Cells lacking DIAPH1 exhibit an inability to repair DNA DSBs by HR.**
**a** Complemented DIAPH1-P1 fibroblasts were irradiated with 1 Gy of IR and then fixed/permeabilised at the time points indicated. γ-H2AX foci were quantified in G1-phase (mitosin negative) and G2-phase (mitosin positive) cells at 1 h and 8 h post-irradiation. The mean number of foci per cell of $n = 3$ independent experiments is shown (red line). Foci in a minimum of 100 cells were counted per time point, per experiment. **b** Quantification SCEs in WT, DIAL syndrome and NBS LCLs by CPT (0.5 nM) or ETOP (10 nM). The mean of $n = 3$ independent experiments is shown (red line). Approximately 30 metaphases for each cell line were scored per experiment. **c** siRNA transfected HeLa cells were transfected with the plasmids required to initiate HR at a Cas9-induced DSB located within the *LMNA* locus. HR (% RFP positive cells with mClover-LMNA staining) was quantified using immuno-fluorescence. The mean of $n = 3$ independent experiments is shown with the SEM. A minimum of 500 cells were counted per time point, per experiment. Representative images of transfected cells are shown. The scale bars represent 10 μm.
**d–f** Complemented DIAPH1-P1 fibroblasts were treated with 100 nM CPT for 1 h and then fixed/permeabilised 1 h and 4 h post-treatment. Cells were stained with antibodies to (**d**) BRCA1, (**e**) RPA2 and (**f**) RAD51 in conjunction with CENPF/mitosin as a marker of S/G2 cells. BRCA1 and RAD51 foci and RPA2 fluorescence intensity were quantified in a minimum of 100 S/G2 cells per timepoint, per experiment. The mean of $n = 3$ independent experiments with the SEM is shown (red line). Statistical significance was calculated using: (**a**) an ordinary one-way ANOVA with multiple comparisons, (**b**) a Kruskal–Wallis test ([untreated] $p = 0.7703$, [24 h post-CPT] $p = <0.0001$, [24 h post-ETOP] $p = <0.0001$), (**c**) one-way ANOVA with multiple comparisons, (**d**) a two-way ANOVA with multiple comparisons and (**e**, **f**) a Kruskal–Wallis test ([**e**] $p = <0.0001$, [**f**] $p = <0.0001$). Source data are provided as a Source Data file.

that detection of DSBs by the MRN complex and subsequent activation of ATM stimulates relocalisation of DIAPH1 to sites of DNA damage, which in turn facilitates further recruitment of the MRN complex to the surrounding chromatin.

## DSBR requires the Arp2/3 complex and γ-actin but not β-actin

Although it is clear that nuclear actin plays a role in regulating both DNA DSBR and replication, the majority of this work has been carried out using either chemical inhibitors that block actin nucleation or by over-expression of β-actin fused to a GFP tag coupled to a nuclear localisation sequence[27,28,32,33,40,41]. Consequently, very little work has been done to define which non-muscle paralog of actin is involved in controlling these DNA damage-responsive pathways. β-actin and γ-actin, which only differ by four amino acids (Fig. 6a) are the predominant actin paralogs expressed in most cells. De novo mutations in the *ACTB* and *ACTG1* genes, which encode β-actin and γ-actin respectively, have been identified in patients with BWCFF syndrome, which exhibit clinical similarities with DIAL syndrome e.g., microcephaly, growth retardation, immunodeficiency, intellectual disability and seizures[15–23]. This suggests that DIAPH1 could be facilitating HR-dependent DSBR in conjunction with β-actin and/or γ-actin. To investigate this further, we examined the repair of IR and CPT-induced DNA damage in a panel of BWCFF syndrome cell lines (Supplementary Table 1 and Supplementary Fig. 9) using γ-H2AX/53BP1 foci and micronuclei as markers of DNA DSBs and genome instability, respectively. To our surprise only cell lines from BWCFF syndrome patients with mutations in *ACTG1* but not *ACTB* displayed a defect in repairing DNA damage induced by IR or CPT (Fig. 6b, c and Supplementary Figs. 10, 11a, b and 12). To validate these findings, we repeated these assays using isogenic A375 melanoma cells lacking either β- or γ-actin[42]. In line with our observations using BWCFF syndrome cell lines, only the γ-actin knockout cells exhibited an inability to repair DNA DSBs induced by IR, CPT and ETOP as judged by quantifying γ-H2AX/53BP1 foci, micronuclei and chromosome breakage (Supplementary Figs. 13 and 14). These data indicate that β-actin and γ-actin have distinct functions with respect to the repair of DNA DSBs. Consistent with this, using PLA we observed DIAPH1-dependent localisation of γ-actin but not β-actin at sites of CPT-induced DSBs specifically in replicating cells (Supplementary Fig. 15).

To strengthen the functional link between DIAPH1 and actin in regulating DSBR, we complemented patient P1 fibroblasts with a mutant DIAPH1 (I862A) that cannot nucleate actin[7,43] and assessed their capacity to repair CPT-induced DSBs. Reassuringly, in contrast to WT DIAPH1, the I862A mutant DIAPH1 failed to complement the DSBR defect caused by a total loss of DIAPH1 (Fig. 6d, e and Supplementary Fig. 16). This observation highlights the importance of the actin nucleation function of DIAPH1 in mediating its capacity to promote DNA repair. Interestingly, cells expressing the I862A mutant of DIAPH1 displayed a worse DSBR defect than cells complemented with an empty vector, indicating that the presence of a mutant DIAPH1 unable to nucleate actin is more detrimental to the repair process than not having DIAPH1 at all.

Although several studies have implicated the Arp2/3 actin nucleation complex in regulating DSBR, primarily these studies have relied on either chemical inhibition or depletion using siRNA[27,28]. However, analysis of Arp2/3-dependent DSBR has not been thoroughly examined in the context of naturally occurring mutations associated with human disease. In this respect, mutations in *ARPC1B*, *ARPC4* and *ARPC5*, which encode components of the Arp2/3 complex, have been identified as the cause of several genetic disorders, characterised by either immunodeficiency and thrombocytopenia or microcephaly and neurodevelopmental defects[44–48] (Supplementary Table 1). Utilising cell lines derived from patients with mutations in these three components of the Arp2/3 complex, we exposed them to IR or CPT and assessed the efficiency of DSBR and its impact on genome stability by quantifying γ-H2AX/53BP1 foci and micronuclei. Strikingly, despite the difference in clinical phenotypes, cells with mutations in all three components of the Arp2/3 complex exhibited defective DSBR and increased genome instability following exposure to CPT (Fig. 6f, g and Supplementary Figs. 10, 11c and 12). These findings are consistent with previous reports documenting the role of the Arp2/3 complex in promoting DSBR[27,28] and indicates that irrespective of whether the inherited mutations are de novo or recessive they are still capable of disrupting the repair function of the Arp2/3 complex.

## DIAPH1 functions with the Arp2/3 complex and γ-actin to promote MRE11-dependent HR

Whilst our data suggests that DIAPH1, the Arp2/3 complex and γ-actin are all important for DNA DSBR, it is not clear whether these proteins function together or in separate pathways to mediate the repair of damaged DNA. With this in mind, we used GFP-Trap to affinity purify GFP-tagged DIAPH1 from whole cell extracts and Western blotting to assess binding to β-actin, γ-actin and components of the Arp2/3 complex. Interestingly, GFP-DIAPH1 bound strongly to γ-actin, very weakly to β-actin and not at all to ARP2 nor ARPC4 (Fig. 7a). Thus, despite only differing by four amino acids, DIAPH1 exhibits a striking preference towards binding γ-actin. To investigate a functional link between DIAPH1, the Arp2/3 complex and γ-actin, we depleted DIAPH1, β-actin, γ-actin or ARPC4 and assessed whether loss of these proteins compromised the recruitment of MRE11 to DSBs using the ER-mCherry-LacI-FokI-DD nuclease system. Consistent with our previous findings, depletion of DIAPH1, γ-actin and ARPC4 but not β-actin affected MRE11 relocalisation to DSBs (Fig. 7b and Supplementary Fig. 17a). To establish whether loss of DIAPH1, γ-actin and ARPC4 are epistatic with respect to regulating HR-dependent repair of DSBs, we used siRNA to deplete these proteins alone or in combination and monitored how this affected DSB end-resection and RAD51 recruitment to DSBs following exposure to CPT in the presence or absence of mirin[49]. As expected depletion of DIAPH1 or treatment with mirin significantly reduced IR-induced native BrdU foci and CPT-induced

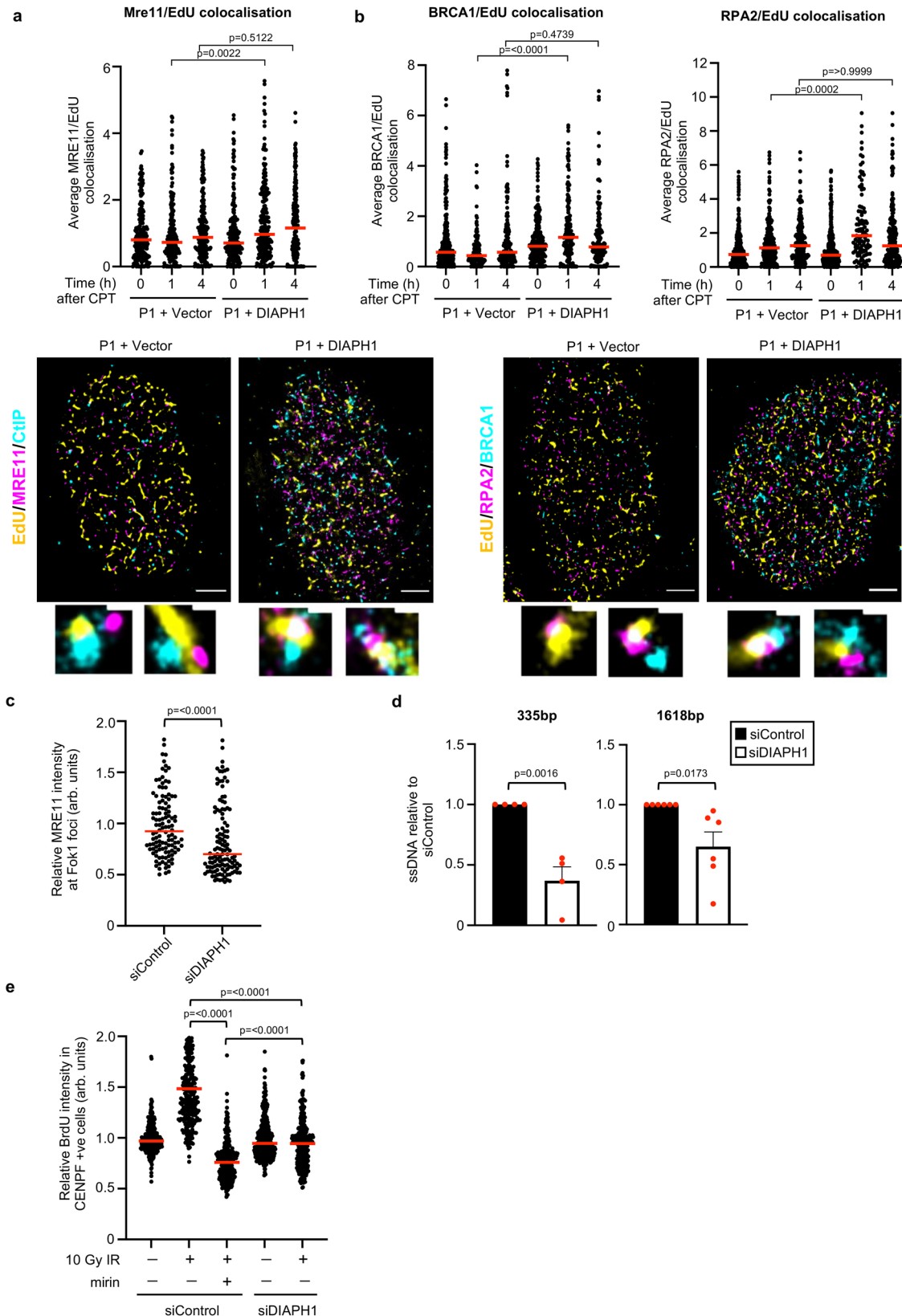

RAD51 foci (Fig. 7c, d and Supplementary Fig. 17b). Interestingly, we observed a similar effect following depletion of γ-actin and ARPC4, which was not further reduced when DIAPH1 was co-depleted (Fig. 7c, d and Supplementary Fig. 17b). Notably however, this phenotype was not recapitulated when β-actin was depleted (Fig. 7c, d and Supplementary Fig. 17b). In keeping with these observations, γ-actin

knockout cells displayed a reduced ability to form RPA2 and RAD51 foci induced following CPT exposure (Supplementary Fig. 18a, b). Moreover, a similar inability to efficiently recruit RAD51 to DNA DSBs was also observed in patient-derived fibroblasts with mutations in *ACTG1, ARPC1B* and *ARPC5* but not *ACTB* (Supplementary Fig. 18c). Lastly, in contrast to WT DIAPH1, expression of a DIAPH1 mutant

**Fig. 4 | DIAPH1 binds to and promotes recruitment of the MRN complex to sites of DSBs. a, b** Complemented DIAPH1-P1 fibroblasts were treated with 100 nM CPT for 1 h, 30 min into treatment cells were exposed to 10 µM EdU for 30 min and then fixed/permeabilised 1 h and 4 h post-treatment. Cells were stained with the antibodies indicated and protein localisation visualised by STORM microscopy. Quantification of the average local density of MRE11, BRCA1 and RPA2 detected around individual EdU molecules within a region of interest (ROI) was carried out and normalised to the untreated controls. A minimum of 100 ROIs were quantified over three independent experiments. The median of $n = 3$ independent experiments is shown (red line). Representative images of the cells treated with CPT are shown. The scale bar represents 1500 nm. **c** U-2 OS-FokI cells were transfected with the indicated siRNAs and then fixed/immunostained with an antibody to MRE11 4 h after FokI induction. The fluorescence intensity of MRE11 per FokI focus was quantified from 100 cells per experiment. The median of $n = 3$ independent experiments is shown (red line). **d** Quantification of resection of an *Asi*S1-induced DSB on chromosome 1 at a position 335 bp and 1618 bp away from the DSB using qPCR in siRNA transfected U-2 OS cells. The mean of $n = 3$ independent experiments is shown with the SEM. **e** siRNA transfected HeLa cells were labelled with 25 µM BrdU for 24 h, treated with 10 Gy IR for 1 h and then permeablised/fixed 4 h post-irradiation. Cells were stained with an anti-BrdU antibody and CENPF (as a marker of S/G2 cells) and the intensity of nuclear BrdU staining was quantified in 100 S/G2 cells per experiment. Cells were treated with 50 µM mirin 1 h prior to irradiation where indicated. The mean of $n = 3$ independent experiments is shown (red line). Statistical significance was calculated using: (**a, b, e**) a Kruskal–Wallis test ([**a**] $p = 0.0034$, [**b**] $p = <0.0001$, [**e**] $p = <0.0001$) and (**c, d**) an unpaired Student's *t*-test (two-sided, equal variance). Source data are provided as a Source Data file.

(I862A) unable to nucleate actin filaments in fibroblasts from DIAL syndrome patient P1 failed to rescue CPT-induced RAD51 foci formation (Fig. 7e; Supplementary Fig. 19a). These findings strongly suggest that DIAPH1 functions together with the Arp2/3 complex and γ-actin to promote HR-dependent repair. To confirm this, we utilised the CRISPR-based mClover-LMNA HR assay to quantify HR in cells depleted of DIAPH1, β-actin, γ-actin and ARPC4. Reassuringly we observed that depletion of DIAPH1, γ-actin and ARPC4 but not β-actin reduced the efficiency of HR-dependent repair and that loss of DIAPH1 in combination with either γ-actin or ARPC4 did not further decrease HR (Fig. 7f and Supplementary Fig. 19b). Taken together our data identify a previously unrecognised DSBR pathway regulated by DIAPH1, the Arp2/3 complex and γ-actin.

## Discussion

Actin is one of the most highly conserved proteins throughout evolution and has been implicated in regulating a multitude of essential processes in the cell, such as motility, division, adhesion and vesicular transport[1,2]. The majority of research aimed at defining how actin controls these various cellular processes has focused on its role in the cytoplasm, where it forms the principal component of the cytoskeleton. However, it has been known for over 40 years that actin also resides in the nucleus, where it has been shown to be involved in regulating transcription, chromatin remodelling, DNA repair and more recently DNA replication[1,2,29,30]. Although actin has fundamental roles in the cell, >20 different inherited human diseases, termed actinopathies, have been linked with abnormal regulation of actin-dependent pathways[50]. Interestingly, despite actin being ubiquitously expressed in all cell types, the majority of these actinopathies predominantly manifest with defects in the haematopoietic system indicating that some cell types depend on actin-dependent pathways more than others[50,51]. Strikingly, mutations have even been identified in the *ACTB* and *ACTG1* genes, which encode β-actin and γ-actin, respectively. However, in contrast to other actinopathies, mutations in the *ACTB* and *ACTG1* genes are associated with a neurodevelopmental disorder affecting multiple organ systems[23].

Given the vast array of cellular processes that actin is involved in regulating, it is difficult to assign the loss of individual actin-dependent pathways to the presentation of specific clinical deficits exhibited by affected patients. Furthermore, since de novo *ACTB* and *ACTG1* mutations can act in either a dominant or haploinsufficient manner and have differential impacts on protein stability and/or filament polymerisation/depolymerisation, this results in extensive clinical heterogeneity. Despite this, some patients with BWCFF exhibit clinical abnormalities that overlap with those typically associated with a DSBR deficiency, such as microcephaly, short stature, intellectual disability, facial dysmorphia, hearing loss, seizures and variable immunodeficiency. This suggests that perhaps a subset of the symptoms associated with actin dysfunction may arise as a consequence of defective repair of DNA damage. Consistent with this, we have identified a cohort of >30 patients with biallelic loss-of-function mutations in the

actin nucleation factor, *DIAPH1*, that exhibit a striking clinical similarity to the DNA repair disorders, NBS and WABS[24–26]. This prompted us to investigate whether DIAPH1 functioned to regulate the actin-dependent DNA DSBR pathway.

Interestingly, whilst other actin nucleating factors, such as WASP and the Arp2/3 complex have been shown to be important for promoting the repair of DNA DSBs, it was suggested that DIAPH1 is not required[27]. Contrary to this, we demonstrate that DIAPH1 is essential for facilitating HR-dependent repair of DNA DSBs induced by IR, CPT and ETOP and that this requires both the Arp2/3 complex and γ-actin but not β-actin. It is currently not clear why HR-dependent repair has a clear specificity towards utilising γ-actin but not β-actin, especially since these two actin paralogs only differ by 4 amino acids. It is possible that this specificity is determined by DIAPH1 and in support of this, our data would indicate that DIAPH1 preferentially binds γ-actin over β-actin in cells. Consistent with our observations, it has been recently shown by the Wilde laboratory, that the linker region of DIAPH1 and DIAPH3 plays an important role in determining which actin paralog they bind[52]. Notably, many of the previous studies identifying a role for nuclear actin in regulating DNA repair have either used chemical inhibitors of actin polymerisation, which would not discriminate between γ-actin and β-actin-containing filaments, or have overexpressed mutants of β-actin[27,28,32]. Given that we have demonstrated that mutation or a complete loss of β-actin does not affect DSBR, the only explanation for this disparity is that over-expression of dominant mutants of β-actin also block the ability of endogenous γ-actin to polymerise. Based on this, some level of caution should be taken when drawing conclusions from phenotypes arising when over-expressing mutant forms of actin.

One of the obvious questions that stems from our observations is whether the presence of a DSBR defect in BWCFF syndrome patients with *ACTG1* mutations is associated with clinical features not displayed by BWCFF syndrome patients with *ACTB* mutations. Unfortunately, given the high degree of clinical heterogeneity seen amongst patients with BWCFF syndrome[15–23], this is difficult to answer without a systematic analysis of DSBR in cell lines derived from all documented *ACTG1* and *ACTB* mutant patients. However, within the patient cohort we analysed, it is interesting to note that the two BWCFF patients with the p.Arg183Trp mutation in β-actin (ACTB-P3 and ACTB-P4) did not exhibit a DSBR defect whereas if the same codon is mutated to glutamine in γ-actin (ACTG1-P3), this did give rise to an inability to repair DNA DSBs. This indicates that even when the mutation is within a region of β-actin and γ-actin that is identical, the impact of the mutation on DNA repair is governed by the nucleation factor and which actin paralog it binds.

Despite the clinical overlap between DIAL syndrome patients and individuals with BWCFF, some differences are apparent. Microcephaly, immunodeficiency, seizures and blindness, which are common to DIAL syndrome patients, are only found in <50% of patients with BCWFF[10–23]. In contrast, hearing loss, cardiac abnormalities, kidney defects and coloboma are more frequently observed in patients with BWCFF but

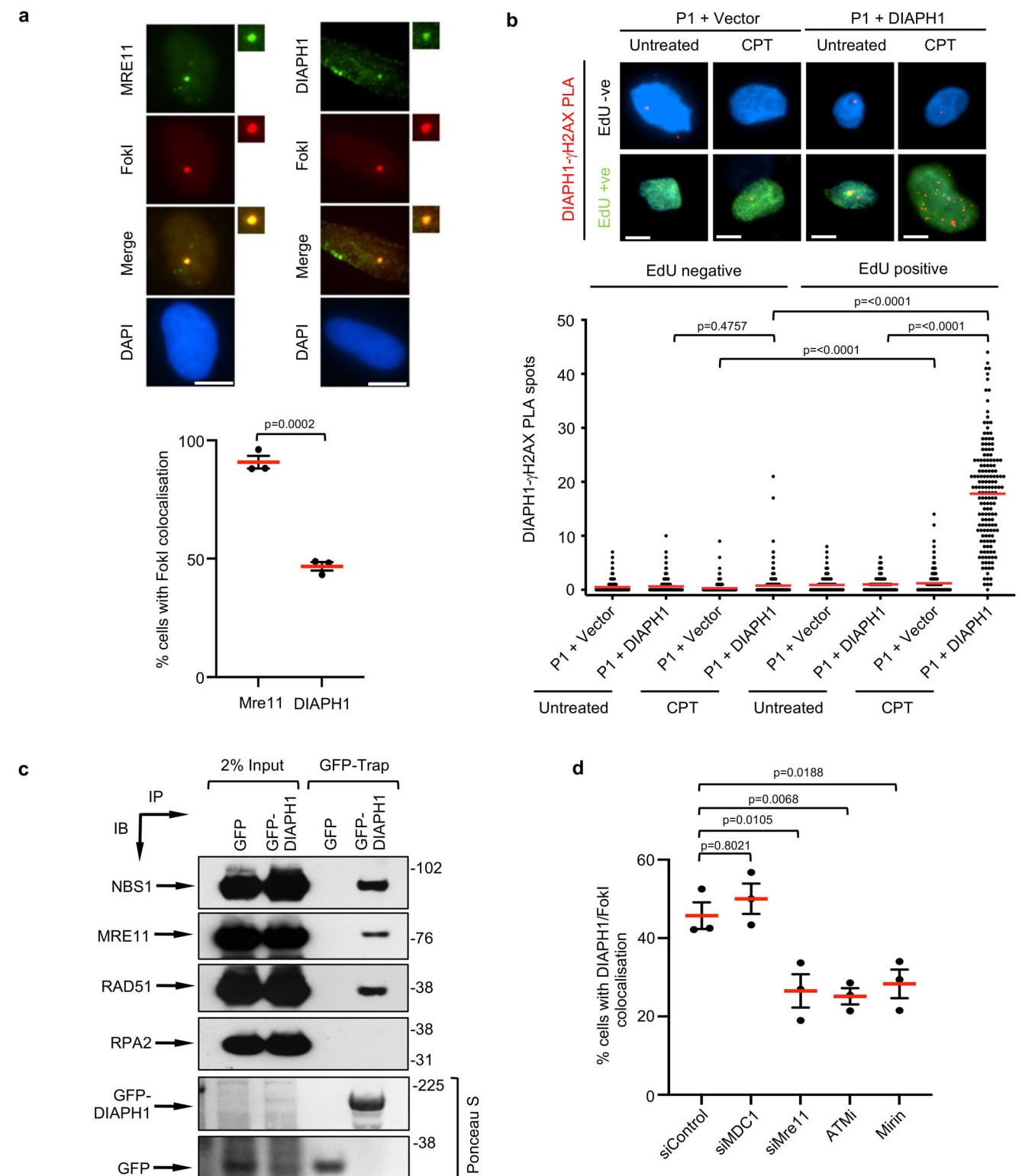

are rarely seen in DIAL syndrome patients[10–23]. Whilst the underlying reason for this phenotypic disparity is not immediately obvious, it is likely that this is heavily influenced by how the inherited gene variants affect protein function and/or stability. In this respect, the majority of identified *DIAPH1* gene variants affect both alleles and give rise to a complete loss of protein. Whereas pathogenic variants in either *ACTB* or *ACTG1* are dominant and can have differential impacts on the ability of cellular actin to form filaments. Moreover, since many actin nucleating factors exist in mammalian cells, loss of DIAPH1 will only

compromise a subset of actin-dependent pathways. In contrast, mutations in *ACTB* or *ACTG1* will negatively affect many more cellular processes than those compromised by a deficiency of a single nucleating factor. Taken together, it is tempting to speculate that the systemic clinical deficits observed in BWCFF patients are caused, in part, by a defect in DSBR affecting multiple different cell types/tissues. Whereas in relation to DIAL syndrome, the clinical symptoms resulting from an underlying DSBR deficiency will be limited to cell types/tissues that rely on DIAPH1 as the predominant actin nucleating factor.

**Fig. 5 | DIAPH1 is recruited to DSBs. a** (Top) Representative images of the recruitment of MRE11 and DIAPH1 to FokI-induced DNA DSBs in U-2 OS cells. The scale bars represent 10 μm. (Bottom) Quantification of the percentage of cells with FokI foci that colocalise with MRE11 or DIAPH1 from $n = 3$ independent experiments. **b** Complemented DIAPH1-P1 fibroblasts were treated with 10 μM EdU for 30 min, exposed to 100 nM CPT for 1 h and then permeabilised/fixed. EdU positive cells were labelled with Alexa-fluor-488 using click chemistry. Cells were then subjected to a PLA reaction using antibodies to DIAPH1 and γH2AX. (Top) Representative images of the PLA reaction (red spots) in EdU positive (green nuclei) and negative (blue) nuclei stained with DAPI. (Bottom) PLA spots were quantified in at least 50 EdU-positive and 50 EdU-negative cells per experiment, per cell line, per condition. The mean number of PLA spots per cells from $n = 3$ independent experiments is shown. **c** GFP or GFP-DIAPH1 was purified from 293FT cell extracts using GFP-Trap and co-purified proteins were subjected to SDS-PAGE and Western blotting with the antibodies indicated. Representative of $n = 2$ independent experiments. **d** Quantification of the percentage of DIAPH1 positive FokI foci in U-2 OS cells depleted of either MDC1 or MRE11 or treated with either 5 μM ATM inhibitor (ATMi) or 50 μM mirin. The mean number of DIAPH1 positive FokI foci per cell from $n = 3$ independent experiments is shown. Statistical significance was calculated using: (**a**) an unpaired Student's $t$-test (two-sided, equal variance), (**b**) a Kruskal–Wallis test ($p = <0.0001$) and (**d**) an ordinary one-way ANOVA. Source data are provided as a source data file.

In addition to γ-actin, our data would suggest that DIAPH1 also functions to promote DSBR in conjunction with the Arp2/3 complex. Whilst the Arp2/3 complex has previously been implicated in the DSBR pathway[27,28], it has not been assessed thoroughly in the context of naturally occurring human mutations. Although hypomorphic mutations in *ARPC1B*, *ARPC4* and *ARPC5* all give rise to a DSBR defect, the clinical symptoms displayed by the affected patients are more consistent with WAS than DIAL or BCWFF syndrome i.e., they predominantly exhibit immunodeficiency and/or haematological abnormalities rather than neurodevelopmental defects (Supplementary Table 1). Whilst the underlying reason for the inconsistency between the clinical phenotypes caused by Arp2/3 complex mutations and loss of DIAPH1 is unclear, it is possible that the Arp2/3 complex only plays a minor role in DIAPH1-dependent DSBR. Conversely, it could be that the Arp2/3 complex in conjunction with the WASP homolog WASH, predominantly functions to promote NHEJ or heterochromatic DSBR, which are likely to be more prominent in non-proliferating lymphocytes[27,53] rather than HR[28].

Although there is substantial evidence that proper regulation of nuclear actin is important for repairing DSBs, it is not understood which repair pathways require nuclear actin polymerisation and which actin nucleators are involved. There are several studies showing that disrupting actin polymerisation can affect NHEJ, HR and SSA-dependent repair of DSBs[27,28,32,33,53]. As such, it is difficult to decipher how a DSBR defect caused by mutations in genes that regulate actin polymerisation/depolymerisation/fragmentation contributes to specific disease-associated symptoms. Despite this, our observations of the clinical and cellular features of DIAL syndrome would indicate that the phenotypic overlap with NBS could be attributed to a role for DIAPH1 in promoting HR-dependent repair of DSBs via binding to and facilitating the localisation of the MRN complex to sites of DNA damage. Since, it was previously reported that the function of actin in promoting HR by relocalising and clustering DSBs to the nuclear periphery is mediated by the MRN complex recruiting the Arp2/3 complex to resected breaks[27,28], this would suggest that DIAPH1 acts upstream of the Arp2/3 complex during the repair process. In contrast, our observations indicating that ATM and the MRN complex are also required to mediate recruitment of DIAPH1 to DSBs would suggest the presence of a feedback mechanism whereby MRN-dependent relocalisation of DIAPH1 to DSBs catalyses the formation of an F-actin-containing structure, which in turn serves to facilitate further recruitment of the MRN complex to the damaged chromatin (Supplementary Fig. 20). However, the nature of the macromolecular complex containing the MRN complex, DIAPH1 and γ-actin remains to be determined.

In addition to the MRN complex, we also detected an interaction between DIAPH1 and RAD51, indicating perhaps that DIAPH1 plays additional roles later on within the repair process potentially to seed and/or stabilise RAD51 filament formation. Based on this, we propose that DIAPH1 binds to the MRN complex and stabilises it at the DSBs, perhaps by stimulating the localised formation of linear γ-actin filaments. This serves to both promote DNA end-resection as well as stimulate the recruitment of the Arp2/3 complex to initiate the formation of branched γ-actin filaments. The formation of these branched actin filaments could then facilitate the mobilisation and clustering of DSBs as well as the relocalisation of other repair factors, such as the SMC5/6 complex to the sites of damaged DNA[27]. In addition, we hypothesize that DIAPH1 and γ-actin could further support HR-dependent DSBR by binding and potentially enhancing and/or stabilising RAD51 filament formation during strand invasion of the undamaged sister chromatid. In relation to this, we observed a greater reduction in the efficiency of DSBR arising from DIAPH1 loss when we assayed HR using CRISPR-based gene editing of the endogenous *LMNA* locus as compared to using the I-SceI-dependent DR-GFP reporter system. This indicates that the chromatin context of the DNA break and/or the type of template used to repair the DSB i.e., an undamaged sister chromatid versus a proximally located, small fragment of the *GFP* gene, may affect whether DIAPH1 and γ-actin are required to facilitate repair the damage. Given the proposed role for the actin-dependent repair in mobilising and clustering DSBs, it is likely that this function would only be minimally required when repairing an I-SceI-induced DSB, which is located in close proximity to an artificial repair template. In contrast, in the context of an DSB induced in an endogenous locus, one could envisage that actin plays a prominent role that not only helps to stabilise the repair machinery, such as the MRN complex or RAD51, at the DNA break but also facilitates chromatin mobilisation to position the sister chromatid close to the DSB to enable strand invasion. However, additional work is required to verify these hypotheses.

Lastly, although our data strongly supports a role for DIAPH1 and γ-actin potentiating HR-dependent repair of DSBs induced by IR, CPT and ETOP, we cannot rule out the possibility that DIAPH1 is also involved in regulating other DNA repair pathways beyond those mediated by HR.

Taken together, we identify DIAL syndrome as a previously undiscovered DSBR deficiency syndrome associated with neurodevelopmental abnormalities and tumour predisposition akin to those observed in patients with NBS. Furthermore, we demonstrate that loss of DIAPH1 results in defective HR-dependent DNA repair, which is consistent with many of the clinical deficits exhibited by affected patients as well as the suggestion that in some cases *DIAPH1* may act as a tumour suppressor. In support of the latter, the *DIAPH1* gene resides within the region of chromosome 5q31, which is commonly lost in myelodysplastic syndrome (MDS)[54] and mice with heterozygous or homozygous loss of mDia1 (the mouse homolog of DIAPH1) develop MDS phenotypes with age[55,56]. Interestingly, patients with Fanconi Anaemia, which is a disorder caused by an inability to repair DNA intra-/inter-strand crosslinks and other replication-associated genotoxic lesions by HR, also have an increased risk of developing MDS and AML[57]. Whilst this emphasizes the importance of HR in protecting against the development of haematological malignancies, it highlights the existence of additional functional links between DIAPH1 and the HR machinery in dealing with other types of DNA damage, which may be important for preventing tumourigenesis. Consistent with this, it was recently reported that loss of DIAPH1 reduced the ability of cells to relocalise RAD51 to damaged replication forks, which compromised replication fork protection in manner similar to mutations in BRCA1/2[29].

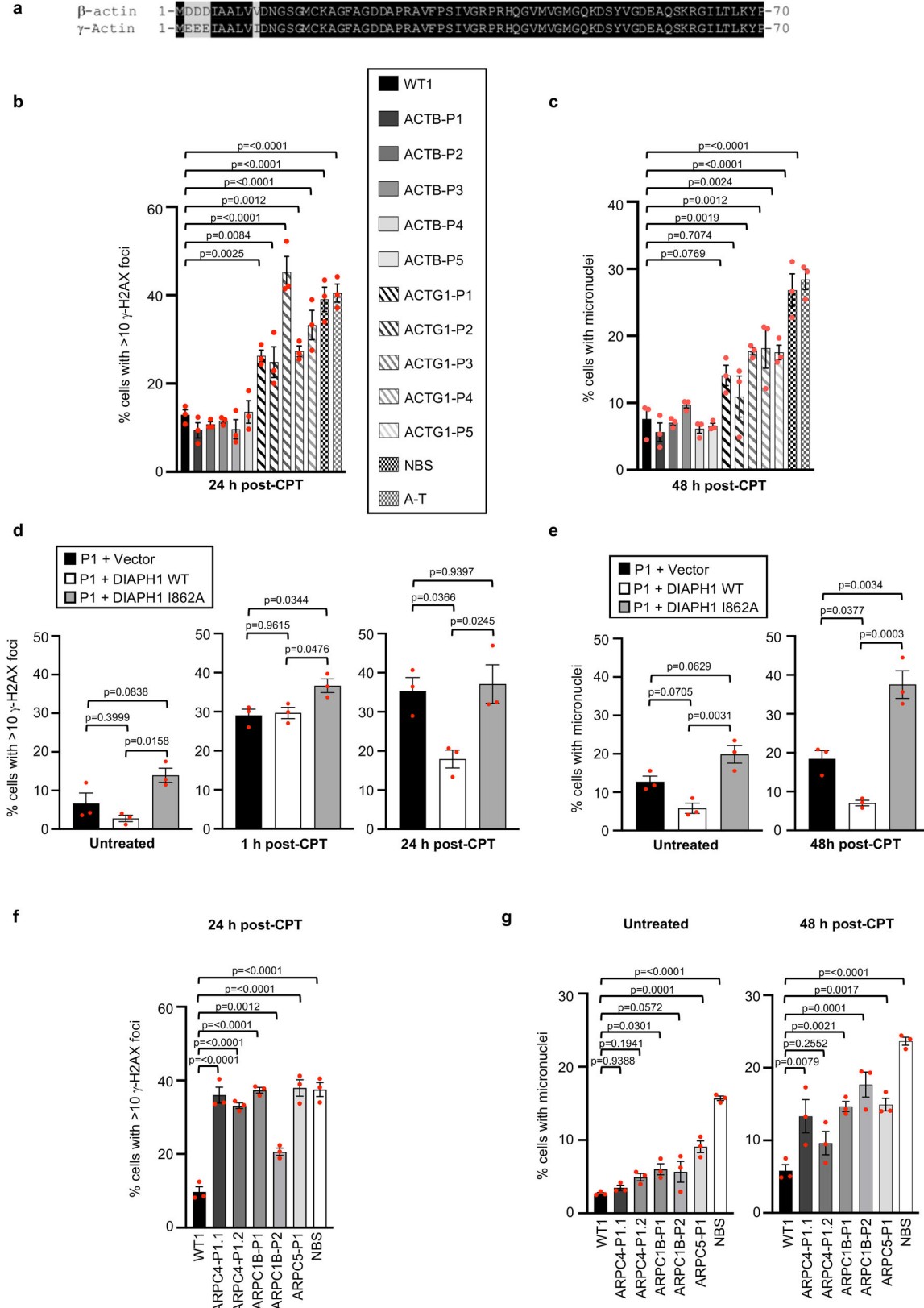

Based on this, one would predict that the HR deficiency in tumours arising as a consequence of DIAPH1 loss could be specifically targeted with therapeutic agents that exhibit synthetic lethality with an HR defect. Unexpectedly however, DIAPH1 depleted cells do not show any increased sensitivity to Olaparib (Supplementary Fig. 21). This indicates that DIAPH1 and BRCA2 have independent roles in regulating

HR-dependent processes at DSBs and damaged replication forks and that the presence of an HR defect does not always equate to a hypersensitivity to PARP inhibitors. This observation serves to highlight our lack of understanding relating to the identity of the deleterious lesion that causes BRCA1/2 deficient cells to be hypersensitive to PARP inhibition. Moreover, this indicates that how a tumour-associated HR

**Fig. 6 | Mutations in *ACTG1* and components of the Arp2/3 complex but not *ACTB* are associated with a DNA DSB repair defect. a** Sequence alignment of the first 70 amino acids of human β- and γ-actin proteins. **b, c** Quantification of γ-H2AX foci (**b**) and micronuclei (**c**) in patient-derived fibroblasts with de novo mutations in *ACTB* or *ACTG1* before and after a 1 h exposure to 100 nM CPT. γ-H2AX foci and micronuclei were visualised by immunofluorescence microscopy and quantified in cells 24 h and 48 h following the removal of the CPT, respectively. The mean of *n* = 3 independent experiments is shown with the SEM. A minimum of 500 cells were counted per time point, per experiment. **d, e** Quantification of γ-H2AX foci (**d**) and micronuclei (**e**) in DIAL syndrome patient P1-derived fibroblasts complemented with either an empty vector, WT DIAPH1 or an actin nucleation defective mutant (I862A) before and after a 1 h exposure to 100 nM CPT. γ-H2AX foci and micronuclei were visualised by immunofluorescence microscopy and quantified in untreated cells and cells 1 h, 24 h and 48 h following the removal of the CPT. The mean of *n* = 3 independent experiments is shown with the SEM. A minimum of 500 cells were counted per time point, per experiment. **f, g** Quantification of γ-H2AX foci (**f**) and micronuclei (**g**) in patient-derived fibroblasts with mutations in *ARPC1B*, *ARPC4* or *ARPC5* before and after exposure to CPT (as in **b, c**). Fibroblasts derived from patients with Nijmegen Breakage Syndrome (NBS) and Ataxia-Telangiectasia (A-T) were used as controls for cells with a DNA DSB repair defect. The mean of *n* = 3 independent experiments is shown with the SEM. A minimum of 500 cells were counted per time point, per experiment. Statistical significance was calculated using: (**b–g**) an ordinary one-way ANOVA ([**b**] *p* = <0.0001, [**c**] *p* = <0.0001, [**f**] *p* = <0.0001, [**g**] *p* = <0.0001). Source data are provided as a Source Data file.

defect determines the response to therapy depends on which HR gene is mutated and the type of genotoxin used for treatment. Despite this, further work is needed to clarify whether *DIAPH1* mutation or loss in tumours creates a genetic vulnerability that can be exploited therapeutically.

## Methods

### Research subject inclusion and ethics

Informed consent was obtained from all participating families to take clinical samples and to publish clinical information in accordance with local approval regulations and in compliance with the Declaration of Helskini principles. This study was approved by the West Midlands, Coventry and Warwickshire Research Ethics Committee (REC: 20/WM/0098) and the International Genetics Collaboration (REC: 22/NE/0080). Due to the rarity of the disorder caused by *DIAPH1* gene mutations, blood and skin samples were obtained from affected individuals where possible. The biological sex of the individuals in our study was not considered as a relevant variable. However, the patient cohort studied contained 8 affected female patients and 2 affected male patients. A collaboration to study the pathological significance of the identified *DIAPH1* variants was established via GeneMatcher[58].

### Cell culture and generation of cell lines

Dermal primary fibroblasts were grown from skin-punch biopsies and maintained in Dulbecco's modified Eagle's medium (DMEM; Thermo Fisher Scientific) supplemented with 20% foetal calf serum (FCS), 5% L-glutamine and 5% penicillin-streptomycin (Merck) antibiotics. Primary fibroblasts were immortalized with a lentivirus expressing human telomerase reverse transcriptase (hTERT) that was generated by transfecting 293FT cells (Thermo Fisher Scientific) with the plasmids: pLV-hTERT-IRES-hygro (Addgene #85140), psPax2 (Addgene #12260) and pMD2.G (Addgene #12259). Selection was performed using Hygromycin (Thermo Fisher Scientific) at 70 µg/ml. All lymphoblastoid cell lines (LCLs) were routinely grown in RPMI-1640 (Thermo Fisher Scientific) supplemented with 10% FCS, 5% L-glutamine and 5% penicillin-streptomycin. 293FT, HeLa and A375 cells were routinely grown in DMEM supplemented with 10% FCS, 5% L-glutamine and 5% penicillin-streptomycin. U-2 OS cells were routinely grown in McCoy's 5 A media supplemented with 10% FCS, 5% L-glutamine and 5% penicillin-streptomycin. Fibroblast complementation was carried out using the pLVX-IRES-Neo lentiviral vector (Takara Bio) encoding 2 × HA-tagged *DIAPH1*. All cell lines were routinely tested for mycoplasma.

### Plasmid generation

The 2 × HA-tagged *DIAPH1* pLVX-IRES-Neo lentiviral vector was generated with the *DIAPH1* open reading frame amplified from cDNA using the primers: DIAPH-LVX-XhoI-Fwd: 5′-GTTGTTCTCGA-GATGGAGCCGCCCGGCGGGAG-3′ and DIAPH-LVX-NotI-Rev: 5′-GTTGTTGCGGCCGCTTAGCTGCACGCCCAACCAACTCCTTG-3′. Generation of the DIAPH1 I862A mutant was achieved using the Q5 Site-DirectedMutagenesis Kit (E0554S, NEB) according to manufacturer's instructions with the following primers: DIAPH-LVX-I862A-Fwd: 5′-GAATCTCTCAGCGTTTTTGGGTTCCTTCC-3′ and DIAPH-LVX-I862A-Rev: 5′-TGGGCTGTCTTTGAATC-3′. The GFP-tagged *DIAPH1* pcDNA5/FRT/TO vector was generated with the *DIAPH1* open reading frame amplified from cDNA using the primers: DIAPH-GFP-NotI-Fwd: 5′-GTTGTTGCGGCCGCATGGAGCCGCCCGGCGGGAG-3′ and DIAPH-GFP-XhoI-Rev: 5′-GTTGTTCTCGAGTTAGCTGCACGCCCAACCAACTCCTTG-3′.

### siRNA transfection

Plasmids and siRNA oligos were transfected in OptiMEM reduced serum medium using Lipofectamine 2000 and Oligofectamine (Thermo Fisher Scientific), respectively according to the manufacturer's guidelines. A custom siRNA targeting lacZ (CGUACGCGGAAUACUUCGAdTdT)[59] and was used as a control siRNA. All other siRNAs used were On-Targetplus SMARTpools (Horizon). Transfected cells were analysed at 72 h post-transfection.

### Genotoxic agents

CPT (Selleckchem), ETOP (Selleckchem), olaparib (Selleckchem), KU-55933 ATM inhibitor (Selleckchem), mirin (Selleckchem) and aphidicolin (Sigma-Aldrich) were used as indicated. IR treatment was carried out using a Faxitron CellRad as indicated.

### Colony survival assay

HeLa cells transfected with control and DIAPH1 siRNA were plated at low density and exposed to increasing doses of ionizing radiation. Colonies were fixed and stained after 14 days with 2% methylene blue (Sigma-Aldrich) in 50% ethanol and counted. Data is expressed as a percentage survival normalized to an untreated control for each siRNA.

### Immunoblot analysis and antibodies

Whole cell extracts were prepared by sonication for 2 × 10 s in 8 M Urea, 50 mM Tris pH 7.5, 150 mM β-mercaptoethanol and centrifuged at 16,000 rpm for 20 min at 4 °C. Fifty micrograms whole cell extract was subjected to SDS-Page and electroblotting[60]. Immunoblotting was performed using antibodies to: pS1981 ATM (AF1655, 1:500) from R&D systems; ARP2 (sc-166103, 1:500) and U1-70K (sc-390988, 1:2000) from Santa Cruz Biotechnology; ATM (A300-299A, 1:500), pS824-KAP1 (A300-767A, 1:1000), KAP1 (A300-274A, 1:3000), pS966-SMC1 (A300-050A, 1:1000), SMC1 (A300-055A, 1:1000), Chk2 (A300-681A, 1:1000), MDC1 (A300-052A, 1:500) and DIAPH1 (A300-077A, 1:500) from Fortis Life Sciences; pT68-CHK2 (2197, 1:500) from Cell Signalling Technology; Histone H2A (07-146, 1:1000); γ-H2AX (05-636, 1:3000), β-actin (A5316, 1:10000) and HA (H9658, 1:3000) from Sigma-Aldrich; pS343-NBS1 (ab47272, 1:500), MRE11 (ab214, 1:2000), Histone H3 (ab1791, 1:2000) and ARPC4 (ab217065, 1:1000) from Abcam; Nbs1 (GTX70224, 1:10000) from GeneTex; BRCA1 (OP95, 1:500), RPA2 (NA18, 1:1000), γ-actin (MABT824, 1:10000) and RAD51 (PC130, 1:500) from Merck; Vinculin (66305-1, 1:5000) from Proteintech. Uncropped versions of all Western blots are provided as a Source Data file.

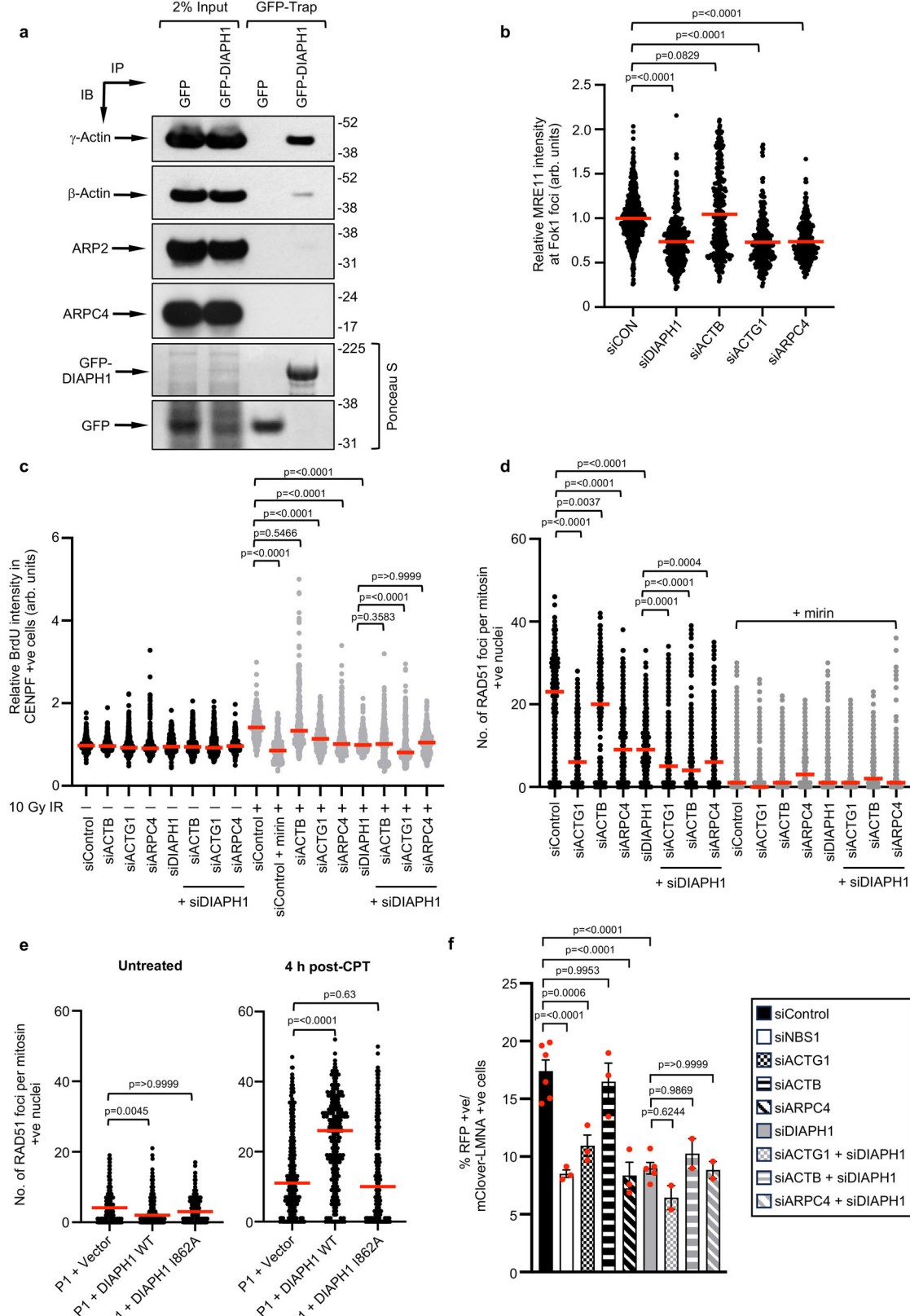

## GFP-Trap affinity purification

For GFP-Trap pulldown experiments with 293FT cells, cells transfected with plasmids using Lipofectamine 2000 and harvested 48 h post-transfection. Cells were incubated in lysis buffer (150 mM NaCl, 50 mM Tris HCl pH 7.5, 2 mM MgCl₂, 1% NP40, 90 U/ml Benzonase (Novagen) and EDTA-free protease inhibitor cocktail [Roche]) for 30 min with

rotation at 4 °C. Cell lysates were then pre-cleared at 44,000 × *g* at 4 °C for 30 min. For GFP-Trap, 3–5 mg of lysate was incubated with GFP-Trap agarose beads (ChromoTek) for 5 h at 4 °C. The resulting GFP-Trap complexes were washed with wash buffer (150 mM NaCl, 50 mM Tris HCl pH 7.5, 0.5% NP40, and complete protease inhibitor cocktail [Roche]) and analysed by SDS−PAGE.

**Fig. 7 | DIAPH1, γ-actin and the Arp2/3 complex function to promote MRE11-dependent localisation to DSBs and DNA end-resection. a** GFP or GFP-DIAPH1 was purified from 293FT cell extracts using GFP-Trap and co-purified proteins were subjected to SDS-PAGE and Western blotting with the antibodies indicated. Representative of $n = 2$ independent experiments. **b** U-2 OS-FokI cells were transfected with the indicated siRNAs and then fixed/immunostained with an antibody to MRE11 4 h after FokI induction. The fluorescence intensity of MRE11 per FokI focus was quantified from 100 cells per experiment. The median of $n = 3$ independent experiments is shown (red line). **c** siRNA transfected HeLa cells were labelled with 25 μM BrdU for 24 h, treated with 10 Gy IR for 1 h and then permeablised/fixed 4 h post-irradiation. The intensity of nuclear BrdU staining was quantified in 100 S/G2 cells per experiment. Cells were treated with 50 μM mirin 1 h prior to irradiation where indicated. The mean of $n = 3$ independent experiments is shown (red line). **d** HeLa cells were transfected with the indicated siRNAs, exposed to 100 nM CPT for 1 h and then fixed/permeabilised 4 h post-treatment. RAD51 foci

were quantified in a minimum of 100 S/G2 cells per experiment. The mean of $n = 3$ independent experiments with the SEM is shown (red line). **e** Quantification of CPT-induced RAD51 foci in DIAL syndrome patient P1-derived fibroblasts complemented with either an empty vector, WT DIAPH1 or the I862A DIAPH1 mutant. RAD51 foci were quantified in a minimum of 100 S/G2 cells per experiment. The mean of $n = 3$ independent experiments with the SEM is shown (red line). **f** siRNA transfected HeLa cells were transfected with the plasmids required to initiate HR at a Cas9-induced DSB located within the *LMNA* locus. HR (% RFP positive cells with mClover-LMNA staining) was quantified using immunofluorescence. The mean of $n = 3$ independent experiments is shown with the SEM (red line). A minimum of 500 cells were counted per time point, per experiment. Statistical significance was calculated using: (**b**–**e**) a Kruskal–Wallis test ([**b**] $p = <0.0001$, [**c**] $p = <0.0001$, [**d**] $p = <0.0001$, [untreated] $p = 0.0044$, [4 h post-CPT] $p = <0.001$) and (**f**) an ordinary one-way ANOVA ($p = <0.0001$). Source data are provided as a Source Data file.

## Immunofluorescence microscopy

Cells were seeded onto coverslips 24 h before extraction and fixation. Cells were pre-extracted for 5 min on ice with ice-cold extraction buffer (25 mM HEPES, pH 7.4, 50 mM NaCl, 1 mM EDTA, 3 mM MgCl₂, 300 mM sucrose and 0.5% Triton X-100) and then fixed with 3.6% paraformaldehyde for 10 min. Fixed cells were stained with primary antibodies specific to γ-H2AX (Sigma-Aldrich, 05-636, 1:1000), RAD51 (Sigma-Aldrich, PC130, 1:500), 53BP1 (Novus-Biologicals, NB100-904, 1:1000), RPA2 (Abcam, ab2174, 1:250), BrdU (Abcam, BU1/75 (ICR1), 1:100), MRE11 (Abcam, ab214, 1:1000), BRCA1 (Santa Cruz Biotechnology, sc-6954, 1:200), CENPF (Antibody Atlas, HPA052382, 1:1000), Mitosin (BD Transduction labs, 610768, 1:1000) with secondary antibodies conjugated to Alexa Fluor 488 and Alexa Fluor 594 (Life Technologies), and then with DAPI (VectaLabs). Images were visualized using a Nikon Eclipse Ni microscope with NIS-Elements software (Nikon Instruments) and captured using a 100× oil-immersion objective. When quantifying γ-H2AX foci in G1 or G2-phase of the cell cycle, 3 μM aphidicolin was added to the cells 1 h prior to irradiation and left in during the entire time course.

## Sample preparation for stochastic optical reconstruction microscopy (STORM)

DIAPH1-P1 fibroblasts expressing either an empty vector or WT DIAPH1 were seeded onto glass coverslips (Fisher Scientific) in a 6-well plate at a density of $2.3 \times 10^5$ cells per well and treated with 100 nM CPT for 1 h. 30 min into CPT treatment, cells were incubated with 10 μM EdU for 30 min prior to harvest to detect nascent DNA synthesis. Cells were then permeabilised with 0.5% Triton X-100 in ice-cold CSK buffer (10 mM HEPES, 300 mM Sucrose, 100 mM NaCl, 3 mM MgCl₂, pH 7.4) for 3 min at room temperature (RT) followed by 3 × PBS washes. Cells were fixed with 4% paraformaldehyde (Electron Microscopy Sciences) in PBS for 10 min, at RT. After 2 × PBS washes, cells were washed with blocking buffer (2% glycine, 2% BSA, 0.2% gelatine, and 50 mM NH₄Cl in PBS) for 3 × 5 min. Incorporated EdU was detected using the Click-iT Plus EdU Imaging Kit (Thermo Fisher Scientific, C10640) and labelled with Janelia Fluor 549-aizde (Tocris Biosciences, 6501). Cells were then washed with blocking buffer for 3 × 5 min followed by an overnight incubation in blocking buffer at 4 °C. Cells were incubated with primary antibodies against MRE11, (Novus [12D7] AF488 conjugated NB100473, 1:3000), BRCA1 (Novus, [RAY] AF488 conjugated NB100598, 1:1700) and RPA70 (Abcam, [EPR3472] AF647 conjugated ab199240, 1:3000) in blocking buffer for 1 h, at RT. After 3 × 5 min washes with blocking buffer, stained coverslips were mounted onto a glass microscope slide and freshly prepared imaging buffer (1 mg/mL glucose oxidase (Sigma, G2133), 0.02 mg/mL catalase (Sigma, C3155), 10% glucose (Sigma, G8270), 100 mM mercaptoethylamine (Fisher Scientific, BP2664100)) was added underneath the coverslip immediately prior to imaging. Cells were then imaged and Auto-PC analysis was performed as previously reported[61,62].

## STORM and data analysis

Following antibody staining, the coverslips were mounted on microscope glass slides followed by injecting freshly prepared super resolution (SR) imaging buffer consisting of (1 mg/mL glucose oxidase (Sigma, G2133), 0.02 mg/mL catalase (Sigma, C3155), 10% glucose (Sigma, G8270), 100 mM mercaptoethylamine (Fisher Scientific, 100995) in PBS, pH 8.0). STORM images were acquired using a custom-built optical imaging platform based on an ASI-RAMM inverted microscope, with similar features as described[61–64]. Briefly, different fluorophores were excited by different laser lines. The 488, 561 and 639 nm laser lines were used to excite the samples at Highly Inclined and Laminated Optical sheet (HILO) illumination mode. Each colour was filtered by a single-band pass filter (Semrock, FF01-676/37 and FF01-607/36 for AF647 and JF549, respectively). The filters were switched via a filter wheel (ASI, FW-1000) for sequential illumination. The imaging sequence in this study was always green (EdU; JF549), blue (BRCA1; monoclonal AF488 conjugated) and red (MRE11/RPA; AF647). A 405 nm laser line (Applied Scientific Pro., SL-405 nm–150 mW) was also introduced to drive photoswitching of Alexa Fluor 647 fluorophores and enhance recovery back to their ground state. The emission profile was expanded with a 2× lens tube followed by chromatic aberration correction using a correction lens (Thorlabs, AC254-300-A) and finally collected on a sCMOS camera (Photometrics, Prime 95B) at 33 Hz (30 ms per frame) for a minimum of 2000 frames. Single-molecule localization was performed following a regular DAOSTORM routine[65–69]. For each colour, 2000 frames were acquired at 33 Hz using Micro-Manager software (Version 2.0).

Each frame acquired during imaging was image stacked and all samples were equally weighted within a square region of the image with a box size of 4 times of the FWHM of a 2D Gaussian PSF. Maximum Likelihood Estimation algorithm was utilized to perform the 2D-Gaussian single-PSF fitting in GPU (Nvidia GTX 1060, CUDA 8.0) and the fitting accuracy was assessed by Cramér-Rao lower bound as described[62,70]. Region-Of-Interests (ROI, $a \sim 4 \times 4\ \mu m^2$ square of the 3-color SMLM image of a nucleus) were created and submitted to the Auto-PC function and fraction analysis as described[61,62,70]. For pair-correlation analyses, localizations appeared within 2.5 times of the average localization precision in consecutive frames was averaged and considered as one localization from one blinking event. The coordinates list was then directly submitted for pair-correlation analyses. To quantify the degree of colocalization we calculated the cross-correlation between nascent DNA and target protein of interest (MRE11/BRCA1/RPA). Next, we multiplied the amplitude for the cross-correlation values with the local density of the coordinates (number of coordinates per ROI) for nascent DNA to determine the likelihood of finding the target protein molecule around nascent DNA on average. The differential between the average distribution of all ROIs between drug treated or non-treated samples were subjected to two-sample *t*-test to obtain *p*-value scores. For the fraction analysis each nucleus was

manually outlined to generate an ROI for independent analysis. Representative images were rendered from the coordinate list to the 10 nm pixel canvas and blurred with a Gaussian kernel ($\sigma$ = 10 nm) for display purpose. SMLM image analysis was performed as previously described using Matlab code[61–64,70].

## Homologous recombination assay

HeLa cells were transfected with 50 nM ON-Targetplus siRNA (Horizon) overnight using Oligofectamine (Thermo Fisher Scientific) as per manufacturer's instructions. siRNA depleted HeLa cells were then seeded onto coverslips 8 h prior to DNA transfection. Two micrograms of pLX330-LMNA-gRNA-1 (Addgene #122507), 2 µg of pCR2.1-mClover-LMNA-Donor (Addgene #122508) and 2 µg piRFP670-N1 (Addgene #45457) and 5 µl of Lipofectamine 2000 (Thermo Fisher Scientific) was added to 200 µl of OptiMEM (Thermo Fisher Scientific) and incubated for 20 min at RT. The plasmid/lipofectamine complex was added to the cells and incubated for 48 h at 37 °C. 48 h post-DNA transfection, cells were harvested by aspirating the media and fixed with 2 ml of 3.6% paraformaldehyde (PFA)/PBS for 10 min at RT. PFA was removed and the cells were permeabilised for 5 min with 0.5% Triton X-100 in PBS at RT. Cells were then incubated in blocking buffer (10% FCS in PBS) and mounted onto microscope slides with Vectashield containing DAPI (VectorLabs). The efficiency of HR was measured by quantifying the number of RFP positive cells with mClover-Lamin rings.

## DR-GFP HR reporter assay

DR-GFP U-2 OS reporter cells (obtained from Jeremy Stark)[71] were transfected with the indicated siRNAs in conjunction with plasmids expressing RFP and the I-SceI endonuclease using FuGene6 (Promega). Twenty-four hours post-transfection, culture media was replaced with fresh media and the cells were left to grow for a further 48 h. Cells were then harvested and fixed for 10 min in ice cold 4% PFA/PBS. Cells expressing RFP alone or GFP and RFP were quantified by flow cytometry using a Beckman Coulter CytoFLEX.

## FokI DSB repair protein recruitment assay

U-2 OS ER-mCherry LacI-FokI-DD expressing cells (provided by Roger Greenberg)[37] were subjected to siRNA transfection. 24 h prior to treatment with Shield1 and 4-hydroxytamoxifen, siRNA depleted cells were seeded onto glass cover slips. Cells were then incubated with 5 µM ATMi or 50 µM mirin 1 h prior to adding 1 µM Shield1 ligand (Takara Bio, cat: 632189) and 1 µM 4-hydroxytamoxifen (Merck, cat: H6278-10MG) for 4 h. After the 4 h drug incubation, cells were fixed in 3.6% PFA/PBS for 10 min then incubated with CSK extraction buffer for 5 min. Coverslips were washed 3 times in PBS and stored at 4 °C in 10% FBS/PBS containing 0.05% Sodium Azide. Coverslips containing cells were stained with an antibody to MRE11 (Abcam, 1:1000, ab214) or DIAPH1 (Fortis Life Sciences, 1:250, A300-077A) and an anti-mouse secondary antibody coupled to Alexa-488 (Invitrogen, 1:1000, A21202). The fluorescence intensity of MRE11 at FokI-mCherry loci was quantified using ImageJ software and normalised to control siRNA.

## Sister chromatid exchange analysis and metaphase spreads

For SCE analysis, cells were incubated with 10 µM BrdU for 48 h before incubating with 0.2 µg/ml demecolcine for 3 h. Cells were then resuspended in 0.075 M KCl and incubated at 37 °C for 1 h, fixed in methanol/acetic acid (3:1) and dropped onto microscope slides. The slides were then incubated in 10 µg/ml Hoechst 33342 for 20 min and exposed to UVA light for 1 h in 2× SSC buffer. Slides were incubated in 2× SSC buffer for 1 h at 60 °C and stained with 5% Giemsa. Metaphase spreads to quantify chromosome breakage were prepared as previously described[59]. Briefly, Colcemid (KaryoMAX, Thermo Fisher) was added at a final concentration of 0.2 µg/ml for 3 h. Cells were then harvested by trypsinization, subjected to hypotonic shock for 30 min at 37 °C in hypotonic buffer (10 mM KCl, 15% FCS) and fixed in 3:1 ethanol:acetic acid solution. Cells were dropped onto acetic-acid-humidified slides, stained for 15 min in Giemsa-modified solution (Merck; 5% v/v in water) and washed in water for 5 min before being visualised by light microscopy. For metaphase spread analysis of cells treated with exogenous DNA damage, cells were incubated with 0.5 nM CPT or 10 nM ETOP or exposed to 2 Gy IR 24 h before harvesting.

## AsiSI real-time PCR DNA resection assay

U-2 OS AsiSI-ER cells[38] were transfected with siRNA as described above. Forty-eight hours after transfection, cells were treated with 300 nM 4-OHT for 4 h in growth media to induce DNA DSB. Cells were harvested and genomic DNA (gDNA) was extracted using a DNAeasy kit as per the manufacturer's instructions (Qiagen), followed by elution in 50 µl of nuclease-free water. Extracted gDNA was treated with 5 units RNaseH for 15 min at 37 °C. Five microliters of DNA (approximately 0.5–1 µg DNA) was used for each condition (BsrG1 digestion and matched undigested sample). DNA digests were carried out in 30 µl with 20 units of BsrG1 enzyme in CUTSMART digestion buffer (NEB), at 37 °C overnight. BsrG1 enzyme was heat-inactivated (80 °C for 20 min). Twenty microliters of nuclease-free water was added to each sample and 2 µl of the sample was used directly for RT-PCR. The percentage of ssDNA (ssDNA%) generated by resection at selected sites was determined as previously described (37). Briefly, for each sample, a ΔCt was calculated by subtracting the Ct value of the mock-digested sample from the Ct value of the digested sample. The undigested sample serves as a loading control for genomic DNA for each experimental condition. The ssDNA% was calculated with the following equation: ssDNA% = $1/(2^{(\Delta Ct-1)} + 0.5) \times 100$. Primers used for the qPCR: AsiSI BsrG1 335 bp For: GAATCGGATGTATGCGACTGATC; AsiSI BsrG1 335 bp Rev: TTCCAAAGTTATTCCAACCCGAT; AsiSI BsrG1 1618 bp For: TGAGGAGGTGACATTAGAACTCAGA; AsiSI BsrG1 1618 bp Rev: AGGACTCACTTACACGGCCTTT.

## Proximity ligation assay

Cells seeded onto coverslips were labeled with 10 µM EdU for 20 min, permeabilized for 5 min using ice-cold nuclear extraction buffer (10 mM PIPES, 300 mM sucrose, 20 mM NaCl, 3 mM MgCl$_2$, 0.5% Triton X-100) and then fixed in 4% PFA for 10 min. When indicated, cells were treated with 100 nM CPT for 1 h following the EdU pulse. The click reaction was performed with the Click-iT Kit (Invitrogen C10337, C10339) according to the manufacturer's instructions. Cells were blocked in 10% FCS/PBS-T for 1 h at room temperature and incubated with primary antibodies anti-γ-H2AX (Sigma-Aldrich, 05-636, 1:2000) and anti-DIAPH1 (Fortis Life Sciences, A300-077A, 1:1000) in 10% FCS/PBS-T for 1 h at RT. Cells were washed 3 × 5 min in PBS-T followed by a 1 × 5 min wash in Wash buffer A. The cells were then incubated with anti-Mouse MINUS and anti-Rabbit PLUS PLA Probes (Sigma Aldrich DUO82004, DUO82002) for 1 h at 37 °C. After incubation with the PLA Probes, cells were subjected to proximity ligation and detection with DUOLINK detection Kit following the manufacturer's instructions (Sigma Aldrich DUO92008, DUO92014). Coverslips were mounted with Prolong antifade with DAPI (Invitrogen P36935) and cells were imaged on a Nikon Eclipse Ni microscope in conjunction with Elements v4.5 software (Nikon). PLA foci in EdU-positive and EdU-negative cells were quantified for each condition and data were analyzed with Prism software.

## Statistical analysis

Statistical analyses were performed as indicated in the figure legends. A p-value of less than 0.05 indicates significance. The number of independent experimental replicates is denoted in each figure legends.

**Reporting summary**

Further information on research design is available in the Nature Portfolio Reporting Summary linked to this article.

## Data availability

The datasets generated during WES are not publicly available due to restrictions associated with the ethical approval. Accession numbers for genes/proteins analysed within this study are: Human DIAPH1 NM_005219 [https://www.ncbi.nlm.nih.gov/nuccore/NM_005219.5] and NM_005210 [https://www.ncbi.nlm.nih.gov/protein/NP_005210.3], Human ACTB NM_001101 [https://www.ncbi.nlm.nih.gov/nuccore/NM_001101.5] and NM_001092 [https://www.ncbi.nlm.nih.gov/protein/NP_001092.1], Human ACTG1 NM_001199954 [https://www.ncbi.nlm.nih.gov/nuccore/NM_001199954.3] and NM_001605 [https://www.ncbi.nlm.nih.gov/protein/NP_001605.1]. Plasmids obtained from Addgene (https://www.addgene.org/) used in this study: pLV-hTERT-IRES-hygro (Addgene #85140), psPax2 (Addgene #12260) and pMD2.G (Addgene #12259). Source data are provided with this paper.

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

## Acknowledgements

We would like to thank the parents and patients from the DIAL syndrome and BWCFF syndrome families for taking part in this study and generously donating tissue samples. We would also like to thank Roger Greenberg, Jeremy Stark and Gaëlle Legube for gifting the ER-mCherry LacI-FokI-DD expressing U-2 OS cells, DR-GFP U-2 OS cells and *Asi*SI-ER U-2 OS cells, respectively. G.S.S., S.S.J., M.R.G. and A.N.G. are funded by a CR-UK Programme Grant (C17183/A23303). MRG is also supported by a Ramón y Cajal Award (RYC2022-036513-I) from the MICIU of Spain and funding from the MICIU/AEI/10.13039/501100011033 of Spain, co-founded by the FEDER program of EU (PID2023-150261NA-I00). B.L.W. is supported by a CR-UK Clinical Academic Training Programme award (C11497/A31309). L.G. was funded by a Lister Institute Summer Studentship. A.S.C. is supported by an MRC project grant (UKRI577). V.G.D., S.E., R.M. and H.H. are funded by the Wellcome Trust (WT093205MA, WT104033AIA), National Institute for Health Research (NIHR), the University College London Hospitals (UCLH), and the Medical Research Council (HH). E.R. and S.L. are supported by the NIH (R35 GM134947, AI153040 and CA247773) and a V Foundation BRCA1,2-Convergence Team Award collaborative grant. A.J.M. was supported by the National Science Center (Poland) (2015/17/B/NZ3/03604).

## Author contributions

V.G.D., R.M., S.E., A.M., H.C.M., E.A.F., S.D.R., S.V., G.D.M., C.C., F.S., E.G.D., P.A., S.I., M.S.Z., A.M.R.T., T.S., A.J.M. and N.D.D. provided clinical information and cell lines. C.G.M. and M.A.S. carried out whole exome sequencing. B.L.W. and L.E.G. carried out SCE and chromosome breakage analysis, immunofluorescence and homologous recombination analysis. A.A. and S.S.J. helped BLW establish specific experimental techniques. T.L.C. carried out assessment of DNA DSB end-resection. E.R. supervised B.L.W. and S.L. when carrying out STORM. A.S.C. performed the colony survival assays, the FokI recruitment assay, the DR-GFP assay and cell fractionation. M.R.G. carried out the PLA. M.P. purified DIAPH1 and the MRN complex and carried out in vitro binding assays. G.S.S. generated complemented cell lines, carried out immunofluorescence and co-precipitation studies and performed Western blotting. A.N.G. generated DIAPH1 expression vectors. A.S. processed skin biopsies to generate fibroblasts for analysis. F.M.H. carried out statistical analysis of the experimental data. H.H. supervised V.G.D., S.E. and R.M. G.S.S. planned and supervised the study and wrote the manuscript. All authors contributed to editing the manuscript.

## Competing interests

The authors declare no competing interests.

## Additional information

Beth L. Woodward[1,24], Sudipta Lahiri[2,24], Anoop S. Chauhan[1,24], Marcos Rios Garcia[1,3], Lucy E. Goodley[1], Thomas L. Clarke[4], Mohinder Pal[5], Angelo Agathanggelou[1], Satpal S. Jhujh[1], Anil N. Ganesh[1], Fay M. Hollins[1], Valentina Galassi Deforie[6], Reza Maroofian[6], Stephanie Efthymiou[6], Andrea Meinhardt[7], Christopher G. Mathew[8,9], Michael A. Simpson[9], Heather C. Mefford[10], Eissa A. Faqeih[11], Sergio D. Rosenzweig[12], Stefano Volpi[13,14], Gigliola Di Matteo[15,16], Caterina Cancrini[15,16], Annarita Scardamaglia[6], Fiona Shackley[17], E. Graham Davies[18], Shahnaz Ibrahim[19], Peter D. Arkwright[20], Maha S. Zaki[21], Tatjana Stankovic[1], A. Malcolm R. Taylor[1], Antonina J. Mazur[22], Nataliya Di Donato[23], Henry Houlden[6], Eli Rothenberg[2] & Grant S. Stewart[1]✉

[1]Department of Cancer and Genomic Sciences, College of Medical and Health, University of Birmingham, Birmingham, UK. [2]Department of Biochemistry and Molecular Pharmacology, New York University School of Medicine, New York, NY, USA. [3]Department of Physiology, CiMUS, University of Santiago de Compostela, Santiago de Compostela, Spain. [4]Department of Pathology and Laboratory Medicine, Boston University Chobanian and Avedisian School of Medicine, Boston, MA, USA. [5]School of Natural Sciences, University of Kent, Canterbury, UK. [6]Department of Neuromuscular Diseases, Queen Square Institute of Neurology, University College London, London, UK. [7]Institute for Clinical Genetics, University Hospital Carl Gustav Carus at TUD Dresden University of Technology and Faculty of Medicine of TUD Dresden University of Technology, Dresden, Germany. [8]Sydney Brenner Institute for Molecular Bioscience, University of the Witwatersrand, Johannesburg, South Africa. [9]Department of Medical and Molecular Genetics, Faculty of Life Science and Medicine, King's College London, Guy's Hospital, London, UK. [10]Center for Pediatric Neurological Disease Research, St. Jude Children's Hospital, Memphis, TN, USA. [11]King Fahad Medical City, Children's Hospital,, Riyadh, Kingdom of Saudi Arabia. [12]Immunology Service, Department of Laboratory Medicine, Clinical Center, National Institutes of Health, Bethesda, MD, USA. [13]UOC Reumatologia e Malattie Autoinfiammatorie, IRCCS Istituto Giannina Gaslini, Genoa, Italy. [14]DINOGMI, Università degli Studi di Genova, Genoa, Italy. [15]Department of Systems Medicine, Tor Vergata University, Rome, Italy. [16]Research Unit of Primary Immunodeficiencies, Unit of Clinical Immunology and Vaccinology, Scientific Institute for Research and Healthcare (IRCCS) Bambino Gesù Children Hospital, Rome, Italy. [17]Paediatric Immunology, Allergy and Infectious Diseases, Sheffield Children's Hospital NHS Foundation Trust, Sheffield, UK. [18]Department of Immunology and Gene Therapy, Great Ormond Street Hospital for Children NHS Foundation Trust, London, UK. [19]Department of pediatrics and child health, Aga Khan University, Karachi, Pakistan. [20]Lydia Becker Institute of Immunology and Inflammation, University of Manchester, Manchester, UK. [21]Clinical Genetics Department, Human Genetics and Genome Research Division, National Research Centre, Cairo, Egypt. [22]Department of Cell Pathology, Faculty of Biotechnology, University of Wroclaw, Wroclaw, Poland. [23]Institute for Human Genetics, Hannover Medical School, Hannover, Germany. [24]These authors contributed equally: Beth L. Woodward, Sudipta Lahiri, Anoop S. Chauhan. ✉e-mail: g.s.stewart@bham.ac.uk

