## [Transparent Peer Review file · Nature Communications]

Inherited deficiency of DIAPH1 identifies a DNA double strand break repair pathway regulated by γ -actin

Corresponding Author: Professor Grant Stewart

Version 0:

Reviewer comments:

Reviewer #1

(Remarks to the Author)

In this study, the authors identify disease-associated mutations in the DIAPH1 gene and find that these lead to a DNA repair defect. They use a variety of readouts to show that the defect is in homologous recombination, and that it impacts on early events, such as recruitment of repair proteins, including the MRN complex, to DNA breaks. Because DIAPH1 is an actin nucleating factor, they further investigate the similarity of this phenotype to cells with mutations in either beta or gamma Actin, and find that the HR repair defect is shared by cells with mutations in gamma Actin and with a DIAPH1 point mutant incapable of nucleating actin filaments shows a nice phenotype. These results provide a nice explanation for the similarity in symptoms between patients with mutations in DIAPH1, gamma Actin, and NBN.

Overall, the experiments are appropriate for identifying the activities in HR that might be impacted by DIAPH1. As the authors point out, there is already evidence that actin and its modulation play a role in DNA repair in the nucleus, so the potential for DIAPH1 acting at DNA breaks is reasonable and interesting and builds on the picture of events at DNA breaks. However, the data here don't go far enough to firmly determine whether DIAPH1 is working at DNA breaks or instead affecting DNA repair indirectly somehow. In addition, there are a lot of issues with data presentation, annotation, and analysis that need to be addressed.

Main issues:

1. The only evidence that DIAPH1 is playing a direct role during HR is the GFP trap experiments that point towards an interaction with a subset of HR proteins (MRE11, NBS1 and RAD51). Because this involves overexpression of a tagged protein and no other evidence is provided, it is too preliminary to draw conclusions. The authors have assays in which they can test the possibility that DIAPH1 is present at DNA breaks (e.g. STORM, ChIP-seq or IF at the FokI locus), so this should be tested to provide some evidence as to whether DIAPH1 contributes directly or not to HR. Additional data regarding protein-protein interactions would also help support this conclusion.
2. What is the fate of DIAPH1-mutant cells following genotoxic stress? Is there a survival defect? Also, can the authors provide data on fitness and cell cycle profile when DIAPH1 is mutated? These properties might impact indirectly on DNA damage responses, so they are important to understand.

Other major issue:

While the assays are all appropriate to the questions being addressed, the presentation of the data lacks clarity in places, and this makes it difficult to assess the data quality as well as the variability and magnitude of defects. This can in most cases be addressed by providing additional information and/or raw data. Specifically,

1. Analysis of gH2AX foci should be plotted as the mean number of foci per cell at each time point and not grouped as in Fig 2d or treated as a binary variable as in Fig 3a or just one threshold used as in Fig 5. Alternatively, provide the raw data in an Excel format as supplementary information to allow readers to evaluate the variability and spread of the data. As presented, the differences are difficult to critically assess.
2. In Figure 3b, does each individual dot represent a single metaphase? In which case, there should be 90 dots, but it appears to be just one experiment plotted here? All data points should be plotted or provided in some format. Also, for Fig 3a and 3b, why use the median rather than the mean? Is it because there is skew in the data? If so, indicate this and discuss. If not, use the mean.
3. Figure 3d and f – both assays involve foci counting, but in one case, the data are presented using presence/absence of foci and in the other, the authors provide the number of foci per cell. The number of foci should be used for both assays,

since this gives much better insights, or an explanation for not using number of foci per cell should be provided (perhaps with the raw data as a supplementary item). Further, the median is used for 3f, while mean is used for 3d. Please provide an explanation for this choice, again commenting on skew in the data if applicable.

Minor issues:

1. The quality of the figures is variable. For example, grainy quality of bar graphs (e.g. Fig 5b and 5c). It looks like these were cut and pasted.
2. The labelling on Fig 1c is cut off at the top.
3. The loading control for Fig 2b only has 5 lanes. Wrong image used?
4. Multiple inconsistencies and mistakes between labelling on figures compared to text and/or figure legend. Some examples: Supplementary figure 2 legend is incorrect (e and f should be d and e). Supplementary figure 3 legend is incorrect (patient annotations in legend don't match figure). Figure 4e – is this IR or CPT? The legend and figure annotation don't match. Figure 5, the legend is mislabelled (panels f and g). Figure 6 panel c also incorrect. Mutations in Figures S7 and S8 don't match. Figure 6, the labels on panel f don't match order in bar graph.
5. Information, such as time points and drug concentrations, is inconsistently provided on figures (for example labels on Fig 5b and 5c, one has this and one doesn't).
6. There is a Supplementary Table referred to in text, but none provided.

Reviewer #2

(Remarks to the Author)

The study by Woodward et al. identified and characterized how disease-associated mutations or deficiency in the actin-related DIAPH1 gene product impacts DSB repair in different human cell systems. The authors propose DIAL (DIAPH1 Loss-of-function) syndrome as a novel genome instability disorder associated with an inability to repair DNA double-strand breaks (DSBs). Individuals afflicted with DIAPH1 mutations can present with microcephaly, short stature, seizures, intellectual disability, and visual impairment (refs 10-14). Overall, the data presented support a role for DIAPH1 in DSB repair and the maintenance of genome stability. For instance, patient-linked DIAPH1 mutations and/or knockdown of DIAPH1 appear to increase gH2AX and 53BP1 foci counts, micronuclei, and chromosome aberrations while decreasing sister chromatid exchanges and HR-based repair of a reporter locus, consistent with decreased genome repair and elevated genome instability. DIAPH1 deficient cells also show less RAP2, RAD51, and Mre11 foci, and less DSB-proximal ssDNA, suggesting a defect in reaching the resection step of HR-mediated DSB repair. Overexpressed and GFP-tagged DIAPH1 exhibits a very strong cytoplasmic localization but appears to interact with RAD51, MRE11, and NBS1 in pulldown assays. Surprisingly, phenotypes linked to DIAPH1 deficiency appear to be shared with those observed following the loss of known actin regulators linked to DSB repair (e.g., ARPC4) and appear to be shared with gamma-ACTIN (gACT) but not beta-ACTIN (bACT), possibly providing an important mechanistic distinction related to the role of different actin isoforms in the maintenance of genome stability. The authors' proposal to rename DIAPH1 deficiency conditions as DIAL to better distinguish it from SCBMS is reasonable and may be helpful on the clinical front.

Overall, the area of study is of great interest and timeliness. In addition, the study is generally well conducted and has a translational research aspect that will interest clinical and basic scientists, particularly at the level of disorders linked to radiosensitivity, genome instability, some developmental defects, and cancer predisposition. That said, several major and minor concerns exist, as outlined below. If satisfactorily revised, this study would be an impactful and important addition to several fields.

Major points:

* Patient data show slight increases in gH2AX only at 24 h in DIAPH1 mutant cells vs normal cells (Fig 2a). In contrast, siDIAPH1 shows increased gH2AX signals at 1, 4, and 8 hrs are seemingly even higher than at 24 h (Fig S1a). These gH2AX kinetics raise some concerns related to the comparison of patient data (DIAPH1 mutants) and siRNA data (DIAPH1 knockdown). Also, these siRNA data especially suggest that repair kinetics implicated in other more rapid DNA repair mechanisms, such as NHEJ, may still be significantly impacted in siDIAPH1 conditions. The study requires an improved assessment of DNA repair kinetics.

* If DIAPH1 is truly important for DSB repair, why is it that all DIAPH1-deficient cells (mutants or siRNA), under untreated conditions (meaning no exogenous genotoxic agent), appear to not exhibit significant changes in gH2AX foci, 53BP1 foci or micronuclei counts, or in unrepaired chromosome break signals. The only exceptions to this appear to be a very subtle increase in global gH2AX levels by WB in Fig2a at time zero and the elevated average number of chromosome aberrations/metaphase spread. Therefore, the current data overall suggests that DIAPH1 may be partly important to promote the repair of endogenous DSBs but is primarily important for the repair of DSBs induced by genotoxic agents. In some cases, is it possible that the quantification approaches used, meaning quantifying gH2AX using arbitrary groups (1-5, 6-10, and >10 foci), may be an explanation? Is this group-based quantification approach sensitive enough to pick up moderate, albeit significant, differences in untreated DIAPH1-deficient conditions? If not, this may mask a potentially greater role for DIAPH1 in repairing DSBs induced by endogenous or exogenous DNA damage sources. Are the categories-based quantifications derived from a single imaging plane or the entire nucleus? What would happen if the total number of gH2AX foci/nucleus is counted for the studied cell populations? Would it change some of the findings/conclusions?

*Some quantification of total DSB foci counts versus size is necessary to better assess the effect of DIAPH1 on DSB counts versus clustering.

*Related to the patient-derived DIAPH1-mutant data, it's important to have all mechanistic analyses conducted using at least one common type of patient cells. For example, most data are obtained using DIAPH1-P1 while some complementary data (such as Fig S4a) were from DIAPH1-P4. Similarly, the inconsistency of presenting genome instability data in the paper is not helpful. For example, gH2AX switches from a categorical quantification of foci numbers to a total quantification of the % of cells with foci. More details and clarifications of the experimental setup used would be helpful.

*The data in Fig 3a is not described accurately in the text (lines 242-244). These data show that the presence of DIAPH1 decreases the % of cells with gH2AX foci slightly in G1 and G2 at 1 h post-IR, but mostly decreases the % of cells with gH2AX foci in G2 at 8 h. These data would be more consistent with the contribution of DIAPH1 to both the slow-acting HR in G2 and the fast-acting NHEJ in G1. Either way, these points should be described and interpreted more accurately. Also related to this point, the information on lines 246-248 is inaccurate since ETOP-induced DSBs can be robustly repaired by both HR and NHEJ. In addition, for the GFP-LMNA reporter, it is important to add some negative control such as the siRNA-mediated knockdown of an NHEJ repair factor. Similar NHEJ controls are necessary for Fig 3d-f and 4c.

*Regarding Fig 4e, why was the statistical analysis not conducted in a way to compare IR/siControl/Mirin to IR/siDIAPH1? The combination of Mirin and siDIAPH1 with a more complete statistical analysis is necessary to evaluate the data accurately. Similarly, statistical analyses comparing plus-Vector to plus-DIAPH1 conditions should be included in Fig4a-b.

*The data in Fig S6d are puzzling. DIAPH1 does not appear to show any significant nuclear localization in the presence or absence of IR, although the quality of the microscopy data shown here appears limited. In fact, the authors show that DIAPH1 is primarily cytoplasmic regardless of DNA damage levels, suggesting that the protein may perform a cytoplasmic function to promote nuclear DSB repair. The study would greatly benefit from improved microscopy. Related to this point, are there NLS or NES signals within DIAPH1, and how does their mutation alter the protein's localization and function? And do any of the disease-linked DIAPH1 mutations fall within such localization signals? The study also necessitates improved microscopic analyses comparing the localization of DIAPH1, gACT, and bACT in the presence or absence of DNA-damaging agents to better understand how these factors may intersect in nucleus and/or cytoplasm during DSB repair.

*It is important to assess whether the reported protein-protein interactions of DIAPH1 can be recapitulated between endogenous proteins by coIP and if assays such as PLA can detect interaction foci within the nucleus and/or cytoplasm in the absence and presence of IR.

*The statistical comparisons in Fig 6c/d are incomplete and confusing. I was looking to compare many more of the conditions shown on these plots, but the necessary statistical information was not provided.

*It would further strengthen the study to assess DSB levels and cell viability when DIAPH1-deficient cells are treated with PARP inhibitors.

*In lines 329-331 the authors are again pointing out general similarities between the BWCF syndrome and their described DIAL syndrome. Similarities have already been mentioned throughout the text multiple times. However, pointing out specific features of DIAL and how it differs from other actin-defect-related syndromes would be more helpful.

Minor points:

* Sentence split on lines 166-167: It is recommended to either remove this sentence or publish this other study or deposit a preprint and cite it here.

*Line 217: missing figure number.

*Regarding refs 32/33 and lines 234/235. The characterization of the evidence in these studies is not entirely accurate and should be revised.

*Why are the HR reporter values in Fig 6f double those in Fig 3c for the same conditions?

*Sentence on lines 413-415: difficult to read/follow and should be improved for clarity.

*The text labels within the figures often show overlapping letters and truncated words (e.g. Fig 1c and 1e). This should be carefully reviewed and corrected throughout the figures.

*The statistical comparisons throughout the figures often fail to compare key conditions. The stats should be carefully reviewed for completion and accuracy.

*Not a single image includes a scale bar in the study.

*Results in Fig4a-b: should the y-axis be "MRE11/EdU colocalization" or similar? The "average MRN11 at DNA" is rather confusing. What exactly was quantified in the graph in Fig 4a? The shown images are overall not clear. Only EdU is visible

in the nuclei. Are both images showing cells after CPT treatment? This information should be specified in the figure itself.

*The data presentation in Fig 6c-d is cumbersome and could be improved.

*It was often difficult to determine exactly which cell type was used in every figure panel.

*Line 399: shouldn't it cite Fig 6f instead of Fig 6e?

*Including a schematic summary model summarizing the findings would be helpful. The model could also depict the order of the proposed repair events better illustrate how exactly DIAL appears to differ from other related but not identical syndromes/diseases.

Reviewer #3

(Remarks to the Author)

In this manuscript, Stewart and colleagues propose a previously unrecognized regulatory role of DIAPH1 on the replication-associated DNA double-strand breaks (DSB) repair pathway by homologous recombination (HR). Significantly, this role was discovered by the systematic characterization of patient-derived cell lines from DIAL (DIAPH1 Loss-of-function) Syndrome patients that all carry homozygous frameshift or nonsense mutations in the DIAPH1 gene, which codes for a member of the formin family of actin nucleators. DIAL cell lines are defective for HR repair of replication associated lesions, probably due to defective initial steps of HR (i.e. DNA end resection). These observations may explain the significant overlap of clinical symptoms of DIAL Syndrome with Nijmegen and Warsaw breakage syndrome, both also HR deficiency disorders. The strength of this manuscript lies in the careful and comprehensive analysis of patient-derived cell lines and their wildtype reconstituted derivatives. Together, these data strongly support the notion that loss-of-function mutations in DIAPH1 yields an increase in chromosomal instability by abrogating early steps of HR repair. Having said this, the manuscript also holds some weaknesses. For example, the mechanistic assessment of DIAPH1 deficiency on MRE11 recruitment is still rather preliminary and is not yet very convincing. However, in summary, this is not only the comprehensive characterization of a new human genetic disorder but also an important and timely contribution to the emerging role of nuclear actin on genomic stability maintenance. Together, these two elements make for a manuscript that is clearly suitable for publication in Nat commun upon appropriate revisions (suggestions follow below).

1) The authors convincingly show that absence of DIAPH1 causes a decrease in sister-chromatid exchange and gene targeting. However, both assays are not ideal HR readouts. For example, an HR-independent mechanism of sister chromatid exchange was recently proposed (Heijink et al., 2022, Nat commun). A quasi-standard HR assay in the field is the GFP reconstitution assay (often referred to as the "Maria Jasin assay"). I would suggest the authors to perform this assay in cells depleted of DIAPH1 (siDIAPH1 cells).

2) The MRE11 recruitment defect in DIAPH1 deficient cells is not very convincing (Fig. 4a). I do not see a clear difference in the STORM images between P1 + Vector and P1 + DIAPH1 cells. The magnified foci examples seem to show a difference in the vicinity of the MRE11 signal to the EdU signal, but it is not clear how this was quantified. Also, the increase in MRE11/EdU colocalization in P1 + DIAPH1 cells is minor. It is also difficult to comprehend why the STORM images represent MRE11 recruited to replication-associated breaks, while the normal IF images in Supplementary Fig 5 represent MRE11 recruitment to damaged chromatin. This distinction appears somewhat random. iPOND analysis has shown that the MRN complex is an intrinsic component of the DNA replication fork. Perhaps the absence of DIAPH1 leads to premature dissociation of MRN from stalled or damaged replication forks rather than defective recruitment of sites of replication-associated lesions. I propose to either investigate the effect of DIAPH1 loss on MRN recruitment more comprehensively or to tone down the MRE11 recruitment aspect and emphasize the defective DNA end resection, which may or may not be caused by a suboptimal recruitment of MRE11.

3) The specific role of gActin in HR is intriguing, given its similarity with bActin. This interesting and important observation could be investigated in more detail. For example, GFP-DIAPH1 seems to preferentially interact with g-Actin, while interaction with b-Actin is minimal (Fig 6a). Is this also the case for the reverse IP (i.e. GFP-gActin or GFP-bActin pull downs)? If yes, which one of the four distinct amino acids is required for the differential interaction? This could be determined by site-directed mutagenesis.

Version 1:

Reviewer comments:

Reviewer #1

(Remarks to the Author)

The revised manuscript is strengthened by the inclusion of new data and text in response to issues raised by the reviewers. A few things still need to be addressed. Additional images should be provided for Fig 5a showing examples of no colocalization (and no FokI induction). Controls (omission of each antibody separately) are not provided for the new PLA assay (Fig. 5b).

Prompted by the responses to reviewers, I looked at the source data file and found that in some cases, the source data is not

adequately provided. For example, Excel tab for Fig 5a shows the percentage colocalization, which is already apparent in the figure. The raw data is not provided. Same for Fig 5d and Sup Fig 5a.

The inclusion of foci numbers in a subset of assays is useful. Presenting the data by group leads to a loss of important information, and just because grouping is used in the literature does not make it best practice.

Reviewer #2

(Remarks to the Author)

The manuscript is much improved, though some concerns remain.

Regarding Major Points 1 and 5:

Although the authors highlight some valuable points, the patient-derived fibroblast results differ significantly from the HeLa results. Specifically, the data presented for the siRNA DIAPH1 in HeLa cells suggest that DIAPH1 may still influence repair mechanisms with faster kinetics than homology repair (HR). Regarding Figure 3a-b, first, the authors should add the DIAPH1-dependent percent decreases in the number of gH2AX foci per cell directly on the graphs to better highlight the magnitude of the change. That said, assessing cells at approximately 2-4 hours (i.e., expanding Figure 3a-b) would have been more informative; currently, one hour could be too early while eight hours could be too late to capture key differences in NHEJ or other fast-acting repair processes within the specific experimental setup used. Also, adding data from an NHEJ reporter would have more definitively eliminated the potential impact of DIAPH1 on this repair mechanism.

In addition, the new DR-GFP reporter results presented in the authors' response to Reviewer #3 raise additional questions. The authors' preferred interpretation that the endogenously integrated LMNA reporter may be more accurate than Jasin's DR-GFP reporter is plausible. However, the weak effect of DIAPH1 loss in the DR-GFP assay compared to the LMNA assay might instead reflect multiple studies linking lamins to double-strand break (DSB) repair through various processes, as well as the well-established connections between LMNA mutations, genome instability, chromatin status, and ageing. In other words, this LMNA assay tests the repair of a locus whose products can facilitate DNA repair through different mechanisms. Consequently, inducing damage within the LMNA gene might give an initial global hit to DNA repair, which could then artificially amplify the effect of DIAPH1 loss on DSBR in this specific assay. Therefore, the DR-GFP assay results shown in the rebuttal letter should be incorporated into the manuscript for completion and transparency. Finally, without additional analyses of repair kinetics and DNA repair pathway choice, the authors could clearly state in the results or discussion section that:

- a. their data cannot entirely rule out the possibility that DIAPH1 is not involved in DNA repair pathways beyond HR.
- b. future research should further explore the potential impact of DIAPH1 on DNA repair pathway choice.

Regarding Major Point 2: The new data, explanation, and statement in the discussion section that "DIAPH1 is essential for facilitating HR-dependent repair of DNA DSBs induced by IR, CPT, and ETOP" effectively address this concern. This statement should be retained in the final version of the paper. The authors should also consider briefly capturing this point in the abstract, particularly that the effect of DIAPH1 on DSBR has only been observed for, or may be limited to, exogenously induced DNA damage.

Regarding Major Points 6 and 9: While the authors have included some useful statistical analyses, a simple t-test seems to be employed throughout the manuscript to compare one sample against multiple other samples. However, the t-test is intended for comparing only two means at a time. Performing multiple t-tests separately for each comparison increases the risk of Type I errors (false positives) due to multiple comparisons. Thus, the statistical analyses still need attention. At a minimum, a warning should be added to the statistical analysis section of the methods to highlight this risk. Since most phenotypes discussed in the manuscript demonstrate strong effects, this reviewer will defer to the handling editor to determine the best course of action on the choice of statistical tests used by the authors.

Regarding Major Point 10: The data shown here are informative. Though it is not required, the authors should consider including them in the manuscript. This could be useful to the ongoing mechanistic debates surrounding PARP inhibition.

Regarding Minor Point 1: This reviewer will defer to the handling editor to determine whether referring to a future manuscript that has yet to be prepared is compatible with their policies. Is this equivalent to "data not shown" and could such a statement be used instead?

Other points:

In the literature, DIAPH1 appears to show characteristics consistent with it being a candidate oncogene or a candidate tumor suppressor (e.g., see PMIDs 36503156, 24105619, 39879317, 26124177, 30094535). The authors refer to DIAPH1 as a tumor suppressor (e.g., see line 172). It is unclear to this reviewer if DIAPH1 meets the criteria for classification as a tumor suppressor or oncogene. What are the authors' thoughts on this point? This point should be better addressed in the text.

Supp Fig. 8a: Why does DNA damage induction appear to lower DIAPH1 levels in the nucleus?

Supp Fig 18 b: This fig shows ACTB KO cells have statistically significant decreases in RAD51 foci formation. This appears to contradict the text on line 418.

Lines 195-197: Unclear sentence; should it be "Prompt g-H2AX recovery was restored or rescued in Patient P1 fibroblasts when complemented with WT DIAPH1"?

Lines 257-260: "To verify this finding, ... the cell cycle"; add reference(s).

Line 273: It should be "following exposure to CPT"; "to" is missing.

Line 340: The callout to the relevant figure(s) is missing; should it be Fig. 5d and Supp. Fig. 8b?

Line 529: correct "the formation of localised an F-actin"

Reviewer #3

(Remarks to the Author)

This is a re-review of the revised version of the manuscript: "Inherited deficiency of DIAPH1 identifies a DNA double strand break repair pathway regulated by gamma-actin" by Stewart and colleagues. My assessment of the first version of the paper was generally positive, with the suggestion of three main directions of further experimental work that would have strengthened the impact of the work. Unfortunately, the authors did not address any of my suggestions satisfactorily. They did follow my suggestion to test the impact of DIAPH1 depletion in the "Maria Jasin assay", but the results show only a minor reduction in percentage of HR upon depletion of DIAPH1 as compared to e.g. BRCA2 depletion. This poses the question as to how relevant DIAPH1 really is for HR.

My criticism of MRE11 localisation was only textually addressed and new, more clear images were provided. To be honest, I still don't understand this assay and therefore, I won't further comment on it.

My final suggestion on site-directed mutagenesis and reverse IP of DIAPH1 with b-Actin and g-Actin was not addressed at all, with the explanation that this was "well outside the scope of this manuscript".

In summary, my impression of this work has therefore not changed. It is potentially interesting in my view, but important issues remain unanswered.

Whether or not this manuscript should thus be accepted for publication in Nature communications must be an editorial decision and will not be further commented on by this reviewer.

Reviewer #1 (Remarks to the Author)

In this study, the authors identify disease-associated mutations in the DIAPH1 gene and find that these lead to a DNA repair defect. They use a variety of readouts to show that the defect is in homologous recombination, and that it impacts on early events, such as recruitment of repair proteins, including the MRN complex, to DNA breaks. Because DIAPH1 is an actin nucleating factor, they further investigate the similarity of this phenotype to cells with mutations in either beta or gamma Actin, and find that the HR repair defect is shared by cells with mutations in gamma Actin and with a DIAPH1 point mutant incapable of nucleating actin filaments shows a nice phenotype. These results provide a nice explanation for the similarity in symptoms between patients with mutations in DIAPH1, gamma Actin, and NBN.

Overall, the experiments are appropriate for identifying the activities in HR that might be impacted by DIAPH1. As the authors point out, there is already evidence that actin and its modulation play a role in DNA repair in the nucleus, so the potential for DIAPH1 acting at DNA breaks is reasonable and interesting and builds on the picture of events at DNA breaks. However, the data here don't go far enough to firmly determine whether DIAPH1 is working at DNA breaks or instead affecting DNA repair indirectly somehow. In addition, there are a lot of issues with data presentation, annotation, and analysis that need to be addressed.

Main issues:

1. The only evidence that DIAPH1 is playing a direct role during HR is the GFP trap experiments that point towards an interaction with a subset of HR proteins (MRE11, NBS1 and RAD51). Because this involves overexpression of a tagged protein and no other evidence is provided, it is too preliminary to draw conclusions. The authors have assays in which they can test the possibility that DIAPH1 is present at DNA breaks (e.g. STORM, ChIP-seq or IF at the FokI locus), so this should be tested to provide some evidence as to whether DIAPH1 contributes directly or not to HR. Additional data regarding protein-protein interactions would also help support this conclusion.

Response: We agree with the reviewers comment regarding our evidence about whether DIAPH1 plays a direct role in regulating DSB repair. As suggested, we have used the FokI DSB assay to determine whether DIAPH1, γ -actin or β -actin are localised to sites of DSBs. This assay showed that endogenous DIAPH1 is robustly recruited to sites of FokI-induced DSBs (Fig.5a). Interestingly, DIAPH1 was only localised to approximately 50% of Mre11 positive, FokI-induced DSBs, suggesting that DIAPH1 may be involved in modulating the repair of a subset of DSBs by HR. In contrast we were unable to assess whether γ -actin or β -actin were recruited to FokI-induced DSBs due to their very high level of expression in cells masking any specific immunofluorescence staining at the DSB.

As an alternative approach to the FokI system, we have also used PLA to assess whether γ -actin or β -actin are localised to CPT-induced DSBs. Importantly, this assay demonstrated that endogenous DIAPH1 and γ -actin but not β -actin are localised to CPT-induced DSBs marked by γ H2AX but only in EdU positive cells (Fig.5b and Supplementary Fig.15). Therefore, this data is consistent with DIAPH1 and γ -actin working directly to regulate HR at sites of DSBs. We have incorporated this new data into the manuscript and altered the main text accordingly.

To extend our understanding of how DIAPH1 is recruited to DSBs, we have used the FokI assay again and either depleted MRE11 or inhibited ATM or MRE11 and assessed the efficiency of DIAPH1 relocalisation to FokI-induced DSBs. Strikingly, this demonstrated that loss or inhibition of ATM or MRE11 partially prevented DIAPH1 localisation to DSBs (Fig. 5d). This indicates that a feedback loop exists in which the MRN complex binds to DSBs and activates ATM. The activation of ATM helps to stimulate the recruitment of DIAPH1, which in turn helps to recruit or stabilise the MRN complex on chromatin surrounding a DSB undergoing resection. We have incorporated this data into a working model that we have presented as a diagram in the supplementary information (Supplementary Fig. 20) as requested by reviewer 2.

2. What is the fate of DIAPH1-mutant cells following genotoxic stress? Is there a survival defect? Also, can the authors provide data on fitness and cell cycle profile when DIAPH1 is mutated? These properties might impact indirectly on DNA damage responses, so they are important to understand.

Response: We have carried out a colony survival assay in DIAPH1 depleted cells following exposure to ionising radiation and shown that cells lacking DIAPH1 are hypersensitive to IR (Supplementary

Fig.5a). We have also examined the cell cycle distribution in cells depleted of DIAPH1. This analysis did not reveal any alterations in the cell cycle when DIAPH1 is absent (Supplementary Fig.5b).

Other major issue:

While the assays are all appropriate to the questions being addressed, the presentation of the data lacks clarity in places, and this makes it difficult to assess the data quality as well as the variability and magnitude of defects. This can in most cases be addressed by providing additional information and/or raw data. Specifically:

1. Analysis of gH2AX foci should be plotted as the mean number of foci per cell at each time point and not grouped as in Fig 2d or treated as a binary variable as in Fig 3a or just one threshold used as in Fig 5. Alternatively, provide the raw data in an Excel format as supplementary information to allow readers to evaluate the variability and spread of the data. As presented, the differences are difficult to critically assess.

Response: We disagree with the reviewer that quantifying the number of cells with different amounts of H2AX foci i.e. 1-5, 6-10 and >10 foci per cell, prevents readers from evaluating the variability and spread of the data. Within the DNA damage field, researchers commonly use either method to quantify DNA damage foci i.e. quantifying the percentage of cells with >5 or >10 foci per cell or quantifying the number of foci per cell. Below are some recent examples of papers published in Nature Communications where the authors quantified the percentage of cells with >5 or >10 DNA damage-induced foci.

1. Yalcin et al. (2024). Nature Commun. 15:5032
2. Tischler et al. (2024). Nature Commun. 15:866
3. Tao et al. (2023). Nature Commun. 14:7430
4. Claessens et al. (2023) Nature Commun. 14:5893
5. Scaramuzza et al. (2023) Nature Commun. 14:5071

Furthermore, this method of quantification is clearly suitable for differentiating between alterations in the repair of DSBs in WT cells, DIAPH1 mutant cells and cells lacking NBS1 (Supplementary Fig. 4a-d). Therefore, we stand by our foci quantification method as being scientifically valid.

In addition to this, we carry out the quantification of foci by eye, looking down the microscope rather than using an automated computer program. As such the main reason why we utilised our approach for quantifying DNA damage-induced foci in groups is that it allows for the quantification of foci to be carried out on a large numbers of cells relatively quickly. As indicated in the figure legend, we quantified the DNA damage-induced foci in >500 cells per time point, per cell line, per experimental repeat i.e. a total of >1500 cells were analysed per time point, per cell line over three independent experiments. As such, we believe that our data is much more representative of the total population of irradiated cells analysed than counting the number of foci in a low number of cells e.g. 100 cells.

Despite this, as requested we have quantified the number of H2AX foci per cell in WT fibroblasts and *DIAPH1*, *ACTB*, *ACTG1*, *ARPC1B*, *ARPC4* and *ARPC5* mutant patient fibroblasts before and 1 and 24 hours after exposure to 1Gy IR. Consistent with our original method of quantifying H2AX foci, we observed an increase in the number of foci per cell remaining at 24 hours post-irradiation when DIAPH1 is absent or when *ACTG1*, *ARPC1B*, *ARPC4* or *ARPC5* but not *ACTB* are mutated (Supplementary Fig.1b and 10). We should point out that since we quantified H2AX foci/cell across an asynchronous population of cells, it is likely that the mean number of H2AX foci in the DIAPH1 mutant cells remaining at 24 hours post-irradiation is lower than if foci had been quantified specifically in S/G2 cells only. However, since we have already analysed IR-induced H2AX foci in G1 vs G2 phase cells (Fig.3a), we opted to carry out the requested quantification of H2AX per cell on an asynchronous population to complement our analysis on asynchronous cells in Fig. 2d.

Finally, all the raw data associated with this manuscript was included as part of our original submission so that readers can evaluate the variability and spread of the data. Furthermore, the raw data will be updated to include all the new data added to our manuscript during the revision process.

2. In Figure 3b, does each individual dot represent a single metaphase? In which case, there should be 90 dots, but it appears to be just one experiment plotted here? All data points should be plotted or provided in some format. Also, for Fig 3a and 3b, why use the median rather than the mean? Is it because there is skew in the data? If so, indicate this and discuss. If not, use the mean.

Response: We thank the reviewer for bringing this to our attention. The data for only one experiment had been plotted in error rather than the combination of the three independent experiments that had been carried out. We apologise for this oversight. Furthermore, we found a similar thing had happened with the SCE quantification in Supp figure 5d. This was to do with how we had laid out the 3 independent experiments in GraphPad. We have amended these figures to include the data for all three independent experiments. We have replotted Figures 3a, 3b and Supp Figure 5d with the mean highlighted rather than the median, since the data is normally distributed. However, the raw data for the quantification of all three independent experiments presented in Figure 3b and Supp Figure 5d analyzing SCE frequency were supplied as part of our initial submission.

3. Figure 3d and f – both assays involve foci counting, but in one case, the data are presented using presence/absence of foci and in the other, the authors provide the number of foci per cell. The number of foci should be used for both assays, since this gives much better insights, or an explanation for not using number of foci per cell should be provided (perhaps with the raw data as a supplementary item). Further, the median is used for 3f, while mean is used for 3d. Please provide an explanation for this choice, again commenting on skew in the data if applicable.

Response: The reason for counting cells with BRCA1 foci rather than the number of BRCA1 foci per cell was that even using relatively low doses of CPT, the amount of BRCA1 foci were too numerous to count accurately. As such, we felt that it was better to count the number of S/G2 cells with BRCA1 foci to normalise for any changes in the cell cycle. We have added a line in the figure legend stating why we carried out this type of analysis. As for Rad51 foci, the same amount of CPT induced far less Rad51 foci per cell than BRCA1 foci, such that they could be accurately counted. Hence the reason why Figure 3f shows Rad51 foci per cell. Lastly, since the distribution of the data shown in Figures 3e and 3f is normally distributed, as suggested, we have amended these figures to show the mean rather than median of three independent experiments. Furthermore, as mentioned above, the raw data for all our quantification presented in this manuscript was supplied as part of our initial submission.

Minor issues:

1. The quality of the figures is variable. For example, grainy quality of bar graphs (e.g. Fig 5b and 5c). It looks like these were cut and pasted.

Response: For some reason these graphs did not format well when the manuscript pdf file was generated. This has been rectified.

2. The labelling on Fig 1c is cut off at the top.

Response: This has been rectified.

3. The loading control for Fig 2b only has 5 lanes. Wrong image used?

Response: We apologize for this. This was a mistake during figure assembly. Five lanes were cropped from the Western blot rather than six. We have rectified this. However, the uncropped Western blot was supplied as part of our initial submission and can also be found in the raw data accompanying the revised manuscript.

4. Multiple inconsistencies and mistakes between labelling on figures compared to text and/or figure legend. Some examples: Supplementary figure 2 legend is incorrect (e and f should be d and e). Supplementary figure 3 legend is incorrect (patient annotations in legend don't match figure). Figure 4e – is this IR or CPT? The legend and figure annotation don't match. Figure 5, the legend is mislabelled (panels f and g). Figure 6 panel c also incorrect. Mutations in Figures S7 and S8 don't match. Figure 6, the labels on panel f don't match order in bar graph.

Response: We apologise for these mistakes. Our manuscript went through many iterations and consequently some small mistakes were overlooked. These have been rectified in the revised version of the manuscript.

5. Information, such as time points and drug concentrations, is inconsistently provided on figures (for example labels on Fig 5b and 5c, one has this and one doesn't).

Response: We apologise for this. This information was not always included on the figure due to space limitations. We have now added the drug concentrations and time points on the figures where possible. However, all information about drug concentrations and time points used are present in the figure legends.

6. There is a Supplementary Table referred to in text, but none provided.

Response: We apologize for this oversight. It appears the supplementary table was not included during original submission. The supplementary table has now been included with our revised manuscript.

Reviewer #2

The study by Woodward et al. identified and characterized how disease-associated mutations or deficiency in the actin-related DIAPH1 gene product impacts DSB repair in different human cell systems. The authors propose DIAL (DIAPH1 Loss-of-function) syndrome as a novel genome instability disorder associated with an inability to repair DNA double-strand breaks (DSBs). Individuals afflicted with DIAPH1 mutations can present with microcephaly, short stature, seizures, intellectual disability, and visual impairment (refs 10-14). Overall, the data presented support a role for DIAPH1 in DSB repair and the maintenance of genome stability. For instance, patient-linked DIAPH1 mutations and/or knockdown of DIAPH1 appear to increase gH2AX and 53BP1 foci counts, micronuclei, and chromosome aberrations while decreasing sister chromatid exchanges and HR-based repair of a reporter locus, consistent with decreased genome repair and elevated genome instability. DIAPH1 deficient cells also show less RAP2, RAD51, and Mre11 foci, and less DSB-proximal ssDNA, suggesting a defect in reaching the resection step of HR-mediated DSB repair. Overexpressed and GFP-tagged DIAPH1 exhibits a very strong cytoplasmic localization but appears to interact with RAD51, MRE11, and NBS1 in pulldown assays. Surprisingly, phenotypes linked to DIAPH1 deficiency appear to be shared with those observed following the loss of known actin regulators linked to DSB repair (e.g., ARPC4) and appear to be shared with gamma-ACTIN (gACT) but not beta-ACTIN (bACT), possibly providing an important mechanistic distinction related to the role of different actin isoforms in the maintenance of genome stability. The authors' proposal to rename DIAPH1 deficiency conditions as DIAL to better distinguish it from SCBMS is reasonable and may be helpful on the clinical front.

Overall, the area of study is of great interest and timeliness. In addition, the study is generally well conducted and has a translational research aspect that will interest clinical and basic scientists, particularly at the level of disorders linked to radiosensitivity, genome instability, some developmental defects, and cancer predisposition. That said, several major and minor concerns exist, as outlined below. If satisfactorily revised, this study would be an impactful and important addition to several fields.

Response: We thank the reviewer for their positive comments about our manuscript.

Major points:

1. Patient data show slight increases in gH2AX only at 24 h in DIAPH1 mutant cells vs normal cells (Fig 2a). In contrast, siDIAPH1 shows increased gH2AX signals at 1, 4, and 8 hrs are seemingly even higher than at 24 h (Fig S1a). These gH2AX kinetics raise some concerns related to the comparison of patient data (DIAPH1 mutants) and siRNA data (DIAPH1 knockdown). Also, these siRNA data especially suggest that repair kinetics implicated in other more rapid DNA repair mechanisms, such as NHEJ, may still be significantly impacted in siDIAPH1 conditions. The study requires an improved assessment of DNA repair kinetics.

Response: We are unsure as to why the reviewer has concerns regarding the differences between the IR-induced gamma-H2AX kinetics in patient-derived skin fibroblasts and HeLa cells (Fig.2a versus Supplementary Fig.1a). HeLa cells are an aneuploid tumour cell line with high levels of endogenous replication stress and multiple pathogenic gene mutations. Whereas in contrast, patient-derived fibroblasts are a diploid cell line only immortalised with hTERT. Given that HeLa cells have more than twice the DNA content than fibroblasts, grow significantly quicker than fibroblasts and have high levels of spontaneous replication stress and DNA damage, it is not surprising that the IR-induced gamma-H2AX kinetics are different between HeLa cells and fibroblasts. As such, one should **only** compare the IR-induced gamma-H2AX response between the WT and patient-derived fibroblasts or the control and DIAPH1 siRNA treated HeLa cells. In both cases, the only conclusion that can be made is that fibroblasts and HeLa cells lacking DIAPH1 display residual amounts of gamma-H2AX at 24h post-irradiation when compared to the appropriate control cell line, which is indicative of a DSB repair defect. Any differences in the kinetics of the IR-induced gamma-H2AX response between fibroblasts and HeLa cells are not relevant. Lastly, at the start of our manuscript, we were not focusing on whether DIAPH1 regulated different DSB repair pathways, we were only interested in assessing whether DIAPH1 played any role in regulating the repair of DSBs. Based on the data presented in Figure 2 and Supplementary Figures 1-4, using complemented patient-derived cell lines, HeLa cells depleted of DIAPH1 and three different DSB-inducing agents, we believe that we have unequivocally demonstrated that when DIAPH1 is lost or depleted, it results in a DSB repair defect. As such, we do not feel that improving the assessment of DSB repair kinetics will substantiate this conclusion any further.

2. If DIAPH1 is truly important for DSB repair, why is it that all DIAPH1-deficient cells (mutants or siRNA), under untreated conditions (meaning no exogenous genotoxic agent), appear to not exhibit significant changes in γ H2AX foci, 53BP1 foci or micronuclei counts, or in unrepaired chromosome break signals. The only exceptions to this appear to be a very subtle increase in global γ H2AX levels by WB in Fig2a at time zero and the elevated average number of chromosome aberrations/metaphase spread. Therefore, the current data overall suggests that DIAPH1 may be partly important to promote the repair of endogenous DSBs but is primarily important for the repair of DSBs induced by genotoxic agents. In some cases, is it possible that the quantification approaches used, meaning quantifying γ H2AX using arbitrary groups (1-5, 6-10, and >10 foci), may be an explanation? Is this group-based quantification approach sensitive enough to pick up moderate, albeit significant, differences in untreated DIAPH1-deficient conditions? If not, this may mask a potentially greater role for DIAPH1 in repairing DSBs induced by endogenous or exogenous DNA damage sources. Are the categories-based quantifications derived from a single imaging plane or the entire nucleus? What would happen if the total number of γ H2AX foci/nucleus is counted for the studied cell populations? Would it change some of the findings/conclusions?

Response: As mentioned to reviewer 1, who had a similar criticism, we carry out the quantification of foci in multiple planes across the entire nucleus by eye, looking down the microscope rather than using an automated computer program. As such the main reason why we utilised our approach for quantifying DNA damage-induced foci in groups is that it allows for the quantification of foci to be carried out on a large numbers of cells relatively quickly. As indicated in the figure legend, we quantified the DNA damage-induced foci in >500 cells per time point, per cell line, per experimental repeat i.e. a total of >1500 cells were analysed per time point, per cell line over three independent experiments. As such, we believe that our data is much more representative of the total population of irradiated cells analysed than counting the number of foci in a low number of cells e.g. 100 cells. Furthermore, this method of quantification is clearly suitable for differentiating between alterations in the repair of DSBs in WT cells, DIAPH1 mutant cells and cells lacking NBS1 (Supplementary Fig. 4a-d).

Lastly, as indicated in our rebuttal to reviewer 1, within the DNA damage field, researchers commonly use different methods to quantify DNA damage foci i.e. quantify the percentage of cells with >5 or >10 foci per cell or quantify the number of foci per cell. Below are some recent examples of papers published in Nature Communications where the authors quantified the percentage of cells with >5 or >10 DNA damage-induced foci. Therefore, we stand by our foci quantification method as being scientifically valid.

1. Yalcin et al. (2024). Nature Commun. 15:5032
2. Tischler et al. (2024). Nature Commun. 15:866
3. Tao et al. (2023). Nature Commun. 14:7430 (Fig.4f, 4h, 5k)
4. Claessens et al. (2023) Nature Commun. 14:5893
5. Scaramuzza et al. (2023) Nature Commun. 14:5071

However, as the reviewer correctly points out, this type of quantification might miss subtle/mild DSB repair defects particularly in untreated cells. As such, we have quantified the number of H2AX foci per cell in WT fibroblasts and *DIAPH1*, *ACTB*, *ACTG1*, *ARPC1B*, *ARPC4* and *ARPC5* mutant patient fibroblasts before and 1 and 24 hours after exposure to 1Gy IR. Consistent with our original method of quantifying γ H2AX foci, we observed an increase in the number of foci per cell remaining at 24 hours post-irradiation when DIAPH1 is absent or when *ACTG1*, *ARPC1B*, *ARPC4* or *ARPC5* but not *ACTB* are mutated (Supplementary Fig. 1b and 10). We should point out that since we quantified H2AX foci/cell across an asynchronous population of cells, it is likely that the mean number of H2AX foci in the DIAPH1 mutant cells remaining at 24 hours post-irradiation is lower than if foci had been quantified specifically in S/G2 cells only. However, since we have already analysed IR-induced H2AX foci in G1 vs G2 phase cells already (Fig.3a), we opted to carry out the requested quantification of H2AX per cell on an asynchronous population to complement our analysis on asynchronous cells in Fig.2b. Finally, this analysis did not revealed any increase in the level of spontaneous H2AX foci in patient cells lacking DIAPH1 or with mutations in *ACTG1*, *ARPC1B*, *ARPC4*, *ARPC5* or *ACTB*. This would suggest that either the DIAPH1- γ -actin pathway is only required to repair DSBs induced by exposure to exogenous genotoxic agents or more likely, the level of spontaneous damage occurring in diploid, non-transformed fibroblasts is sufficiently low that it does not reach a threshold to require the DIAPH1- γ -actin-dependent repair pathway or is beyond the level of detection using conventional fluorescence microscopy.

3. Some quantification of total DSB foci counts versus size is necessary to better assess the effect of DIAPH1 on DSB counts versus clustering.

Response: To address these points we have performed additional SMLM experiments of γ H2AX and nascent DNA in DIAPH1 patient fibroblasts complemented with either an empty vector or WT DIAPH1. This allows precision detection of the fluorescence signal and enables quantification of the specific molecular features of foci. This new data is presented below.

For analysis of the specific features of foci detecting in the SMLM data, we utilized the DBSCAN clustering algorithm along with Nearest Neighbouring Distance (NND) approach (see Yin et al. (2021) Mol Cell. 81:4243-4257 and Lee et al. (2021) Nature Commun. 12:2525), which provides the degree of co-localization of γ H2AX and nascent DNA clusters. This analysis clearly demonstrates that there is an increase in the association of γ H2AX with nascent DNA at broken forks (seDSBs) upon CPT treatment in DIAPH1-deficient cells compared to cells re-expressing WT DIAPH1 (see figure above part 'a').

Based on the reviewer's suggestion, we further applied the DBSCAN clustering algorithm to obtain the geometric features of γ H2AX clusters that were detected via SMLM. The enhanced detection sensitivity of SMLM makes it especially useful for resolving low-level signal associated with basal damage (see Yin et al. (2021) Mol Cell. 81:4243-4257). This analysis revealed a noticeable increase in the average cluster size of γ H2AX foci in DIAPH1-deficient cells compared to cells re-expressing DIAPH1 both at a basal level and upon the induction of damage (see figure above part 'b'). These results verify the effect and importance of DIAPH1 in promoting DSB repair under unperturbed conditions and highlights the greater role for DIAPH1 in repairing DSBs induced by endogenous as well as exogenous DNA damage sources.

Finally, it is worth noting that Western blotting and conventional fluorescence microscopy do not provide the level of detection sensitivity that is provided by single molecule imaging (as seen above), nor are these methods useful for detecting singular localisation events or sub-populations of localisation events within individual cells. Further information about our high resolution microscopy technology can be found in our previously published papers e.g. Yin and Rothenberg (2016) Science Reports. 6:30819; Yin et al. (2019) Nature Commun. 10:119; Yin et al. (2021) Mol Cell. 81:4243-4257.

4. Related to the patient-derived DIAPH1-mutant data, it's important to have all mechanistic analyses conducted using at least one common type of patient cells. For example, most data are obtained using DIAPH1-P1 while some complementary data (such as Fig S4a) were from DIAPH1-P4. Similarly, the

inconsistency of presenting genome instability data in the paper is not helpful. For example, gH2AX switches from a categorical quantification of foci numbers to a total quantification of the % of cells with foci. More details and clarifications of the experimental setup used would be helpful.

Response: Throughout the manuscript, we have tried to carry out all our mechanistic analysis using multiple different cell lines to rule out the possibility that the phenotypes we have observed are cell line specific. In this respect, we have validated the presence of a DIAPH1/actin-dependent DNA repair pathway using multiple patient-derived cell lines and tumour cell lines:

- (1) 6 DIAL syndrome cell lines, 5 BWCFF syndrome cell lines with *ACTB* mutations, 5 BWCFF syndrome cell lines with *ACTG1* mutations, 2 *ARPC1B* mutant cell lines, 1 *ARPC4* mutant cell line and 1 *ARPC5* mutant cell line
- (2) complemented patient-derived cell lines
- (3) siRNA-mediated gene knockdown in HeLa and U2OS cells
- (4) CRISPR knockout A375 cells.

We feel it is very important to show that we observe similar cellular phenotypes by disrupting the same pathway using different approaches. Occasionally, we have switched between using fibroblasts and LCLs from patients because fibroblasts are better for quantifying foci due to them being adherent, whereas LCLs are better for carrying metaphase spreads or assessment of the ATM-dependent DNA damage response because they grow quickly. However, in most cases, we have verified that we observe similar cellular phenotypes in HeLa cells depleted of DIAPH1 or A375 cells lacking *ACTB* or *ACTG1*. Based on this, we believe that using multiple different approaches and cell models to validate a phenotype is a strength not a weakness.

Lastly, we switched from counting the '% of cells with foci' (Fig.2d) to counting the 'number of foci per cell' (Fig. 3a) because in Fig.3a we were using low dose irradiation and examining foci resolution at 8h post-irradiation to ensure that we were assessing DSB repair in irradiated G2 cells before they transitioned through mitosis and into the next G1 phase, which would occur at later time points post-irradiation. From our experience we cannot distinguish mild DSB repair defects in G2 phase cells by using the '% of cells with foci' method as all the cells have >10 foci. Despite this, whether we switch from quantifying foci per cell or the number of cells with foci, as long as this quantification is sufficient to distinguish between our control samples, then we feel that this is scientifically valid way of quantifying data. Furthermore, many papers switch between different methods of quantifying genome stability depending on the assay used. Therefore, we disagree with the reviewer that altering the method of quantification according to the assay used should be a problem for readers being able to assess the quality of the data.

5. The data in Fig 3a is not described accurately in the text (lines 242-244). These data show that the presence of DIAPH1 decreases the % of cells with gH2AX foci slightly in G1 and G2 at 1 h post-IR, but mostly decreases the % of cells with gH2AX foci in G2 at 8 h. These data would be more consistent with the contribution of DIAPH1 to both the slow-acting HR in G2 and the fast-acting NHEJ in G1. Either way, these points should be described and interpreted more accurately. Also related to this point, the information on lines 246-248 is inaccurate since ETOP-induced DSBs can be robustly repaired by both HR and NHEJ. In addition, for the GFP-LMNA reporter, it is important to add some negative control such as the siRNA-mediated knockdown of an NHEJ repair factor. Similar NHEJ controls are necessary for Fig 3d-f and 4c.

Response: We would like to highlight that the difference in the mean number of foci per cell 1h post-irradiation in G1-phase and G2-phase patient P1+ vector cells versus patient P1+ DIAPH1 cells is: 24.7 versus 23.9 respectively for G1 phase cells and 52.9 versus 51.4 respectively for G2 cells. Therefore, despite the statistical significance when comparing H2AX foci per G1 or G2 cell in P1+ vector cells versus patient P1+ DIAPH1 cells at 1h post-irradiation, we do not believe this difference is biologically meaningful. Based on this, the only obvious difference in foci between irradiated P1+ vector cells versus patient P1+ DIAPH1 cells is in G2 cells at 8h post-irradiation, which is consistent with HR being affected. If NHEJ were defective there would still be an obvious DNA repair defect present at 8h post-irradiation in both G1 and G2 cells, which is not the case. Whilst NHEJ is 'fast-acting', a deficiency in this process gives rise to a massive DNA repair defect irrespective of cell cycle. As evidence of this, we have previously demonstrated that fibroblasts from patients with mutations in either XRCC4 or DNA Ligase IV exhibit a striking inability to resolve H2AX foci in G1 cells even at 24-post-irradiation (Murray et al., 2015, Am J Hum Genet. 96:412-424). This is in stark contrast to cells lacking DIAPH1, which can fully

resolve DSBs induced in G1 cells within 8h. Based on this, we strongly believe that a deficiency of DIAPH1 has no impact on NHEJ-dependent DSB repair.

In regard to the reviewer's comment that the information in lines 246-248 of our manuscript is inaccurate, we do not say that ETOP-induced DSBs are only repaired by HR. We state that we monitored sister chromatid exchange as a readout of productive HR in response to treatment with CPT or ETOP because they predominantly induce DSBs in S-phase, which is factually correct. Whether ETOP-induced DSBs can be repaired by NHEJ is not relevant to this statement, since NHEJ-repaired breaks do not lead to sister chromatid exchanges. Moreover, further on within our manuscript (lines 269-271), we even state 'We opted to use CPT since DNA DSBs induced by CPT are solely repaired by HR, whereas in contrast DSBs induced by IR or ETOP can be repaired by either NHEJ or HR, which would complicate our analysis'.

Lastly, it is not clear what the logic is behind requesting an NHEJ knockdown control for our HR reporter assay. By knocking down an NHEJ factor, either HR will remain unaffected or it will increase. As such, we are unsure how this further validates the function of DIAPH1 in promoting HR. The CRISPR-based, LMNA-mClover HR reporter assay has been fully validated by the Dellaire group, who generated this assay (Pinder et al. 2015, *Nucleic Acids Res.* 43:9379-9392), using knockdown and inhibition of both the HR and NHEJ pathways. Furthermore, we have used knockdown of BRCA2 and NBS1 as positive controls in our assays verifying the dependence on HR for mClover-LMNA expression. Consistent with our approach, the following papers (listed below) published in *Nature Communication* documenting the role of different factors in the regulation of HR also have only used positive controls in their HR assays. None of these studies included knocking down an NHEJ factor to validate their HR assays.

1. Tao et al. (2023). *Nature Commun.* 14:7430
2. Ambjorn et al. (2021). *Nature Commun.* 12:5748
3. Chakraborty et al. (2021). *Nature Commun.* 12:4126
4. Zhou et al. (2020) *Nature Commun.* 11:2639
5. Peng et al. (2019) *Nature Commun.* 10:1224
6. Lou et al. (2017) *Nature Commun.* 8:985

Based on this, we do not feel knocking down an NHEJ factor and redoing all our HR assays is necessary or would strengthen our conclusions that DIAPH1 regulates HR. Despite this, we were asked by reviewer 3 to assess HR in DIAPH1 depleted cells using the DR-GFP assay, so we have included an NHEJ knockdown control in this assay and demonstrated that depleting DNA ligase IV has no impact on the efficiency of HR (see figure in our response to reviewer 3).

6. Regarding Fig 4e, why was the statistical analysis not conducted in a way to compare IR/siControl/Mirin to IR/siDIAPH1? The combination of Mirin and siDIAPH1 with a more complete statistical analysis is necessary to evaluate the data accurately. Similarly, statistical analyses comparing plus-Vector to plus-DIAPH1 conditions should be included in Fig4a-b.

Response: As requested we have redone the statistical analysis on the data presented in Figures 4a, 4b, 4e and Supplementary Figure 6d.

7. The data in Fig S6d are puzzling. DIAPH1 does not appear to show any significant nuclear localization in the presence or absence of IR, although the quality of the microscopy data shown here appears limited. In fact, the authors show that DIAPH1 is primarily cytoplasmic regardless of DNA damage levels, suggesting that the protein may perform a cytoplasmic function to promote nuclear DSB repair. The study would greatly benefit from improved microscopy. Related to this point, are there NLS or NES signals within DIAPH1, and how does their mutation alter the protein's localization and function? And do any of the disease-linked DIAPH1 mutations fall within such localization signals? The study also necessitates improved microscopic analyses comparing the localization of DIAPH1, gACT, and bACT in the presence or absence of DNA-damaging agents to better understand how these factors may intersect in nucleus and/or cytoplasm during DSB repair.

Response: We agree with the reviewer that the majority of DIAPH1 is cytoplasmic. However, as stated in our manuscript, it has been previously shown that a small amount of DIAPH1 is also nuclear (Baarlink et al. 2013, *Science.* 340:864-867). Furthermore, we have verified this observation ourselves using cellular fractionation and Western blotting (Supplementary Fig. 8). DIAPH1 does contain both an NLS (amino acids 829-847) and an NES (amino acids 47-60). Currently, we do not have any patients with mutations in either the NLS nor NES. However, since all the patient mutations are destabilising, these would not be useful for studying the localisation of DIAPH1 in the presence/absence of DNA damage.

In response to a similar point raised by reviewer 1, we have used the FokI DSB assay to determine whether DIAPH1, γ -actin or β -actin are localised to sites of DSBs. This assay showed that endogenous DIAPH1 is robustly recruited to sites of FokI-induced DSBs (Fig.5a). Interestingly, DIAPH1 was only localised to approximately 50% of Mre11 positive, FokI-induced DSBs, suggesting that DIAPH1 may be involved in modulating the repair of a subset of DSBs by HR. In contrast we were unable to assess whether γ -actin or β -actin were recruited to FokI-induced DSBs due to their very high level of expression in cells masking any specific immunofluorescence staining at the DSB.

As an alternative approach to the FokI system, we have also used PLA to assess whether γ -actin or β -actin are localised to CPT-induced DSBs. Importantly, this assay demonstrated that endogenous DIAPH1 and γ -actin but not β -actin are localised to CPT-induced DSBs marked by γ H2AX but only in EdU positive cells (Fig.5b and Supplementary Fig.15). Therefore, this data is consistent with DIAPH1 and γ -actin working directly to regulate HR at sites of DSBs. We have incorporated this new data into the manuscript and altered the main text accordingly.

To extend our understanding of how DIAPH1 is recruited to DSBs, we have used the FokI assay again and either depleted MRE11 or inhibited ATM or MRE11 and assessed the efficiency of DIAPH1 relocalisation to FokI-induced DSBs. Strikingly, this demonstrated that loss or inhibition of ATM or MRE11 partially prevented DIAPH1 localisation to DSBs (Fig. 5d). This indicates that a feedback loop exists in which the MRN complex binds to DSBs and activates ATM. The activation of ATM helps to stimulate the recruitment of DIAPH1, which in turn helps to recruit or stabilise the MRN complex on chromatin surrounding a DSB undergoing resection. We have incorporated this data into a working model that we have presented as a diagram in the supplementary information (Supplementary Fig.20).

8. It is important to assess whether the reported protein-protein interactions of DIAPH1 can be recapitulated between endogenous proteins by colP and if assays such as PLA can detect interaction foci within the nucleus and/or cytoplasm in the absence and presence of IR.

Response: Unfortunately due to the low levels of DIAPH1 in the nucleoplasm (see Supplementary Fig.8), it is very difficult to immunoprecipitate enough nuclear DIAPH1 to observe a detectable interaction with the MRN complex, without over-expressing it. However, we have been able to detect an association of endogenous DIAPH1 with DSBs in the nucleus using the FokI nuclease assay (Fig.5a) and PLA (Fig.5b). Moreover, we have also been able to observe an association of endogenous γ -actin but not β -actin with DSBs using PLA (Supplementary Fig.15).

9. The statistical comparisons in Fig 7c/d are incomplete and confusing. I was looking to compare many more of the conditions shown on these plots, but the necessary statistical information was not provided.

Response: As requested we have redone the statistical analysis on the data presented in Figures 7c and 7d. Details of the statistical tests carried out are all indicated in the figure legends. Moreover, the raw data for all our quantification presented in this manuscript was supplied as part of our initial submission.

10. It would further strengthen the study to assess DSB levels and cell viability when DIAPH1-deficient cells are treated with PARP inhibitors.

Response: We have used the colony survival assay to assess whether cells depleted of DIAPH1 are sensitive to Olaparib. Notably, this assay demonstrated that cells depleted of DIAPH1 do not exhibit an increased sensitivity to Olaparib, unlike cells depleted of BRCA2 (see figure below). This indicates that whilst DIAPH1 and BRCA2 both regulate HR at DSBs and damaged replication forks, they play different roles in responding to PARP inhibition. Given that it is not clear what the replicating stalling lesion is when PARP1/2 are inhibited e.g. single ended-double strand breaks, reversed forks, trapped PARP1 on DNA, increased single strand gaps or increased R-loops, we don't believe that an absence of a hypersensitivity to Olaparib in DIAPH1 depleted cells weakens our conclusion that DIAPH1 and γ -actin play an important role regulating HR-dependent repair of DNA DSBs.

11. In lines 329-331 the authors are again pointing out general similarities between the BWCF syndrome and their described DIAL syndrome. Similarities have already been mentioned throughout the text multiple times. However, pointing out specific features of DIAL and how it differs from other actin-defect-related syndromes would be more helpful.

Response: We have included a paragraph in the discussion about the clinical deficits that differ between DIAL and BWCF patients (lines 475-491).

Minor points:

1. Sentence split on lines 166-167: It is recommended to either remove this sentence or publish this other study or deposit a preprint and cite it here.

Response: Originally, we had intended to include all the clinical information about the >30 patients with biallelic mutations in *DIAPH1* into this manuscript. However, when we assembled the manuscript, it became very clear that it was far too long. Furthermore, the focus of the paper became very confused i.e. is it a clinical paper describing the phenotypes and mutation status of a large cohort of patients or is it a cell biology paper focusing on the mechanisms underlying the DNA repair deficiency arising when *DIAPH1* was mutated. As such, we chose to focus the paper on the latter but we still needed to include some clinical information about the patients we had derived cell lines from, which had been used in our functional studies. Unfortunately the accompanying manuscript on *DIAPH1* describing the >30 patients with biallelic mutations is not quite ready to be deposited on BioRxiv. Since the Stewart lab is not leading on collecting all the clinical information and writing this manuscript, we do not have any control as to when it will be ready to deposit. Whilst we agree that it would be useful to reference an accompanying manuscript in the first section of the results, we feel that if we don't say that a description of the patients will be described elsewhere then it becomes difficult to justify why we have not included all the clinical information relating to the entire patient cohort in this manuscript. As such, we would prefer to keep this sentence in the paper.

2. Line 217: missing figure number.

Response: This has been rectified.

3. Regarding refs 32/33 and lines 234/235. The characterization of the evidence in these studies is not entirely accurate and should be revised.

Response: We are unsure as to what the reviewer is referring to as being 'not entirely accurate'. The reference by Andrin et al. (Ref 32) demonstrates that actin is important for regulating DSB repair using various inhibitors of actin polymerisation e.g. Latrunculin A, Swinholide and Cytochalsin D (Figures 4 and 5 of paper). Pfitzer et al. (Ref 33) demonstrates that the actin inhibitors, Latrunculin B and Jasplakinolide, affect NHEJ-, HR- and SSA-dependent DSB repair as demonstrated using specific reporter assays (Figure 3 of paper). Since these inhibitors are relatively non-specific, we believe that we are justified in saying that it is 'hard to identify how loss of individual factors that regulate actin nucleation affect the different DSB repair pathways' when these inhibitors are used. However, if the reviewer can be more specific about how our summation of the evidence presented by Andrin et al and Pfitzer et al is not entirely accurate, then we would be more than happy to alter the appropriate text.

4. Why are the HR reporter values in Fig 6f double those in Fig 3c for the same conditions?

Response: This assay is carried out by transiently transfecting cells with siRNA and then transiently transfecting the cells with 2 plasmids, one encoding Cas9 and one encoding the repair template

containing the mClover gene flanked by regions of homology to the *LMNA* locus. Since this protocol requires two rounds of transfection, we do see quite a bit of cell death and as such, there is some variability in the level of HR achieved depending on the transfection efficiency and cell viability etc. The data from Figure 3c was generated when the PhD student carrying out the experimentation first started using this assay in the lab and as such, we had not quite optimised the protocol to obtain the best transfection efficiency and quantification possible. The data presented in Figure 7f was generated later on during this project when we had managed to tweak the protocol to improve the level of detectable HR. Despite this, since we have used appropriate controls for each set of experiments shown in Figure 3c and Figure 7f i.e. a control siRNA and siRNAs to deplete BRCA2 and NBS1, we do not believe the difference in the absolute amount of HR between the two figures is that important. Rather, what is important is the reduction in HR caused by knocking down the gene of interest relative to the amount of HR observed in cells transfected with control siRNA. In both figures, the reduction in HR when either *DIAPH1*, *BRCA2* or *NBS1* are knocked down is approximately 2 fold lower than what is observed in the control siRNA transfected cells.

5. Sentence on lines 413-415: difficult to read/follow and should be improved for clarity.

Response: This has been rectified.

6. The text labels within the figures often show overlapping letters and truncated words (e.g. Fig 1c and 1e). This should be carefully reviewed and corrected throughout the figures.

Response: We apologise for this. Our manuscript went through several major iterations prior to submission and sometimes this makes it difficult to spot mistakes no matter how many times one looks through it. In addition, some of the figures had so much data in them, it made it difficult to cram everything into the same figure without some text overlapping each other. We have rectified these mistakes.

7. The statistical comparisons throughout the figures often fail to compare key conditions. The stats should be carefully reviewed for completion and accuracy.

Response: We apologise for the inconsistency of our statistical comparisons. We have been through these and altered them to compare the most appropriate aspects of the data. All information about what statistical tests were carried out are specified in the figure legends.

8. Not a single image includes a scale bar in the study.

Response: We apologise for this oversight and have added scale bars to all the representative immunofluorescence images. However, all the microscopy images were taken on the same microscope using the same magnification.

9. Results in Fig4a-b: should the y-axis be “MRE11/EdU colocalization” or similar? The “average MRE11 at DNA” is rather confusing. What exactly was quantified in the graph in Fig 4a? The shown images are overall not clear. Only EdU is visible in the nuclei. Are both images showing cells after CPT treatment? This information should be specified in the figure itself.

Response: We apologize for the confusion in terminology and agree with the reviewer regarding the images – we note that these images are in fact very large and detailed, but their quality and resolution were degraded in the conversion of the Figures when uploaded. We included high resolution images in the manuscript, and we hope these will be provided to the reviewers. As for the images and analysis, SMLM images present the coordinates of individual fluorescent molecules that were localized with an accuracy of several nanometers (shown as single pixels). In the manuscript we utilize a robust and unbiased data-mining statistical approach that measures the pair-wise distances between every localized molecule of one colour to all the molecules in a different colour, which is computed directly from the molecular coordinates. Based on all the measured pair-wise distances the algorithm then automatically generates the probability distribution of the distances (probably density) using a cross-correlation (or pair-correlation) function. To simplify, this provides a robust statistical measure for the distance distribution that only converge if there is non-random probability whereby a number of molecules of one colour are within a certain range (or distribution) of distances from the molecules of another colour. Importantly, if molecules are placed at random distances the probability will be zero even at incredibly high-densities of molecules since it will yield similar frequency for all distances, whereas even a small subset of molecules (within a field of high-density of randomly localized molecules) are positions within a range of distances from one another will yield a non-zero probability. This means that if there are few non-random events (statistically significant) where pairs of molecules are within (or near) a distance of 20 nm or 30 nm of each other, these are considered as being

associated with the same process or complexes, even though their specific molecular coordinates might not overlap as these are resolved at 10 nm. We therefore used the term “average” to indicate changes in the frequency of molecular pairs (DNA and MRE11 in this case) that are co-localized to these regions. In our measurements of replication forks and repair complexes, the range of distances we obtained are less than 100 nm, which also reflect the spread of the SLM signals including that of EdU at individual forks. Importantly, these distances are far below the diffraction limit of light, and as such will be considered as co-localization of overlapping clusters when analysed using standard confocal or epifluorescence microscopy. However, we emphasize that the present work builds on established methods, assays and computational tools that are described in great detail in our previous publications as well as work from other labs (see Sengupta et al. (2011) *Nature Methods*. 8:969-975; Veatch et al. (2012) *PLoS One*. 7:e31457; Coleman et al. (2022) *Nature Commun*. 13:1740; Yin et al. (2021) *Mol Cell*. 81:4243-4257; Yin et al. (2019) *Nature Commun*. 10:119; Chen et al. (2015) *Mol Cell*. 58:323-338).

10. The data presentation in Fig 6c-d is cumbersome and could be improved.

Response: We have reworked the figures to make them less cumbersome.

11. It was often difficult to determine exactly which cell type was used in every figure panel.

Response: The cell type used in every figure is clearly indicated in the corresponding figure legend.

12. Line 399: shouldn't it cite Fig 6f instead of Fig 6e?

Response: This has been rectified.

13. Including a schematic summary model summarizing the findings would be helpful. The model could also depict the order of the proposed repair events better illustrate how exactly DIAL appears to differ from other related but not identical syndromes/diseases.

Response: We have included a model of DIAPH1 and γ -actin potentially regulates HR-dependent of DSBs in the supplementary information (Supplementary Fig.20). We have also included a paragraph in the discussion about how DIAL syndrome differs from other actinopathies.

Reviewer #3 (Remarks to the Author)

In this manuscript, Stewart and colleagues propose a previously unrecognized regulatory role of DIAPH1 on the replication-associated DNA double-strand breaks (DSB) repair pathway by homologous recombination (HR). Significantly, this role was discovered by the systematic characterization of patient-derived cell lines from DIAL (DIAPH1 Loss-of-function) Syndrome patients that all carry homozygous frameshift or nonsense mutations in the DIAPH1 gene, which codes for a member of the formin family of actin nucleators. DIAL cell lines are defective for HR repair of replication associated lesions, probably due to defective initial steps of HR (i.e. DNA end resection). These observations may explain the significant overlap of clinical symptoms of DIAL Syndrome with Nijmegen and Warsaw breakage syndrome, both also HR deficiency disorders.

The strength of this manuscript lies in the careful and comprehensive analysis of patient-derived cell lines and their wildtype reconstituted derivatives. Together, these data strongly support the notion that loss-of-function mutations in DIAPH1 yields an increase in chromosomal instability by abrogating early steps of HR repair. Having said this, the manuscript also holds some weaknesses. For example, the mechanistic assessment of DIAPH1 deficiency on MRE11 recruitment is still rather preliminary and is not yet very convincing. However, in summary, this is not only the comprehensive characterization of a new human genetic disorder but also an important and timely contribution to the emerging role of nuclear actin on genomic stability maintenance. Together, these two elements make for a manuscript that is clearly suitable for publication in *Nat Commun* upon appropriate revisions (suggestions follow below).

Response: We thank the reviewer for their positive comments about our manuscript.

Main issues:

1) The authors convincingly show that absence of DIAPH1 causes a decrease in sister-chromatid exchange and gene targeting. However, both assays are not ideal HR readouts. For example, an HR-independent mechanism of sister chromatid exchange was recently proposed (Heijink et al., 2022, *Nat Commun*). A quasi-standard HR assay in the field is the GFP reconstitution assay (often referred to as the “Maria Jasin assay”). I would suggest the authors to perform this assay in cells depleted of DIAPH1 (siDIAPH1 cells).

Response: We agree that SCEs are not an ideal readout of HR as it can be mediated by Alt-EJ under very specific circumstances i.e. loss of BRCA2. However the CRISPR-based gene targeting assay is a *bona fide* HR reporter assay fully validated by the Dellaire lab (Pinder et al. 2015, Nucleic Acids Res. 43:9379-9392). We are unsure as to why the reviewer indicates that this assay is 'not ideal' as an HR readout. If anything, it is a more physiologically relevant readout of HR than the 'Maria Jasin assay' because it assesses the levels of HR at a DSB-induced at a specific endogenous gene locus rather than the Maria Jasin assay, which assesses HR at an I-SceI-induced DSB in a non-physiological reporter construct. Despite this, we have carried out the DR-GFP assay and shown that depletion of DIAPH1 reduces HR (see figure below). However, in contrast to the CRISPR-based gene targeting assay, we did not observe a large reduction in HR when DIAPH1 is depleted using DR-GFP assay. We suspect that the underlying reason for this relates to the context of the DSB i.e. an endogenous locus for the *LMNA/C* gene targeting assay versus a non-physiological reporter construct for the DR-GFP assay.

2) The MRE11 recruitment defect in DIAPH1 deficient cells is not very convincing (Fig. 4a). I do not see a clear difference in the STORM images between P1 + Vector and P1 + DIAPH1 cells. The magnified foci examples seem to show a difference in the vicinity of the MRE11 signal to the EdU signal, but it is not clear how this was quantified. Also, the increase in MRE11/EdU colocalization in P1 + DIAPH1 cells is minor. It is also difficult to comprehend why the STORM images represent MRE11 recruited to replication-associated breaks, while the normal IF images in Supplementary Fig 5 represent MRE11 recruitment to damaged chromatin. This distinction appears somewhat random. iPOND analysis has shown that the MRN complex is an intrinsic component of the DNA replication fork. Perhaps the absence of DIAPH1 leads to premature dissociation of MRN from stalled or damaged replication forks rather than defective recruitment of sites of replication-associated lesions. I propose to either investigate the effect of DIAPH1 loss on MRN recruitment more comprehensively or to tone down the MRE11 recruitment aspect and emphasize the defective DNA end resection, which may or may not be caused by a suboptimal recruitment of MRE11.

Response: We have updated the images in the revised manuscript. We note that these images are in fact very large and detailed, but their quality and resolution were degraded in the conversion of the Figures when uploaded. We included high resolution images in the manuscript, and we hope these will be provided to the reviewers. It is also important to emphasize that the SMLM images are provided as examples of the type of SMLM data obtained in our experiments but are not meant to be used for quantification of colocalization trends by means of visual inspection. To ensure robust and unbiased quantification of the specific molecular features and trends in these images, we utilized an automated images analysis pipeline that is applied directly on the raw unprocessed SMLM images that is capable of detecting even subtle changes in related patterns. The power of SMLM arises from its enhanced molecular detection sensitivity coupled to a molecular localization step, utilizing sequential and stochastic switching of fluorophores between a bright ("on") and a dark ("off") state. The centroid positions of the sparsely distributed active fluorophores can then be determined by mathematical models with nanometer precision. Analysis methods aimed at utilizing coordinate lists rather than rendered images have been developed to attain maximal molecular association information. Therefore, quantitative measures of molecules-of-interest, such as its count, density, spatial distributions, and

relationship (colocalization) within distinct structures, can be obtained at near-single-molecule precision.

Please refer to our response to point 9 raised by reviewer 2 that succinctly explains the analyses of two-color SMLM signal co-localization and related terminology (see Sengupta et al. (2011) *Nature Methods*. 8:969-975; Veatch et al. (2012) *PLoS One*. 7:e31457; Coleman et al. (2022) *Nature Commun*. 13:1740). We would like to draw the attention of the reviewer to the actual quantification of MRE11 recruitment in the DIAPH1 deficient cells, which is derived from multiple cells, multiple ROIs and multiple molecular clusters and uses an automated unbiased image analysis pipeline. This analysis shows an inability to recruit MRE11 to damaged replication forks in cells lacking DIAPH1. Briefly, our SMLM analysis utilizes a cross-correlation probability density function analysis between two species that does not rely on a specific distance threshold between two species for consideration of proximity. This analysis utilizes robust and unbiased computational framework that is based on pair-correlation function where it derives the probability density (or amplitude) as a function of all the pair-wise distances, providing a distance range rather than a finite distance. We emphasize that this sort of calculation does not utilize any distance threshold or data thresholding. In the present work this analysis revealed that the pair-wise interactions of fork associated events occurred on a scale (or range) of within 100 nm, which is the anticipated molecular scale that would encompass such events. We also note that there is a pre-extraction step that removes much of the cytosolic and soluble nuclear components to detect only chromatin bound proteins in our super resolution imaging. Based on this, we are unsure what the reviewer means by "It is also difficult to comprehend why the STORM images represent MRE11 recruited to replication-associated breaks, while the normal IF images in Supplementary Fig 5 represent MRE11 recruitment to damaged chromatin. This distinction appears somewhat random." The quantification of MRE11 colocalisation directly with an EdU spot is a measure of how much MRE11 is bound to a damaged replication fork. Quantification of the total 'extraction-resistant' pool of MRE11, whether it colocalises with EdU or not, is a measure of how much chromatin-bound MRE11 there is. However, we would like to bring to the reviewers attention that just because MRE11 is present at forks it does not mean that additional molecules of MRE11 are not recruited to the surrounding chromatin. To emphasize this, whilst it is well established that the MRN complex localises to DSB ends where it activates ATM and catalyzes DNA end-resection, additional MRN complex is recruited to chromatin surrounding a DSB by MDC1. Consequently, loss of MDC1 compromises MRN localisation to damaged chromatin (visualised as a loss of DNA damage-induced MRN foci) but it does not affect the binding of the MRN complex to DSB ends or the activation of ATM.

Finally, we would like to highlight the fact that the MRE11 recruitment defect in cells lacking DIAPH1 observed by STORM was validated using an independent approach based on inducing a site-specific DSB using the Fok1 nuclease. Using this system, we have demonstrated that cells depleted of DIAPH1 exhibit a significant defect in the ability to recruit MRE11 to sites of DSBs, which are generated by an enzyme and are not linked to DNA replication (Figure 4c and Supplementary 7a). Furthermore, we observed the same reduction in recruitment of MRE11 to FokI-induced DSBs when we depleted gamma-actin and components of the ARP2/3 complex (Figure 7b and Supplementary Figure 17a). Lastly, we have demonstrated that loss of DIAPH1 gives rise to a DSB end-resection comparable to the inhibition of Mre11 and that the resulting reduction in Rad51 foci associated with the loss of DIAPH1 is epistatic with the inhibition of Mre11. Based on this, we believe that the STORM data combined with the data generated using the FokI system strongly indicates that DIAPH1 and actin regulate the recruitment/retention of MRE11 at DSBs.

3) The specific role of gActin in HR is intriguing, given its similarity with bActin. This interesting and important observation could be investigated in more detail. For example, GFP-DIAPH1 seems to preferentially interact with g-Actin, while interaction with b-Actin is minimal (Fig 6a). Is this also the case for the reverse IP (i.e. GFP-gActin or GFP-bActin pull downs)? If yes, which one of the four distinct amino acids is required for the differential interaction? This could be determined by site-directed mutagenesis.

Response: We agree that the preferential binding of DIAPH1 with gamma-actin is interesting but unfortunately it is not quite as simple as DIAPH1 specifically binding to the divergent 4 amino acids at the N-terminus of gamma-actin but not beta-actin. It has been shown in mitotic cells that DIAPH1 and the related formin, DIAPH3, differentially regulate the formation of gamma-actin and beta-actin chains respectively in spatially distinct regions of mitotic cells (Chen et al. (2021) *Nature Commun*. 12:2409). Whilst It is not clear how either DIAPH1 or DIAPH3 selectively regulate gamma- and beta-actin chain formation, it is known to be dependent on the specific localisation of each formin to different regions of

the cell as well as the presence of a specific Rho family GTPase and activating factor such as IQGAP1, which converts DIAPH1 from an inactive to an active dimer capable of nucleating actin filaments. Furthermore, DIAPH-related formins do not simply bind globular actin directly. Globular actin is bound by a chaperone factor, called profilin, which binds to the FH2 linker region of DIAPH1 (Shah et al. (2024) Nature Commun. 15:5250). Somehow, profilin-bound actin is then moved from the linker region of DIAPH1 to its FH2 domain where nucleation of the filament takes place. Despite this understanding of how DIAPH1 regulates actin filament formation, it is still not known how DIAPH1 preferentially facilitates the formation of gamma-actin over beta-actin filaments. Profilin does not appear to differentiate between gamma- and beta-actin. Moreover, DIAPH1 can readily form beta- and gamma-actin filaments *in vitro* using recombinant proteins. Based on this, whilst it would be interesting to understand how DIAPH1 specifically regulates the formation of gamma-actin filaments and why the DSB repair response specifically requires gamma-actin filaments over beta-actin filaments, we feel this is well outside the scope of this manuscript.

Reviewer #1 (Remarks to the Author)

The revised manuscript is strengthened by the inclusion of new data and text in response to issues raised by the reviewers. A few things still need to be addressed.

1. Additional images should be provided for Fig 5a showing examples of no colocalization (and no FokI induction). Controls (omission of each antibody separately) are not provided for the new PLA assay (Fig. 5b).

Response: As suggested by the reviewer, we have now included the 'no FokI induction' and 'no colocalization' images in Supp. Fig.8b. Regarding the 'no antibody' control for Fig 5b, we would like to clarify that this is not typically considered the most informative control for a PLA assay. Similar to performing a Western blot without the primary antibody, a 'no antibody' control in PLA would likely result in a completely blank signal. Instead, a more appropriate control is to perform the PLA assay using both antibodies on cells where one of the antigens has been depleted (e.g. via siRNA depletion or CRISPR knockout), ensuring that any observed signal is specific. As shown in Figure 5b, we have already quantified the PLA signal for DIAPH1 and γ H2AX in the presence and absence of DIAPH1. We believe this effectively demonstrates the specificity of our assay. However, we appreciate the reviewer's input and hope this explanation clarifies our approach.

Prompted by the responses to reviewers, I looked at the source data file and found that in some cases, the source data is not adequately provided. For example, Excel tab for Fig 5a shows the percentage colocalization, which is already apparent in the figure. The raw data is not provided. Same for Fig 5d and Sup Fig 5a.

Response: We apologise for this oversight. We have updated the source data as requested.

The inclusion of foci numbers in a subset of assays is useful. Presenting the data by group leads to a loss of important information, and just because grouping is used in the literature does not make it best practice.

Response: We appreciate the reviewer's perspective and the opportunity to clarify our approach, but we respectfully disagree with this reviewer's comments. If quantifying foci by grouping is sufficient to distinguish between our positive and negative controls, we believe this method is robust and appropriate for drawing meaningful conclusions. While we acknowledge the reviewer's point that the presence of a method in the literature does not inherently make it best practice, we would also note that the same argument applies to quantifying foci per cell. Both methods are widely used in the field, and differences in preference do not necessarily indicate that one approach is more valid than the other. Importantly, multiple studies published in *Nature Communications* have employed our method of foci quantification, further supporting its scientific validity. Given this and the consistency of our results, we are confident that our approach is appropriate for the conclusions drawn in our manuscript.

Reviewer #2 (Remarks to the Author)

The manuscript is much improved, though some concerns remain.

Regarding Major Points 1 and 5:

Although the authors highlight some valuable points, the patient-derived fibroblast results differ significantly from the HeLa results. Specifically, the data presented for the siRNA DIAPH1 in HeLa cells suggest that DIAPH1 may still influence repair mechanisms with faster kinetics than homology repair (HR). Regarding Figure 3a-b, first, the authors should add the DIAPH1-dependent percent decreases in the number of γ H2AX foci per cell directly on the graphs to better highlight the magnitude of the change. That said, assessing cells at approximately 2-4 hours (i.e., expanding Figure 3a-b) would have been more informative; currently, one hour could be too early while eight hours could be too late to capture key differences in NHEJ or other fast-acting repair processes within the specific experimental setup used. Also, adding data from an NHEJ reporter would have more definitively eliminated the potential impact of DIAPH1 on this repair mechanism.

Response: As requested, we have added the mean number of foci per cell onto Figure 3a. However, with respect to adding additional timepoints to figure 3a, we do not feel this will improve the data that is

already presented or give insights as to whether DIAPH1 regulates NHEJ. As mentioned in our previous rebuttal, from our analysis of cell lines from *XRCC4* mutant patients, an NHEJ defect is evident in both G1 and G2 phase of the cell cycle even at 24 hour post-irradiation (Murray et al. 2015. *Am J Hum Genet.* 96:412-424). Therefore, if DIAPH1 loss did give rise to an NHEJ defect, it would be evident at the 8 hour timepoint in G1 cells, which is not the case. Consequently, we feel adding a timepoint at 2-4 hours post-irradiation would not provide any additional information about the repair defect in DIAPH1 null cells than isn't already presented in the figure. Based on this, we don't feel that carrying out analysis of NHEJ in the absence of DIAPH1 using an NHEJ reporter cell line is warranted.

In addition, the new DR-GFP reporter results presented in the authors' response to Reviewer #3 raise additional questions. The authors' preferred interpretation that the endogenously integrated LMNA reporter may be more accurate than Jasin's DR-GFP reporter is plausible. However, the weak effect of DIAPH1 loss in the DR-GFP assay compared to the LMNA assay might instead reflect multiple studies linking lamins to double-strand break (DSB) repair through various processes, as well as the well-established connections between LMNA mutations, genome instability, chromatin status, and ageing. In other words, this LMNA assay tests the repair of a locus whose products can facilitate DNA repair through different mechanisms. Consequently, inducing damage within the LMNA gene might give an initial global hit to DNA repair, which could then artificially amplify the effect of DIAPH1 loss on DSBR in this specific assay. Therefore, the DR-GFP assay results shown in the rebuttal letter should be incorporated into the manuscript for completion and transparency. Finally, without additional analyses of repair kinetics and DNA repair pathway choice, the authors could clearly state in the results or discussion section that:

- a. their data cannot entirely rule out the possibility that DIAPH1 is not involved in DNA repair pathways beyond HR.
- b. future research should further explore the potential impact of DIAPH1 on DNA repair pathway choice.

Response: We appreciate the reviewer's feedback on the CRISPR mClover-LMNA DNA repair reporter assay. However, we disagree with the reviewer's criticisms of this DNA repair reporter assay. The original manuscript describing this assay as an alternative to the DR-GFP assay for measuring HR-dependent DSB repair (Pinder et al. 2015, *Nucleic Acids Research*, 43(19):9379-9392) thoroughly validated the system using *BRCA1* knockdown, inhibitors of DNA ligase IV, and activators of Rad51. Additionally, Pinder et al. observed similar results when targeting the PML rather than the LMNA locus for CRISPR-dependent mClover integration as a readout for HR. Furthermore, the broad acceptance of this assay within the scientific community is reflected in its citation record, with the original manuscript having been cited 344 times. Given this strong validation and widespread use, we believe this assay is a well-established approach for assessing HR-dependent DSB repair. For complete transparency we have now included the DR-GFP assay graph in Supp. Fig.5f. That said, we appreciate the reviewer's suggestion and have revised our manuscript to include a statement acknowledging that our data does not entirely rule out a role of DIAPH1 in other DNA repair pathways (Lines 545–548). We hope this addresses the reviewer's concerns and enhances the clarity of our work.

Regarding Major Point 2: The new data, explanation, and statement in the discussion section that "DIAPH1 is essential for facilitating HR-dependent repair of DNA DSBs induced by IR, CPT, and ETOP" effectively address this concern. This statement should be retained in the final version of the paper. The authors should also consider briefly capturing this point in the abstract, particularly that the effect of DIAPH1 on DSBR has only been observed for, or may be limited to, exogenously induced DNA damage.

Response: We have altered the manuscript accordingly (Lines 545–548).

Regarding Major Points 6 and 9: While the authors have included some useful statistical analyses, a simple t-test seems to be employed throughout the manuscript to compare one sample against multiple other samples. However, the t-test is intended for comparing only two means at a time. Performing multiple t-tests separately for each comparison increases the risk of Type I errors (false positives) due to multiple comparisons. Thus, the statistical analyses still need attention. At a minimum, a warning should be added to the statistical analysis section of the methods to highlight this risk. Since most phenotypes discussed in the manuscript demonstrate strong effects, this reviewer will defer to the handling editor to determine the best course of action on the choice of statistical tests used by the authors.

Response: We thank the reviewer for their comment. To address this we have employed the help of someone with considerable experience using statistics to redo all the statistical analysis within our manuscript. Whilst the 'p-values' have changed, importantly the new statistical analysis has not changed the overall outcomes of any of the experiments presented within our manuscript.

Regarding Major Point 10: The data shown here are informative. Though it is not required, the authors should consider including them in the manuscript. This could be useful to the ongoing mechanistic debates surrounding PARP inhibition.

Response: We have altered the manuscript to include this data (Supp. Fig. 21).

Other points:

In the literature, DIAPH1 appears to show characteristics consistent with it being a candidate oncogene or a candidate tumor suppressor (e.g., see PMIDs 36503156, 24105619, 39879317, 26124177, 30094535). The authors refer to DIAPH1 as a tumor suppressor (e.g., see line 172). It is unclear to this reviewer if DIAPH1 meets the criteria for classification as a tumor suppressor or oncogene. What are the authors' thoughts on this point? This point should be better addressed in the text.

Response: We have based the labelling of DIAPH1 as a tumour suppressor because some of the affected DIAL syndrome patients have developed lymphoid tumours, which is consistent with an underlying DNA repair defect in cells from these patients. However, this does not rule out that DIAPH1 may act as an oncogene in some cancer types when over-expressed. Consequently, we have toned down the text in the discussion specifically labelling DIAPH1 as a tumour suppressor gene.

Supp Fig. 8a: Why does DNA damage induction appear to lower DIAPH1 levels in the nucleus?

Response: We think the very mild reduction of DIAPH1 levels in the nucleus after irradiation is due to slightly lower levels of protein loading in this sample. However, we don't think this is biologically relevant.

Supp Fig 18 b: This fig shows ACTB KO cells have statistically significant decreases in RAD51 foci formation. This appears to contradict the text on line 418.

Response: We have altered the manuscript accordingly.

Lines 195-197: Unclear sentence; should it be "Prompt g-H2AX recovery was restored or rescued in Patient P1 fibroblasts when complemented with WT DIAPH1"?

Response: We have altered the manuscript accordingly.

Lines 257-260: "To verify this finding, ... the cell cycle"; add reference(s).

Response: We have added an additional reference to the text as suggested.

Line 273: It should be "following exposure to CPT"; "to" is missing.

Response: We have altered the manuscript accordingly.

Line 340: The callout to the relevant figure(s) is missing; should it be Fig. 5d and Supp. Fig. 8b?

Response: We have altered the manuscript accordingly.

Line 529: correct "the formation of localised an F-actin"

Response: We have altered the manuscript accordingly.

Reviewer #3 (Remarks to the Author)

This is a re-review of the revised version of the manuscript: "Inherited deficiency of DIAPH1 identifies a DNA double strand break repair pathway regulated by gamma-actin" by Stewart and colleagues. My assessment of the first version of the paper was generally positive, with the suggestion of three main directions of further experimental work that would have strengthened the impact of the work.

Unfortunately, the authors did not address any of my suggestions satisfactorily. They did follow my suggestion to test the impact of DIAPH1 depletion in the "Maria Jasin assay", but the results show only a minor reduction in percentage of HR upon depletion of DIAPH1 as compared to e.g. BRCA2 depletion. This poses the question as to how relevant DIAPH1 really is for HR.

Response: We appreciate the reviewer's feedback but respectfully find the comment questioning the relevance of DIAPH1 in the repair of DSBs by HR unfounded. Throughout our manuscript, we have presented extensive data demonstrating that DIAPH1 is essential for DSB repair using multiple cell lines, including patient-derived cell lines. Specifically, we have shown that loss of DIAPH1 impairs the repair of DSBs induced by IR, etoposide, and camptothecin (CPT). Given that CPT-induced DSBs can **only** be repaired by HR, this provides strong evidence supporting DIAPH1's role in this pathway. Furthermore, we have demonstrated that DIAPH1 depletion prevents the MRN complex from localising efficiently to DSBs, leading to a reduction in DSB end-resection and Rad51 loading. This is further supported by results from both the CRISPR-based mClover-Lamin A/C DSB repair reporter assay and the sister chromatid exchange (SCE) assay, both of which show a reduction in HR-dependent repair when DIAPH1 is lost. We believe the wealth of data we have provided is more than enough evidence supporting a role for DIAPH1 in facilitating HR.

Regarding the DR-GFP assay, while it is a widely used DNA repair reporter assay, it monitors the repair of a non-physiologically induced DSB in an artificial repair substrate using a non-physiological template. In contrast, both the SCE assay and the CRISPR-based mClover-Lamin A/C DSB repair reporter assay assess HR-dependent repair at endogenous loci. As such, we believe that these assays offer more physiologically relevant insights into DIAPH1's role in repairing DSBs by HR. For the complete transparency we have now included the DR-GFP assay graph in Supp. Fig.5f. We appreciate the opportunity to further clarify these points and hope this explanation addresses the reviewer's concerns.

My criticism of MRE11 localisation was only textually addressed and new, more clear images were provided. To be honest, I still don't understand this assay and therefore, I won't further comment on it.

Response: We appreciate the opportunity to clarify how we used STORM to assess the localisation of MRE11 at stalled forks and damaged chromatin in the presence or absence of DIAPH1. Our data demonstrate that, in the absence of DIAPH1, MRE11 is less efficiently recruited to damaged forks, as marked by EdU, and shows a reduced localisation to damaged chromatin, as reflected by the quantification of extraction-resistant MRE11 per nucleus. Furthermore, these findings are supported by our observation that MRE11 recruitment to FokI-induced DSBs is also diminished when DIAPH1 is lost. Taken together, our results provide strong evidence that DIAPH1 is important for the proper localisation of MRE11 to sites of DNA damage. We hope this explanation clarifies our approach and findings.

My final suggestion on site-directed mutagenesis and reverse IP of DIAPH1 with b-Actin and g-Actin was not addressed at all, with the explanation that this was "well outside the scope of this manuscript".

Response: We indicated in our previous rebuttal that we cannot carry out the reverse IP with tagged gamma- or beta-actin to assess differential binding to DIAPH1 as tagging actin blocks its ability to be nucleated in filaments, which would affect its association with DIAPH1. Furthermore, we also indicated that the selective binding of gamma-actin to DIAPH1 is not specifically related to the 4 amino acids that differ between gamma- and beta-actin. As such the reviewer's request to 'identify which one of the four distinct amino acids that differ between gamma- and beta-actin is required for the differential interaction with DIAPH1 using site-directed mutagenesis' is not possible. As mentioned in our previous rebuttal, since it is still not clear after decades of research in the actin field how the Diaphanous-related formins preferentially utilise the different actin paralogs, it is unreasonable to request that we try to identify the underlying mechanism. Given that 'simply mutating one or more of the four amino acids that differ between gamma- and beta-actin' will not address how DIAPH1 selectively utilises gamma-actin for nucleation during the DNA repair process, we felt that it was appropriate to say that this was 'outside the scope of our manuscript'. We hope this explanation is sufficient as to why these comments cannot be addressed in our manuscript.